# Selective gene expression maintains human tRNA anticodon pools during differentiation

Lexi Gao[1,3], Andrew Behrens[1,3], Geraldine Rodschinka[1], Sergio Forcelloni[1], Sascha Wani[1], Katrin Strasser[1] & Danny D. Nedialkova [1,2] ✉

Transfer RNAs are essential for translating genetic information into proteins. The human genome contains hundreds of predicted tRNA genes, many in multiple copies. How their expression is regulated to control tRNA repertoires is unknown. Here we combined quantitative tRNA profiling and chromatin immunoprecipitation with sequencing to measure tRNA expression following the differentiation of human induced pluripotent stem cells into neuronal and cardiac cells. We find that tRNA transcript levels vary substantially, whereas tRNA anticodon pools, which govern decoding rates, are more stable among cell types. Mechanistically, RNA polymerase III transcribes a wide range of tRNA genes in human induced pluripotent stem cells but on differentiation becomes constrained to a subset we define as housekeeping tRNAs. This shift is mediated by decreased mTORC1 signalling, which activates the RNA polymerase III repressor MAF1. Our data explain how tRNA anticodon pools are buffered to maintain decoding speed across cell types and reveal that mTORC1 drives selective tRNA expression during differentiation.

During protein synthesis, ribosomes match messenger RNA codons to their cognate amino acids via base pairing with complementary anticodons in transfer RNAs. These short (approximately 76 nucleotides (nt)) adaptor molecules are highly similar in sequence and structure, and much more abundant than ribosomes in cells[1], which enables rapid mRNA decoding. Their similarity poses a challenge as ribosomes must faithfully distinguish tRNAs that may differ by a single nucleotide but carry distinct amino acids. The relative abundance of charged tRNAs matching a given codon can modulate the rate and fidelity of mRNA translation[2], and changes in the levels or function of individual tRNAs have been linked to cancer and neurological diseases[3].

The tRNA repertoires of human cells and the mechanisms that control them remain largely unknown. This is due to the multicopy nature and simple promoter structures of tRNA genes, which makes their regulation challenging to predict or quantify. It is similarly challenging to accurately quantify mature tRNA transcripts because of their stable structure and abundant chemical modifications, which has resulted in poor characterization of tRNA expression in specific cellular contexts. There are 619 predicted tRNA genes in the human

nuclear genome, which can potentially generate 432 unique tRNA transcripts from 57 tRNA anticodon families[4]. RNA polymerase III (Pol III) is directed to nuclear-encoded tRNA genes by transcription factor IIIC, which binds to short intragenic promoter elements (A box and B box) and recruits transcription factor IIIB (TFIIIB) to the variable regions upstream of tRNA loci[5–8]. TFIIIB then positions Pol III for initiation and can retain it for multiple rounds of transcription[9–11]. Because of this simplistic regulatory model, tRNA gene copy number is often used as a proxy for tRNA expression levels. However, although nearly all tRNA loci in yeast are transcribed during rapid growth[12,13], Pol III enrichment at tRNA genes varies among mammalian tissues[14–18]. The molecular basis and quantitative impact of this selective transcription on tRNA levels are unknown. The lack of high-resolution measurements of tRNA levels has also led to controversy over their role in setting elongation rates in mammalian cells[19,20].

Here we address these questions by combining modification-induced misincorporation tRNA sequencing (mim-tRNAseq)—a method we recently developed to quantify the abundance of mature tRNA with high accuracy and resolution[21,22]—with ribosome profiling

[1]Mechanisms of Protein Biogenesis, Max Planck Institute of Biochemistry, Martinsried, Germany. [2]Department of Bioscience, TUM School of Natural Sciences, Technical University of Munich, Garching, Germany. [3]These authors contributed equally: Lexi Gao, Andrew Behrens. ✉e-mail: nedialkova@biochem.mpg.de

and Pol III chromatin immunoprecipitation–sequencing (ChIP–Seq). We find that tRNA repertoires are extensively remodelled following the differentiation of human induced pluripotent stem cells (hiPSC) into neuronal and cardiac cell types. These changes, however, are not driven by altered codon usage across the transcriptome and they have a minimal impact on tRNA anticodon availability, which remains largely stable. Mechanistically, differential Pol III occupancy at tRNA loci determines mature tRNA levels and is driven by sequence features in the tRNA gene body and 5′ flanking regions. Decreased mTORC1 signalling following differentiation activates the Pol III repressor MAF1, which restricts Pol III to a subset of tRNA genes we define as 'housekeeping'. We find that these genes are stably expressed and constitute the most abundant isodecoders in each anticodon family. This mechanism underlies the broad stability of tRNA anticodon pools and decoding rates in different cell types despite tRNA-repertoire remodelling during differentiation.

## Results

### Differentiation extensively remodels tRNA transcript pools

To define the composition of human tRNA pools in physiological settings, we designed a workflow using a reference hiPSC line (*kucg-2*)[23], which circumvents the genetic variability of immortalized human cell lines[24,25] and the changes in Pol III regulation following cellular transformation[26–29]. We used established small molecule-based protocols to direct hiPSC to differentiate into cardiomyocytes (CM)[30,31] or dividing neuronal progenitor cells (NPC) from which we then obtained neurons[32,33] (Fig. 1a). This isogenic panel contains cell types that are particularly affected by dysregulated tRNA metabolism in human diseases[3]. Immunostaining demonstrated the expected cell morphology and uniform expression of the pluripotency markers POU5F1 and SOX2 in hiPSC, the neural progenitor markers PAX6 and NESTIN in NPC, the neuronal markers MAP2 and CHAT in neurons, and α-actinin-2 and cardiac troponin T in CM, confirming culture purity (Fig. 1b). Distinct and characteristic transcriptomic signatures in these cell lines and the robust expression of defined marker genes were also confirmed through RNA sequencing (RNA-Seq; Fig. 1c and Extended Data Fig. 1a,b).

We then profiled the abundance of mature tRNA in these isogenic cell types using mim-tRNAseq[21,22], which enables the accurate quantitation of individual tRNA transcripts (Fig. 1d). We obtained approximately 80% uniquely mapped and ≤2% multi-mapped reads for all samples, a median of approximately 80% of which were full length and >95% of which contained the post-transcriptionally added 3′ CCA tail (Extended Data Fig. 1c–e), indicating that they were derived from mature and translationally competent tRNAs. Less than 4% of uniquely mapped reads were derived from mitochondrial tRNAs (Extended Data Fig. 1f), and we obtained single-transcript resolution data for 373 of the 413 (90%) predicted nuclear-encoded tRNA transcripts in our curated reference. Seven of the remaining transcripts had <10 mapped reads and the others were mostly only distinguishable at sites that can carry misincorporation-inducing nucleotide modifications, which precludes read deconvolution (for example, tRNA-Pro-AGG-1 and tRNA-Pro-AGG-2; Extended Data Fig. 1g)[22]. To validate our ability to capture known instances of differential tRNA abundance, we examined tRNA-Arg-UCU-4, which is highly expressed in the nervous system of mice[34] and humans[35]. This expression pattern was recapitulated in our workflow, as the proportion of reads mapping to tRNA-Arg-UCU-4 was significantly higher in neurons compared with hiPSC, NPC and CM (Extended Data Fig. 1h).

We next used DESeq2 (ref. 36) to compare the expression of individual tRNAs in differentiated cells relative to hiPSC. Principal component analysis demonstrated high reproducibility among replicates. The first principal component (reflecting cellular differentiation) accounted for 89% of the variation; tRNA transcript pools could also accurately distinguish cell types (Fig. 1e). Of the 373 unique tRNA

transcripts we could deconvolute, 161 showed significant differences in expression of up to about 70-fold in one or more differentiated cell types compared with hiPSC (adjusted *P* (*P*adj) ≤ 0.05; Fig. 1f). The changes measured by mim-tRNAseq were highly concordant with northern blotting analysis performed as validation for three transcripts (Extended Data Fig. 1i,j). Among the remaining tRNAs 205 had zero or low counts in all cell populations (<0.005% of tRNA-mapped reads; Supplementary Table 1). These data demonstrate that tRNA transcript pools in human cells are extensively remodelled during differentiation.

### Transfer RNA anticodon levels are largely stable across cell types

To define how this reprogramming impacts the abundance of tRNA anticodon families (Fig. 1d), we aggregated uniquely mapped tRNA reads by anticodon before DESeq2 analysis. Among the 57 anticodon families encoded by the full set of predicted human tRNA genes, we found no evidence of expression for nine, while 47 were robustly expressed across all cell types and tRNA-Ile-GAU was only detectable at very low levels in hiPSC (0.002% of uniquely mapped reads compared with 2.9% for tRNA-Ile-AAU and 0.8% for tRNA-Ile-UAU). The different cell types were well-resolved by principal component analysis of anticodon-aggregated data (Fig. 1g) and 46 anticodon families were differentially regulated in at least one cell type (*P*adj ≤ 0.05; Fig. 1h and Supplementary Table 1). However, the differences in abundance of the tRNA anticodon families were much smaller compared with those for individual tRNA transcripts, which is consistent with moderate variability among tRNA anticodon pools across mouse tissues[37]. Apart from a strong decrease for the poorly expressed tRNA-Ile-GAU, the largest changes were for tRNA-Gly-CCC (increased by 1.7–2.5-fold) and the selenocysteine-inserting tRNA-SeC-UCA (increased by threefold); all other changes were between 0.7- and 1.7-fold. There was no separation of anticodon pools among proliferating (hiPSC and NPC) and non-dividing (neuron and CM) cells or a substantial overlap of tRNA anticodon changes in NPC, neurons and CM with previously identified differentiation- or proliferation-linked tRNAs[38] (Fig. 1h and Supplementary Table 1). Although tRNA-Arg-UCU-4-1 was strongly upregulated in neurons (approximately 40% of tRNA-Arg-UCU), the abundance of this anticodon family decreased by approximately 1.4-fold because other isodecoders were downregulated (Extended Data Fig. 1k,l and Supplementary Table 1). Thus, despite extensive reprogramming of tRNA transcript pools, the availability of tRNA anticodons in human cells remains largely unchanged following differentiation.

### Codon usage and decoding rates do not vary across cell types

We next investigated whether the small but significant differences in tRNA anticodon pools among cell types are linked to differences in codon demand as previous studies on its potential coordination with tRNA supply have reached conflicting conclusions[17,18,38]. Codon usage weighted for expression correlated significantly with tRNA anticodon abundance (Pearson's correlation coefficient (*r*) = 0.57–0.66; Fig. 2a) and was strikingly stable across all four cell types (coefficient of variation (CV), 0.77–13.22%; Fig. 2b), in accordance with previous data from mouse and human tissues[17,18]. Notably, the codon usage of mRNAs that are highly abundant in all cells correlated significantly more strongly with tRNA anticodon levels than that of cell type-specific mRNAs that are expressed at high levels (Extended Data Fig. 2a,b). Despite their remodelling during differentiation, tRNA anticodon pools are thus equally well adapted to global codon demand across cell types.

To test whether the small differences in tRNA anticodon abundance we detected altered the decoding rates, we performed ribosome profiling[39] in hiPSC and NPC using cycloheximide to inhibit elongation and tigecycline to block tRNA entry into empty ribosomal A sites[40]. This yielded two predominant ribosome footprint sizes: 20–23 nt (short) and 28–33 nt (long; Extended Data Fig. 3a). The A-site (but not P-site) codon dwell times were strongly anticorrelated with the abundance of

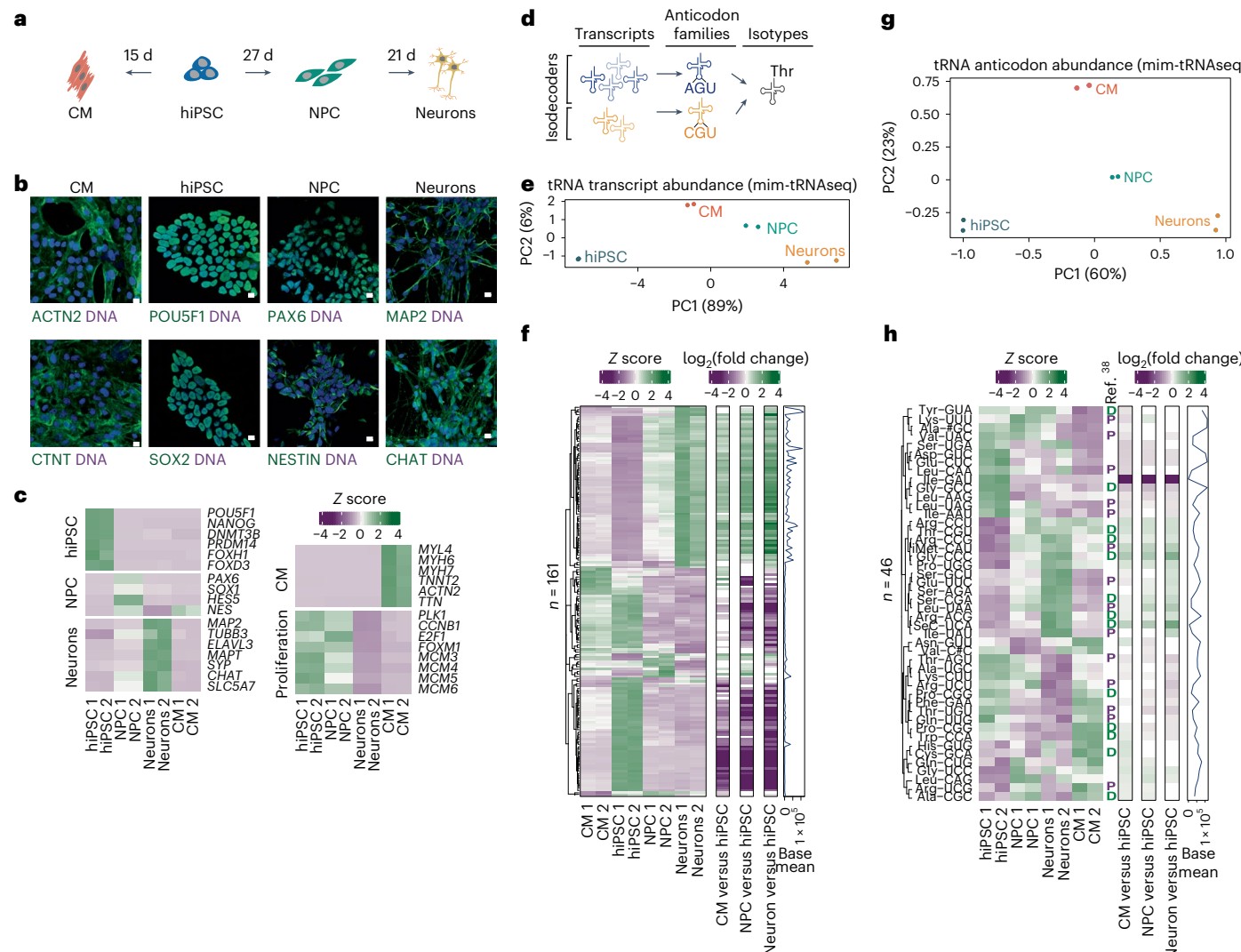

**Fig. 1 | Transfer RNA anticodon pools are maintained largely stable across cell types despite extensive reprogramming of tRNA repertoires during differentiation. a**, Schematic of hiPSC differentiation into NPC, neurons and CM. **b**, Representative fluorescence microscopy images (from at least three independent experiments) of immunostaining for cell type-specific marker proteins and DAPI. CTNT, cardiac troponin T; ACTN2, α-actinin-2. Scale bars, 10 μm. **c**, Gene expression heatmaps for known cell type- and proliferative state-specific markers in hiPSC, NPC, neuron and CM cultures (*n* = 2 biological replicates). Standardized *Z* scores were calculated using DESeq2-normalized RNA-Seq gene counts across samples. **d**, Schematic depicting tRNA classification. Distinct tRNA transcripts sharing an anticodon are called isodecoders and collectively constitute anticodon families. Members of different anticodon families that carry the same amino acid belong to the same isotype. **e**, Principal component (PC) analysis of variance-stabilizing-transformed count data for

cognate tRNA anticodon families in both hiPSC and NPC (Fig. 2c,d and Extended Data Fig. 3b), demonstrating the key role of tRNA anticodon availability for decoding rates in human cells. Unlike in yeast[40,41] the correlation in human cells was stronger for long footprints (Fig. 2c,d); it was much more modest for short (Pearson's *r* = 0.32) and not significant for long footprints (or P sites) when tigecycline was not added to the cell extracts (Extended Data Fig. 3c–e).

The A-site codon dwell times in NPC and hiPSC were highly correlated (Pearson's *r* = 0.9) and showed no clear relationship with differences in cognate tRNA abundance (Fig. 2e). For example, the tRNA-Gly-CCC levels were 1.7-fold higher in NPC cells but the average

tRNA transcripts from DESeq2 for each cell line (*n* = 2 biological replicates; variance explained by each principal component in parentheses). **f**, Heatmap of tRNA transcript expression dynamics showing only differentially expressed transcripts in at least one differentiated cell line relative to hiPSC (Benjamini–Hochberg-adjusted Wald test, *P*adj ≤ 0.05). Hierarchically clustered expression heatmap showing the scaled *Z* score of normalized unique transcript counts in hiPSC, NPC, neurons and CM (*n* = 2 biological replicates; left). Differential expression for NPC, neurons and CM relative to hiPSC reported as log₂-tranformed fold changes (middle). Base mean normalized to the tRNA transcript across all samples (right). **g**, Principal component analysis as in **e** calculated from variance-stabilizing-transformed count data summed by tRNA anticodon. **h**, Heatmap as in **f** for count data summed by tRNA anticodon. Anticodon families previously[38] associated with proliferating (P) or differentiated (D) cells are shown. Source numerical data and unprocessed blots are provided.

dwell time of ribosomes at GGG increased by only 5% (Extended Data Fig. 3f). We next compared the A-site codon dwell times for mRNAs that are expressed at high levels in both hiPSC and NPC (shared, *n* = 393) or that are cell type-specific (*n* = 114 in hiPSC and *n* = 80 in NPC; Extended Data Fig. 2a). Although approximately 20% of the ribosome footprints originated from shared mRNAs, <3% mapped to the cell type-specific transcripts, resulting in much less concordant codon dwell times between replicates (Extended Data Fig. 3g). Accordingly, the A-site codon dwell times for shared mRNAs were more highly correlated between NPC and hiPSC than the dwell times for cell type-specific mRNAs (Pearson's *r* = 0.87 versus 0.54) but the higher variance of

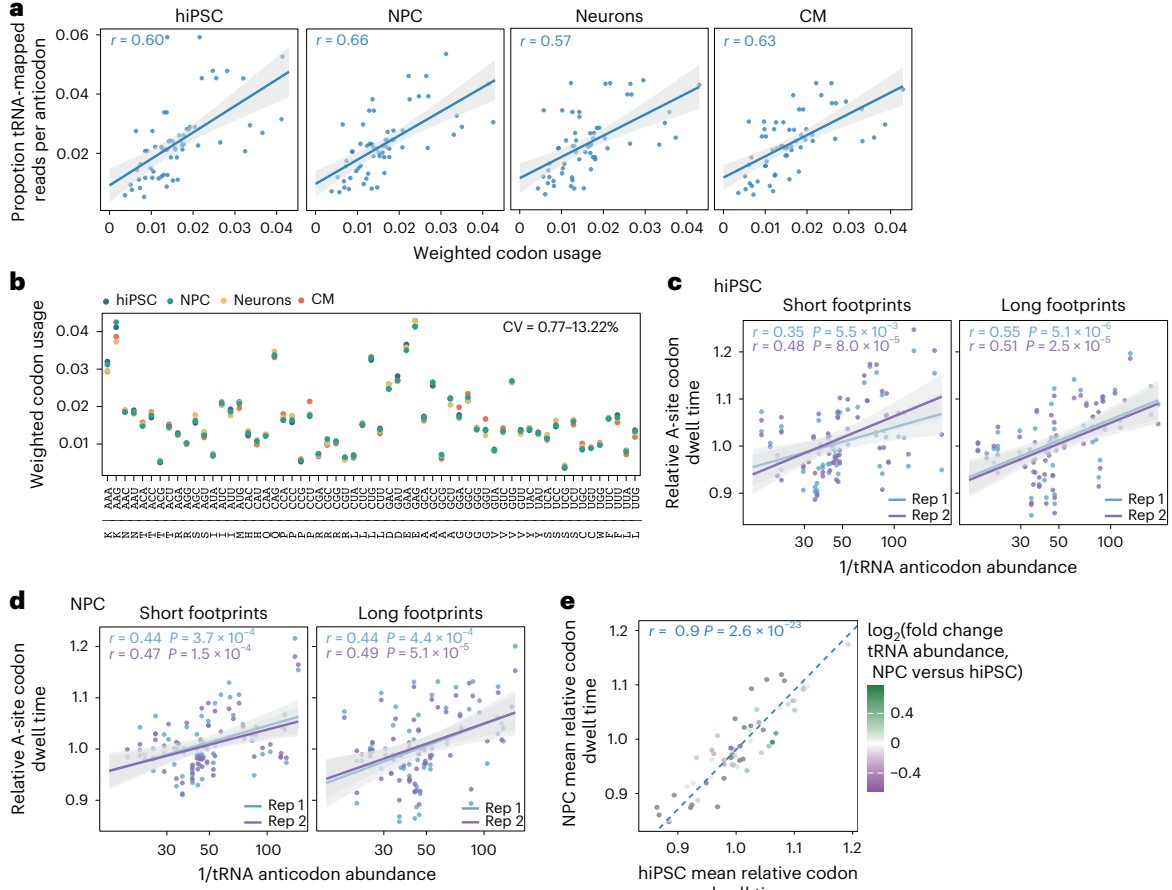

**Fig. 2 | Transfer RNA anticodon levels correlate with invariant codon demand and decoding rates across cell types. a**, Correlation of mean weighted codon usage to mean tRNA anticodon abundance from mim-tRNAseq scaled to the proportions of total tRNA-mapped reads ($n = 2$ biological replicates for each cell type). **b**, Aggregated codon usage weighted by mRNA expression (in transcripts per million, TPM) across all transcripts based on RNA-Seq data. The values shown per codon are the mean weighted codon frequencies per cell type, represented as the proportion of total codon usage. The range of CV per codon is indicated. Codon sequences and their corresponding amino acids (single-letter code) are provided. **c,d**, Correlation between the reciprocal of tRNA anticodon abundance calculated from the proportion of all tRNA-mapped reads to relative

A-site codon dwell time measured separately for short (20–23 nt; left) and long (28–33 nt; right) footprints from hiPSC (**c**) and NPC (**d**) libraries prepared with cycloheximide and tigecycline ($n = 2$ biological replicates denoted as rep1 and rep2). **a,c,d**, Solid lines, linear regression model; grey shading, 95% confidence interval. The Pearson's correlation coefficients are provided. **e**, Correlation between mean relative A-site codon dwell time in hiPSC and NPC from **c,d** (long footprints), coloured by $\log_2$ fold changes in tRNA anticodon abundance in NPC relative to hiPSC, $P$adj $\leq 0.05$; grey dots denote non-significant changes; dashed line, $y = x$; the Pearson's correlation coefficient is indicated. Source numerical data are provided.

the latter was not explained by differences in cognate tRNA levels (Extended Data Fig. 3h,i). Therefore, the divergence of tRNA anticodon pools between NPC and hiPSC is not sufficient to substantially alter decoding speed.

### Buffering of tRNA anticodon levels through major isodecoders

We investigated whether the stark disparity in the magnitudes of changes between tRNA transcript and anticodon abundance is due to the unequal contribution of distinct isodecoders to mature tRNA pools. To test this, we calculated the number of 'major' isodecoders that cumulatively contribute ≥90% of the tRNA-mapped reads for each anticodon family. Although the number of isodecoders in predicted human tRNA genes varies between one and 26, most mature tRNA anticodon families were comprised of 1–4 major isodecoders in hiPSC and only up to two in NPC and neurons (Fig. 3a,b). For example, the human genome encodes nine tRNA-Ala-UGC isodecoders, six of which are detectable in mature tRNA pools from hiPSC and two of which became predominant in differentiated cells (Fig. 3c). Similarly, three of the five

predicted tRNA-Pro-UGG isodecoders are expressed in hiPSC and two of those account for most mature tRNA-Pro-UGG in differentiated cells (Fig. 3c). Globally, most minor isodecoders (>70%) were strongly downregulated in differentiated cells (by up to 70-fold); conversely, most major isodecoders were upregulated, albeit much more modestly (approximately 1.2–4-fold; Fig.3d). Thus, tRNA anticodon pools in different human cell types are buffered through major isodecoders that are more stably expressed.

### Pol III occupancy at tRNA genes predicts mature tRNA levels

To define the regulatory mechanisms that favour the expression of major tRNA isodecoders, we generated genome-wide Pol III occupancy maps using ChIP–Seq for the Pol III catalytic core component RPC1 (ref. 42) and BRF1, the TFIIIB subunit that recruits Pol III to tRNA[5] (Fig. 4a). As expected, the ChIP signals for both RPC1 and BRF1 were highly localized and overlapped with predicted tRNA genes (Fig. 4b), whereas BRF1 signal was absent from the spliceosomal small nuclear RNA gene *RNU6-1*, which recruits Pol III via a BRF2-containing TFIIIB complex[5] (Extended Data Fig. 4a). Strikingly, we obtained a near-perfect linear

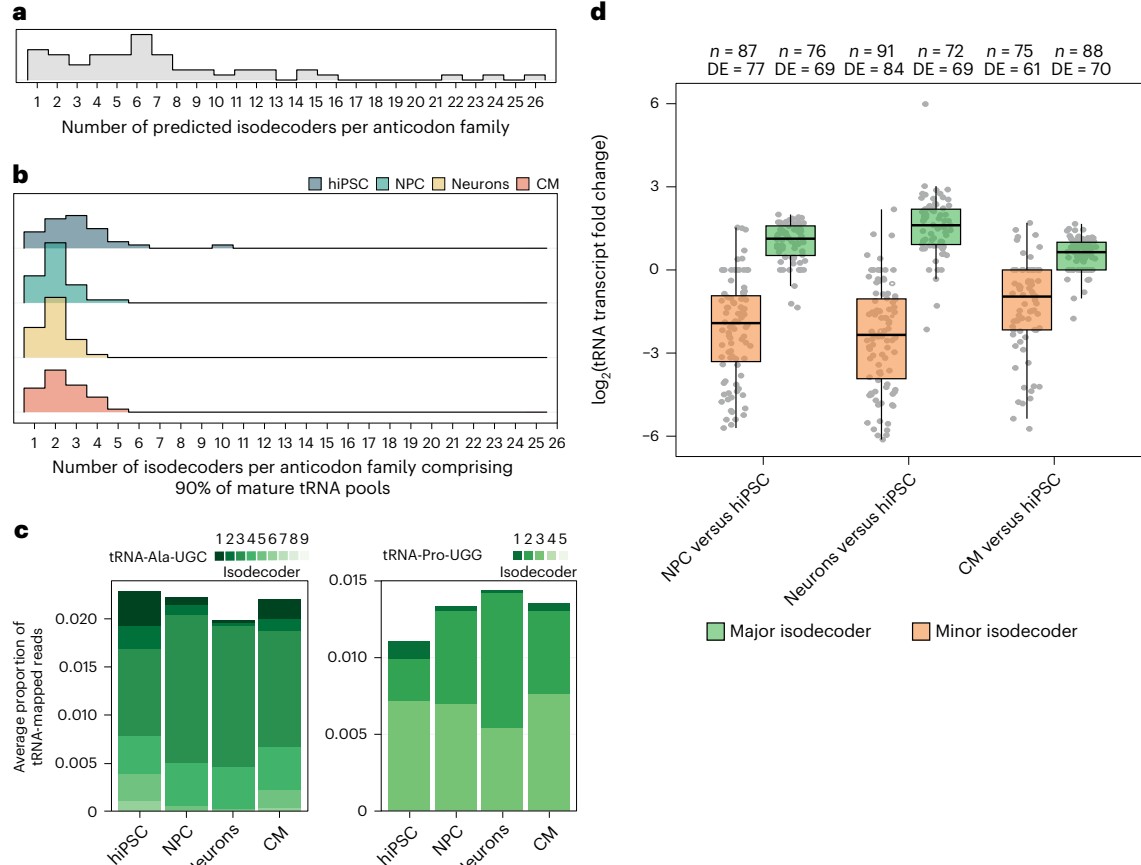

**Fig. 3 | Human tRNA anticodon pools are buffered through stably expressed major isodecoders. a**, Distribution of gene copy numbers per isodecoder in predicted human tRNA genes. **b**, Distribution of isodecoder count cumulatively constituting ≥90% of each anticodon per cell type (mean read proportions from mim-tRNAseq; $n = 2$ biological replicates) for detectable anticodon families with fully resolved unique transcripts. **c**, Proportional isodecoder composition (mean from $n = 2$ biological replicates) for tRNA-Ala-UGC (left) and tRNA-Pro-UGG (right). **d**, Change in tRNA expression (Benjamini–Hochberg-adjusted Wald test, $P$adj ≤ 0.05) for transcripts that are detectable in at least one cell type (≥0.005% of tRNA-mapped reads). DE, differentially expressed. Box plots: centre line, median; box limits, upper and lower quartiles; whiskers, 1.5× the interquartile range. Source numerical data are provided.

correlation between RPC1 ChIP–Seq signal strength aggregated by transcript and tRNA levels, measured by mim-tRNAseq, in all the cell types we tested ($r^2 = 0.88$–0.9; Fig. 4c and Extended Data Fig. 4b). Differences in Pol III occupancy thus explain nearly all of the variation in mature tRNA abundance in hiPSC as well during their differentiation.

**Differentiation restricts Pol III to housekeeping tRNA loci**
To determine how the Pol III-transcribed tRNA repertoire changes during differentiation, we performed genome-wide peak identification in the RPC1 and BRF1 ChIP–Seq datasets. We found a nearly complete overlap between peaks at predicted tRNA genes for the same protein across biological replicates as well as between consensus RPC1 and BRF1 peaks in the same cell type (Extended Data Fig. 4c,d). Defining consensus tRNA peaks and filtering out tRNA genes with ≥25% ambiguously assigned reads enabled single-gene resolution analysis of 558 of the 619 (90%) predicted human tRNA genes.

Based on the striking reduction in the number of Pol III peaks at tRNA genes we observed following differentiation (Fig. 4d), we defined three distinct classes of human tRNA genes. The first comprised genes occupied by Pol III in all cell populations ($n = 205$), which we defined as housekeeping. This set encodes transcripts from all 47 anticodon families with detectable expression in the mim-tRNAseq datasets (Supplementary Table 1). Housekeeping genes represented 70 and 94% of major isodecoders in hiPSC and neurons, respectively (Extended Data Fig. 4e). The second set included tRNA genes that were not bound by Pol III in any cell type ($n = 159$), which we called 'inactive'. The third set included tRNA genes at which a significant RPC1 ChIP peak in hiPSC is lost in one or more differentiated cell populations ('repressed'; $n = 194$, Supplementary Table 2). The largest set of tRNA-overlapping RPC1 ChIP peaks was in hiPSC ($n = 397$), and differentiated cells contained subsets of these peaks (Fig. 4e and Supplementary Table 2). No tRNA genes gained RPC1 ChIP peaks specifically in CM, whereas peak sets from NPC and neurons each contained one cell type-specific peak. Consistent with these data, none of the mature tRNA transcripts that were present in a differentiated cell type were undetectable (<0.005% of tRNA-mapped reads) in hiPSC (Supplementary Table 1).

To rule out cell line-specific effects, we performed RPC1 ChIP–Seq in other hiPSC and immortalized cell lines. Of the 397 RPC1 tRNA peaks in *kucg-2* hiPSC, 362 were detectable in an independent reference line, *wibj-2* hiPSC[23], as well as in HEK293T cells (Extended Data Fig. 4f). Datasets from *kucg-2* and *wibj-2* contained 24 tRNA peaks that were not detected in HEK293T cells, whereas only ten tRNA peaks were present in HEK293T or *wibj-2* but not in *kucg-2* cells. Approximately one-third of the predicted human tRNA genes are thus not bound by RPC1 in two independent hiPSC lines and the immortalized HEK293T cells. Note that 97% of housekeeping genes (199/205) were predicted to be active by a random forest classifier trained on tRNA gene sequence and genomic context[43] (Extended Data Fig. 4g). However, almost half of the tRNA genes that we found to be bound by Pol III in hiPSC and repressed during differentiation were not predicted as active by this approach.

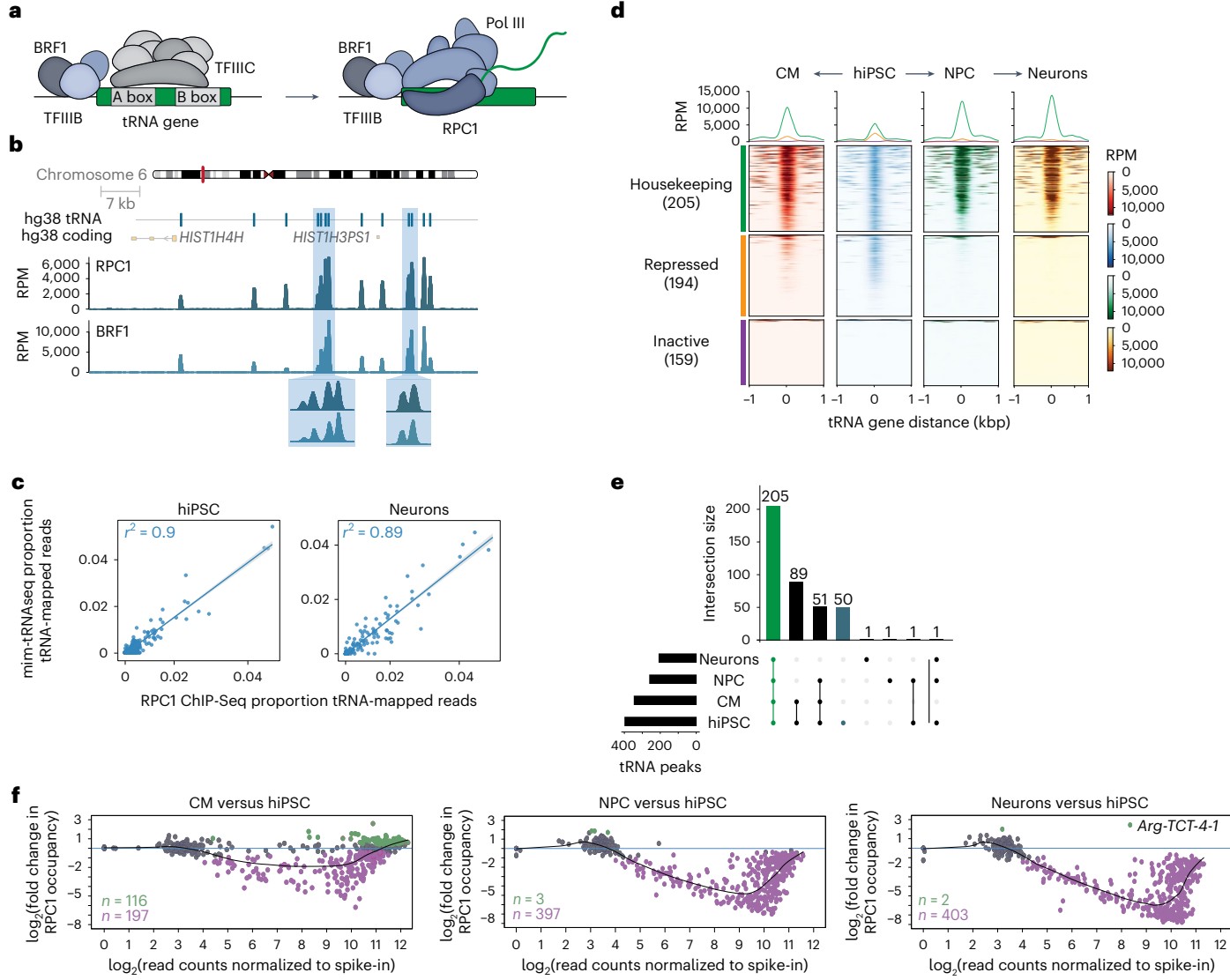

**Fig. 4 | Differentiation restricts Pol III to a housekeeping tRNA gene set encoding major isodecoders. a**, Schematic of Pol III recruitment to tRNA genes. **b**, Representative ChIP–Seq signal at predicted tRNA genes in a locus on human chromosome 6 for RPC1 and BRF1. Data are from one biological replicate of hiPSC normalized to the estimated library sizes from counts over extended tRNA features (±125 bp) and scaled to reads per million (RPM). Insets: magnified views of regions around closely spaced tRNA loci. **c**, Correlation between tRNA abundance estimated by mim-tRNAseq ($n = 373$ unique transcripts) and RPC1 ChIP–Seq signal aligned to extended tRNA features (±125 bp) for hiPSC and neurons (mean from $n = 2$ biological replicates for each cell type; scaled to the proportions of total tRNA-mapped reads in each dataset). Solid blue lines, linear regression model; grey shading, 95% confidence interval; the Pearson's correlation coefficients are indicated. **d**, Heatmaps of normalized RPC1 ChIP–Seq signal around tRNA gene start sites (±1 kbp) for single replicates. The tRNA genes were separated into housekeeping, repressed and inactive based on significant peaks in RPC1 ChIP–Seq data (FDR-adjusted $P \leq 0.05$). **e**, UpSet plot of significant consensus RPC1 peaks (±125 bp) of annotated tRNA genes (FDR-adjusted $P \leq 0.05$). Housekeeping tRNAs are shown in green and hiPSC-specific tRNAs in blue; the numbers in each group are indicated. **f**, Spike-in-normalized RPC1 ChIP–Seq read counts over tRNA features (±125 bp) versus the $\log_2$-transformed fold change in CM (left), NPC (middle) and neurons (right) relative to hiPSC presented as MA plots ($n = 2$ biological replicates for each cell type). Green and purple denote significantly higher and lower occupancies, respectively (FDR-adjusted $P \leq 0.05$). Source numerical data are provided.

To capture both global and gene-specific regulation[44], we analysed differential RPC1 binding to tRNA genes after spike-in normalization[45]. Pol III occupancy at tRNA genes was significantly reduced in differentiated cells compared with hiPSC ($n = 197$ in CM, $n = 397$ in NPC and $n = 403$ in neurons; false-discovery rate (FDR) $\leq 0.05$; Fig. 4f). The largest effect sizes were for genes with low-to-medium RPC1 signal, mirroring the decrease in low-abundance tRNA transcripts (Figs. 1f and 3d). The RPC1 signal increased by less than threefold at 116 tRNA genes with mid to high occupancy in CM. By contrast, only two tRNA genes had significantly higher occupancy in neurons: one did not pass the peak-calling threshold due to low counts and the other (*tRNA-Arg-TCT-4-1*) encodes

the neuron-specific tRNA-Arg-UCU-4 (Fig. 4f). Differentiation is thus accompanied by a general reduction in Pol III binding to tRNA genes, which disproportionately affects lower-occupancy loci. This is not accounted for by an overall reduction in Pol III abundance because the levels of its core subunits RPC1 and RPC2 decreased only modestly in neurons and CM (Extended Data Fig. 4h).

## Chromatin remodelling at tRNA genes following differentiation

To determine how chromatin state impacts tRNA gene activity[46–48], we profiled nucleosome-free regions (NFRs) using the assay for

transposase-accessible chromatin with sequencing (ATAC–Seq)[49] and performed ChIP–Seq for the trimethylation of histone H3 at K4 (H3K4me3; which flanks the transcription start sites (TSS) of active Pol II genes)[50], K27 (H3K27me3; which marks Pol II genes repressed in specific cell states)[51] and K9 (H3K9me3; which marks repeat-rich heterochromatin)[52]. We observed a marked concordance between the presence of H3K4me3 and RPC1 ChIP signal at tRNAs, in line with previous data[46,47] (Fig. 5a). Although the NFR signal generally coincided with the bodies of RPC1-bound tRNA genes, it was a worse predictor of mature tRNA levels than RPC1 ChIP signal, particularly in differentiated cells ($r^2 = 0.56$ for NFR ATAC–Seq versus $r^2 = 0.9$ for RPC1 ChIP–Seq in neurons; Fig. 4c and Extended Data Figs. 4b and 5a,b). Selective loss of RPC1 occupancy coincided with a loss of NFR and H3K4me3 signals as well as the appearance of H3K27me3. By contrast, inactive tRNA genes were in closed chromatin with weak H3K9me3 enrichment and not marked by H3K4me3 or H3K27me3 (Fig. 5a and Extended Data Fig. 5a). Analysis of whole-genome bisulfite sequencing datasets from human embryonic stem cells[53] revealed near-complete CpG methylation at inactive tRNAs (Extended Data Fig. 5c), suggesting that DNA methylation may contribute to their silencing.

Nearly half of predicted human tRNA genes cluster on chromosomes 1 and 6; we thus investigated whether nearby loci modulate Pol III occupancy. The majority (80%) of housekeeping and repressed tRNA genes were in proximity to other active tRNA genes (median distance of $0.96 \times 10^3$ and $3.69 \times 10^3$ base pairs (bp)), which could facilitate the concentration of active Pol III in transcription 'factories'[54] to enable its recycling. By contrast, inactive tRNAs were more distant from other tRNAs (median of 380.5 kbp; Fig. 5b). Although half of the tRNAs with gene-resolved RPC1 ChIP data were either within (234/558, 42%) or near coding genes (≤500 bp; 44/558, 8%), their linear distance from active or inactive Pol II genes was not related to RPC1 occupancy (Extended Data Fig. 5d). Active human tRNA genes are thus most often in close proximity to each other but, in contrast to previous reports[46,55], we found no clear association between Pol III signal and nearby Pol II activity.

### Sequence-dependent features underlie tRNA gene regulation

To test whether selective tRNA gene expression is driven by sequence-dependent mechanisms, we first examined the relationship between RPC1 occupancy and overall bit scores from the tRNA gene prediction tool tRNAScan-SE[4]. All housekeeping genes surpassed the 55-bit score threshold suggested to distinguish functional tRNAs based on anticodon–isotype congruence, A- and B-box consensus match and secondary structure[4], whereas 131 of the 159 (82%) inactive genes fell below this threshold (Fig. 5c and Extended Data Fig. 6a). However, 45 tRNA genes with bit scores of <55 had RPC1 peaks in hiPSC and there were no detectable RPC1 peaks at 28 loci with bit scores of >55 (Fig. 5c), confirming that the tRNAScan-SE score alone is not an accurate predictor of tRNA gene expression potential[43].

To quantify the contribution of A- and B-box promoters to differential RPC1 binding, we first compared both promoters separated by activity status (Extended Data Fig. 6b). In line with previous data from mouse liver[16], we found a high degree of sequence similarity and only subtle sequence differences across the three tRNA gene groups. To quantify these differences, we defined a consensus sequence for each promoter based on all predicted human tRNA genes (Extended Data Fig. 6c). There was a significantly higher density of both the A- and B-box consensus sequences in housekeeping tRNAs relative to repressed and inactive tRNAs (Fig. 5d), which suggests that subtle differences in A- and B-box promoters contribute to differential Pol III occupancy across human cell types. To experimentally validate this prediction, we replaced the housekeeping *tRNA-Pro-TGG-2-1* gene with the repressed *tRNA-Pro-TGG-1-1* in hiPSC using clustered regularly interspaced short palindromic repeats (CRISPR)–CRISPR associated protein 9 (Cas9) gene editing. These isodecoders differ by three nucleotides, one of which is in the A box (Fig. 5e), and the RPC1 ChIP signal at *tRNA-Pro-TGG-1-1* is lost following differentiation to NPC. Conversely, the RPC1 occupancy at *tRNA-Pro-TGG-2-1* increased (Extended Data Fig. 6d). ChIP–Seq revealed a strong reduction in RPC1 occupancy at the edited *tRNA-Pro-TGG-2-1* locus in both hiPSC and NPC but comparable signal at the neighbouring *tRNA-Pro-AGG-2-4* and the unedited *tRNA-Pro-TGG-1-1* (Fig. 5e), confirming the importance of gene body sequences for Pol III occupancy strength.

Interestingly, the RPC1 ChIP signal at the edited *tRNA-Pro-TGG-2-1* locus still increased in NPC compared with hiPSC, indicating that gene body sequences are not the sole determinant of transcriptional activity. In many instances identical tRNA genes with different flanking sequences are indeed characterized as different activity classes and become selectively occupied by Pol III during differentiation (for example, *tRNA-Tyr-GTA-5* copies; Extended Data Fig. 6e). In such cases, differences in the 5′ flanking sequence might result in differential TFIIIB recruitment. As transcription initiates at a variable distance from the tRNA gene start[11], matching position weight matrix models would miss over-represented motifs in these regions. We therefore adapted the BPNet convolutional neural network (CNN) architecture[56], which can predict the sequence specificity of DNA-binding factors from experimental data[56–59], to build a CNN called tRNet for predicting BRF1 binding from upstream tRNA sequences (Extended Data Fig. 6f). Trained using a 200-bp 5′ flanking sequence and tRNA activity status (housekeeping, repressed and inactive), tRNet was highly accurate in predicting tRNA activity in unseen data (75–78% across all folds) and could confidently distinguish housekeeping (area under the receiver operating characteristic (AUROC) = 0.91) and inactive (AUROC = 0.92) tRNA genes relative to all other classes, whereas repressed tRNAs were comparably more difficult to classify (AUROC = 0.81; Fig. 5f). Sequence motif detection[56,60] revealed that

**Fig. 5 | Transfer RNA gene body and upstream sequences govern differential Pol III recruitment. a**, Representative heatmaps (bottom) and metagene profiles (top) of the ChIP–Seq signal for RPC1, H3K4me3 (K4me3), H3K27me3 (K27me3), H3K9me3 (K9me3) and NFRs from ATAC–Seq around tRNA gene start sites (±1 kbp) for hiPSC and neurons, separated by tRNA gene activity. **b**, Distances between tRNA genes from different activity classes to their nearest tRNA gene. Box plots: centre line, median; box limits, upper and lower quartiles; whiskers, 1.5× the interquartile range. **c**, Relationship between RPC1 occupancy at tRNA genes (mean from $n = 2$ biological replicates) and the predicted tRNAScan-SE score, separated by tRNA gene activity. Dashed lines, median tRNAScan-SE score and RPC1 occupancy; solid blue line, 55-bit score tRNAScan-SE threshold for functional tRNAs. **d**, Two-dimensional binned kernel motif density of the A (left) and B (right) box of each tRNA gene ($n = 558$) separated by tRNA activity (centre line, median). Motif counts for density estimation were based on a 90% match to the consensus motif. Dot colour is used to indicate whether the tRNAScan-SE score is above or below the 55-bit threshold for predicted functionality (Wilcoxon

test). **e**, Schematic of CRISPR–Cas9 editing to replace *tRNA-Pro-TGG-2-1* with *tRNA-Pro-TGG-1-1* (left). Fraction of tRNA-mapped RPC1 ChIP–Seq reads at wild-type and CRISPR-edited hiPSC and NPC ($n = 2$ biological replicates for each cell type; bar, median) at the indicated tRNA genes. **f**, Receiver operating characteristic for tRNet performance on test data for each task. **g**, Top three significant TF-Modisco sequence motifs (FDR-adjusted $P \le 0.01$) for housekeeping tRNA genes. **h**, Schematic of CRISPR–Cas9 editing to insert the 100-bp sequence upstream of *tRNA-Pro-TGG-1-1* in front of *tRNA-Pro-TGG-2-1* (left). Fraction of tRNA-mapped RPC1 ChIP–Seq reads at wild-type (data from **e**) and CRISPR-edited hiPSC and NPC ($n = 2$ biological replicates; bar, median) at the indicated tRNA genes (right). **i**, Mean expression of *CADM3* in RNA-Seq datasets across cell types ($n = 2$ biological replicates). **j**, Representative plot of normalized ChIP–Seq signal for RPC1 ChIP, K4me3 ChIP and RNA-Seq signal surrounding the *tRNA-Arg-TCT-4-1* and *CADM3* genes in hiPSC and NPC as well as neurons. WT, wild-type; edit, CRISPR-edited. Source numerical data are provided.

GC-rich sequences and polyA stretches in 5′ flanking regions drive the predictive ability of tRNet for housekeeping tRNAs (Fig. 5g). Consistent with this, gene activity predictions based on chromatin state have suggested a regulatory role for GC content around tRNA loci[43], whereas a polyA stretch may enhance DNA binding by TFIIIB through its TATA-binding protein subunit. By contrast, the upstream regions of inactive genes were enriched for polyT stretches (Extended Data Fig. 6g), which constitute Pol III termination signals that inhibit tRNA transcription in vitro[7]. To experimentally test whether tRNA 5′

flanking sequences can alter Pol III binding, we inserted the 100-bp sequence preceding *tRNA-Pro-TGG-1-1* (GC content of 30%) directly upstream of *tRNA-Pro-TGG-2-1* (GC content of 60%) in hiPSC using CRISPR–Cas9. The RPC1 and BRF1 ChIP signals were reproducibly decreased at *tRNA-Pro-TGG-2-1* in NPC harbouring this edit (Fig. 5h and Extended Data Fig. 6h), corroborating the predictions from tRNet. Collectively, our data indicate a combinatorial effect of intragenic and 5′ flanking sequence features in determining Pol III occupancy at individual human tRNA genes.

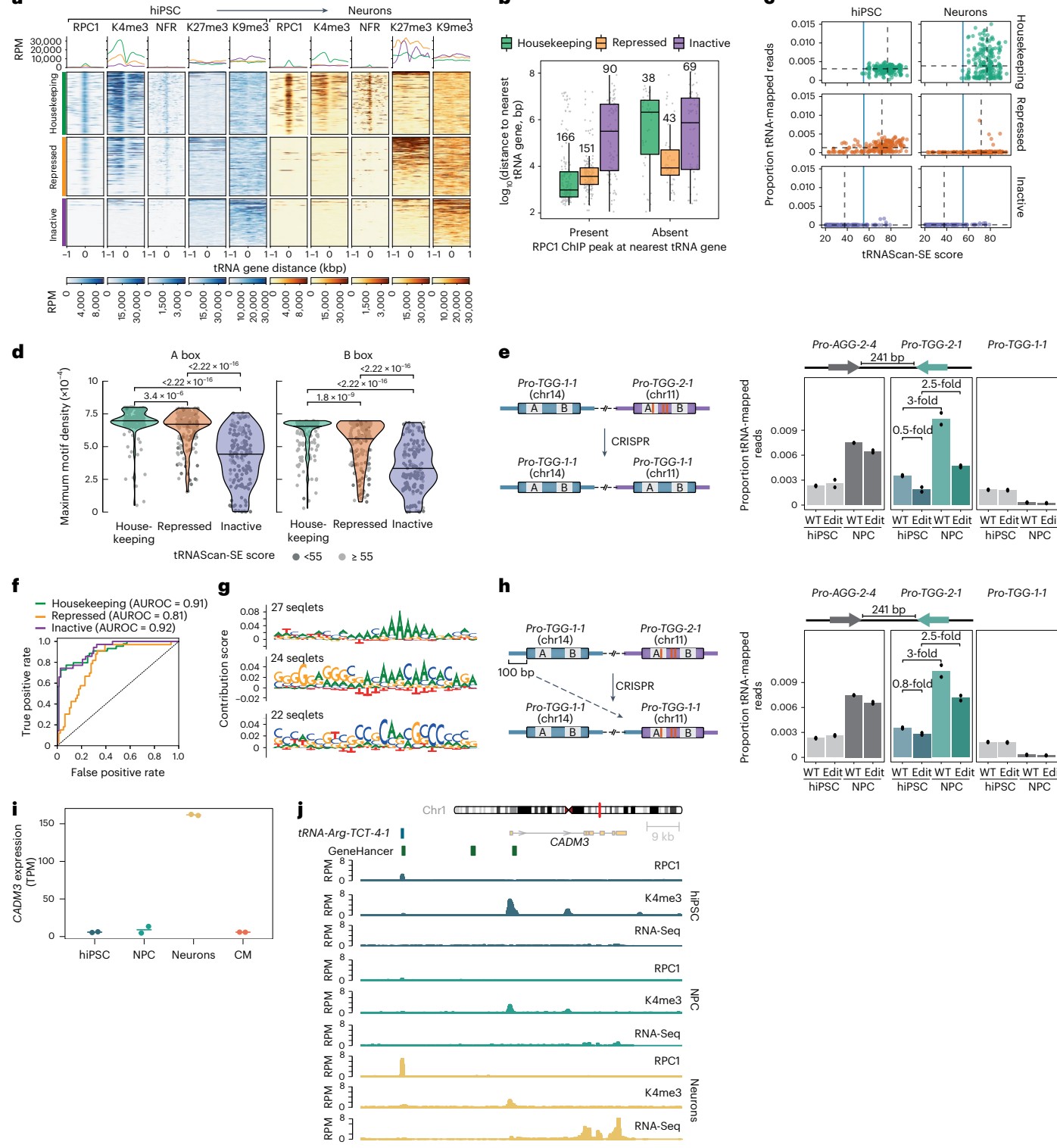

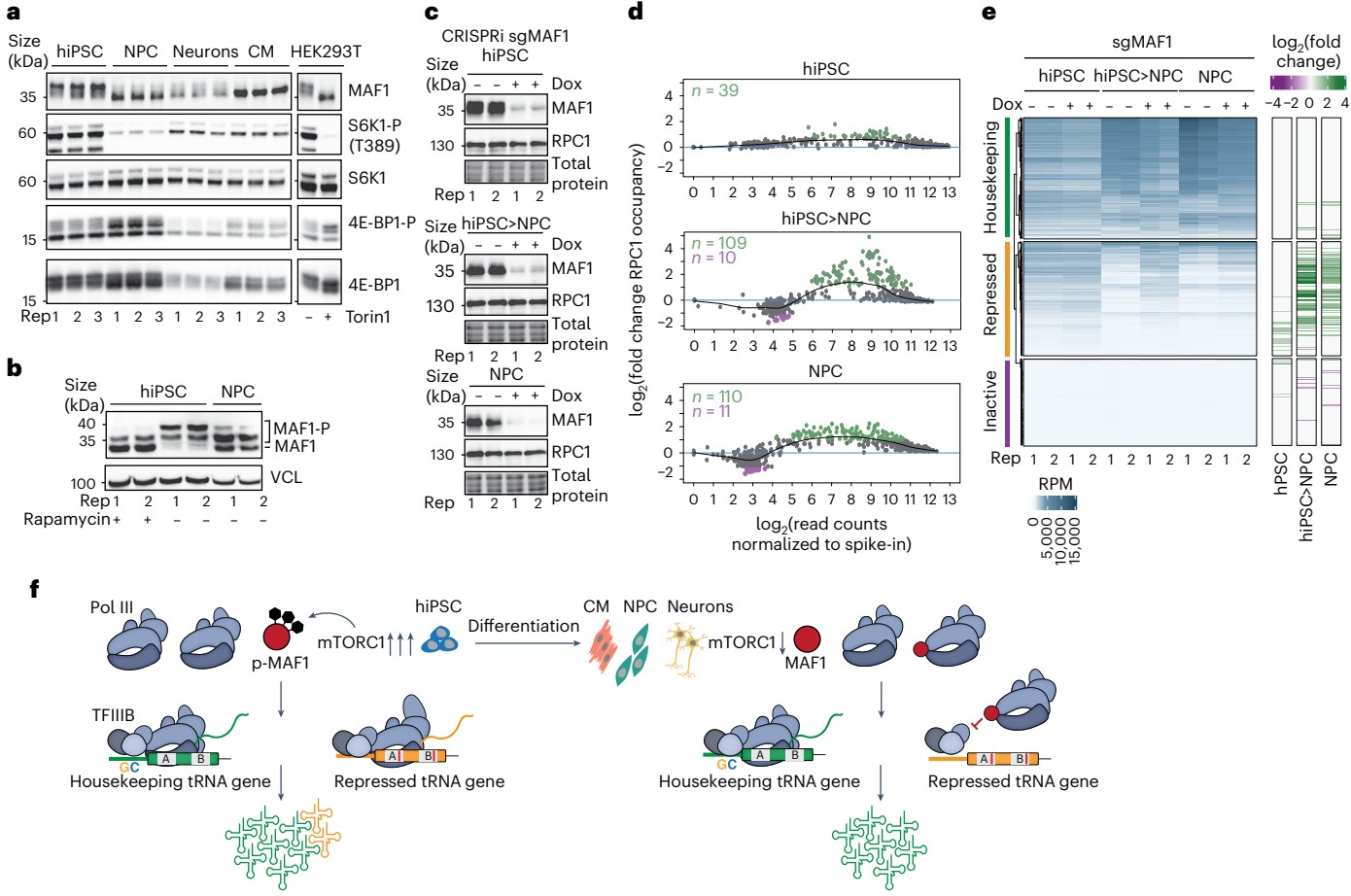

**Fig. 6 | Diminished mTORC1 signalling in differentiated cells triggers MAF1-dependent selective tRNA gene repression. a**, Immunoblots of MAF1, phospho-S6K1 (S6K1-P) and phospho-4E-BP1 (4E-BP1-P) in hiPSC, NPC, neurons and CM (*n* = 3 biological replicates). Samples from both untreated and Torin 1-treated HEK293T cells (250 nM; 1 h) served as controls for mTOR inhibition. **b**, Immunoblot of a Phos-tag gel for MAF1 in hiPSC, NPC and hiPSC treated with 10 nM rapamycin for 8 h (*n* = 2 biological replicates). Vinculin (VCL) served as a loading control. **c**, Immunoblots of MAF1 in CRISPRi lines carrying an sgRNA targeting *MAF1*. Gene knockdown was induced by the addition of 2 μM doxycycline (Dox) for 3 (hiPSC; top) or 6 d (NPC; bottom). For the hiPSC>NPC samples (middle), 2 μM Dox was added to *MAF1* sgRNA-containing hiPSC for 3 d, followed by NPC derivation under continuous Dox treatment (*n* = 2 biological replicates). **d**, MA plots (generated by DiffBind) of spike-in-normalized RPC1 counts over tRNA features (±125 bp) versus the log$_2$-transformed fold change

for Dox-induced hiPSC (top), hiPSC>NPC (middle) and NPC (bottom) samples carrying an sgRNA targeting *MAF1* relative to uninduced controls (*n* = 2 biological replicates for each cell type). Significantly higher and lower occupancies (FDR ≤ 0.05) are shown in green and purple, respectively. **e**, Heatmap of RPC1 ChIP–Seq occupancy changes following MAF1 depletion by inducible CRISPRi (+Dox) relative to uninduced controls (−Dox). RPC1 ChIP–Seq read counts over extended tRNA features (±125 bp; left). Normalized signal, accounting for estimated library sizes, was generated from these counts scaled to RPM. DiffBind differential occupancy analysis using spike-in normalization for induced samples relative to the corresponding uninduced controls reported as the log$_2$-tranformed fold change (right; *n* = 2 biological replicates for each cell type and condition; FDR-adjusted *P* ≤ 0.05). **f**, Model of selective tRNA expression following hiPSC differentiation. Rep, replicate. Source numerical data and unprocessed blots are provided.

## *tRNA-Arg-TCT-4-1* is co-regulated with *CADM3* in neurons

The *tRNA-Arg-TCT-4-1* gene, which is upregulated in neurons, represented a rare case of strong selectivity given the significant decrease in RPC1 occupancy at all other tRNA genes in comparison with hiPSC (Fig. 4f). It was classified as a housekeeping gene based on the presence of a significant RPC1 ChIP peak in consensus sets for all cell types (Supplementary Table 2), suggesting that it is also active in non-neuronal cells. Given that its A- and B-box sequences are identical to those in other tRNA-Arg-UCU isodecoders, we investigated whether its genomic context drives increased expression in neurons. The human *tRNA-Arg-TCT-4-1* locus is >2.25 Mbp from other tRNA genes but it is 30 kbp from the TSS of *CADM3*, whose expression is particularly high in neuronal cells in the brain and eye[61] (Human Protein Atlas, https://www.proteinatlas.org/). The genomic co-localization of *CADM3* and *tRNA-Arg-TCT-4-1* is conserved across vertebrates and the levels of *CADM3* mRNA mirrored the *tRNA-Arg-TCT-4-1* expression pattern we

observed during hiPSC differentiation, with a strong upregulation specifically in neurons (Fig. 5i and Extended Data Fig. 1h).

As very few tRNA genes in mice and humans are in locations with conserved synteny like *tRNA-Arg-TCT-4-1* (refs. 6,15,62), we thus considered that *tRNA-Arg-TCT-4-1* could overlap with a distal *cis*-regulatory element of *CADM3*. Comparison of the *tRNA-Arg-TCT-4-1* loci in mice and humans revealed a striking conservation not only of the tRNA gene body but also of a 140-bp region upstream (99% sequence identity; Extended Data Fig. 7a). Inspection of the GeneHancer database revealed that a neuron-specific in vivo-transcribed enhancer overlapping human *tRNA-Arg-TCT-4-1* has been predicted based on cap analysis of gene expression (CAGE) data from the FANTOM5 panel of samples, with *CADM3* as one of its potential targets[63,64] (Fig. 5j). In accordance with enhancer-based regulation, the *CADM3* mRNA levels were very low in hiPSC and NPC despite a high H3K4me3 ChIP signal at the TSS of the gene (Fig. 5j), which could be due to Pol II pausing. Pol II has indeed been

found in a paused state at the *Cadm3* promoter in NPC from the developing mouse cortex, with pausing relieved in their daughter neurons[65]. Neuron-specific *CADM3* enhancer activation could thus potentiate Pol III transcription of the overlapping *tRNA-Arg-TCT-4-1* by establishing a permissive chromatin state, which would account for the exceptionally high levels of tRNA-Arg-UCU-4 in the central nervous system[34]. Overall, 55 human tRNA genes overlap with predicted enhancers[64], although only half of these enhancers (27) are transcribed based on FANTOM5 CAGE (Supplementary Table 3). Among other tRNA genes overlapping transcribed enhancers, *tRNA-Lys-TTT-3-1* and *tRNA-Lys-TTT-3-2* may also be co-regulated with enhancer targets in NPC and neurons (Extended Data Fig. 7b) but this seems to be a rare regulatory mechanism based on our dataset.

### Selective tRNA gene repression is not driven by RPC7α loss

We investigated whether the selective repression of a tRNA gene subset following differentiation is linked to changes in Pol III composition. The human Pol III complex comprises 17 subunits[66], one of which (RPC7) has two isoforms (RPC7α and RPC7β) encoded by two gene paralogues (*POLR3G* and *POLR3GL*). High RPC7α levels are a hallmark of embryonic stem cells and cancer; in healthy differentiated cells, RPC7α is largely replaced by RPC7β[67–70]. Accordingly, the levels of *POLR3G* mRNA and RPC7α protein were strongly decreased in NPC and nearly undetectable in neurons and CM (Extended Data Fig. 8a,b). The switch from RPC7α to RPC7β in Pol III thus coincides temporally with selective tRNA repression (Fig. 4d). We identified 294 consensus tRNA peaks in RPC7α ChIP–Seq from hiPSC, 292 of which were shared with RPC1 consensus peaks (Extended Data Fig. 8c). In contrast to 200 of 205 housekeeping tRNA genes (98%), only 93 of 194 repressed tRNA loci (48%) had significant RPC7α peaks (Extended Data Fig. 8d,e), indicating that RPC7α-containing Pol III is not preferentially enriched at these loci. We also found no significant changes in the RPC1 ChIP signal at tRNA genes in hiPSC depleted for RPC7α by inducible CRISPR interference[71] (CRISPRi; Extended Data Fig. 8f,g), indicating that selective tRNA gene repression following differentiation does not result from RPC7α loss.

### Selective tRNA repression correlates with MAF1 activation

We next focused on the Pol III repressor MAF1, which is kept inactive through phosphorylation by mTORC1 (refs. 72,73). Following a decrease in mTORC1 signalling triggered by low nutrient availability, MAF1 becomes dephosphorylated and inhibits Pol III (ref. 74). The gel migration pattern of MAF1 suggested that it is mostly phosphorylated in hiPSC (Fig. 6a), consistent with the requirement for high mTORC1 activity for pluripotency maintenance[75]. mTORC1 activity (measured by S6K1 and 4E-BP1 phosphorylation) was strongly diminished in differentiated cells, consistent with previous studies of human embryonic stem cell differentiation[76] and mouse neurogenesis[77]. In agreement with this, MAF1 from differentiated cells migrated faster, which is indicative of phosphorylation loss. Interestingly, a small fraction of MAF1 remained partially phosphorylated when hiPSC were treated with the mTORC1 inhibitor rapamycin and MAF1 from NPC exhibited a similar pattern (Fig. 6b), indicating that phosphorylation at one or more sites in MAF1 (S60, S68 and S75)[72] may be less sensitive to this drug. Diminished mTORC1 activity in differentiated cells thus activates MAF1 by altering its phosphorylation status.

To experimentally test whether MAF1 activation mediates selective tRNA gene repression, we used inducible CRISPRi to perform *MAF1* knockdown in hiPSC and NPC as well as before NPC derivation from hiPSC. MAF1 depletion did not alter the levels of RPC1 (Fig. 6c) and only modestly increased Pol III occupancy at 39 tRNA genes in hiPSC. By contrast, more than 100 loci had significantly higher RPC1 ChIP signal strength in NPC derived in the absence of MAF1 ($n = 109$) or depleted of MAF1 after derivation ($n = 110$), with effect sizes that were primarily >fourfold, and up to approximately 30-fold, higher (Fig. 6d). Remarkably, nearly all of these genes belong to the set of

tRNAs that are repressed following differentiation (Fig. 6e). None of the inactive tRNA genes gained significantly more RPC1 ChIP signal in MAF1-depleted NPC and only eight did so in hiPSC (Fig. 6e). The RPC1 ChIP signal strength at housekeeping tRNA genes also remained largely unaffected by MAF1 depletion. These data indicate that cell type-specific human tRNA repertoires are established in a MAF1- and mTORC1-dependent manner (Fig. 6f).

## Discussion

Despite their crucial importance for faithful and efficient mRNA decoding, the composition of tRNA pools in human cells and their regulation have remained poorly defined due to technical limitations. Understanding this regulation is critical for identifying the molecular triggers of human diseases caused by tRNA dysregulation[3,78] as well as for the design of effective mRNA- and tRNA-based therapeutics[79,80]. By applying orthogonal methods in hiPSC-based models, we show that despite extensive remodelling of tRNA repertoires, the levels of mature tRNAs with specific anticodons are maintained largely stable across diverse human cell types. This is mediated by constitutively high transcription of one-third of the predicted human tRNA genes, which we define here as housekeeping. These genes have distinct intragenic promoters and 5′ flanking sequences, and their products comprise the most abundant mature transcripts in each tRNA anticodon family. Housekeeping tRNA genes are largely resistant to MAF1-mediated Pol III repression, which we identify as the mechanism for silencing low-occupancy tRNA loci on differentiation. We propose that the maintenance of stable tRNA anticodon pools and global codon usage across cell types ensures consistent decoding rates throughout development, independently of cell identity.

By combining Pol III ChIP–Seq with high-resolution tRNA quantification in homogeneous populations of distinct isogenic and untransformed human cell types, we found that differences in Pol III occupancy explain nearly all of the variation in mature tRNA levels ($r^2 = 0.9$). This extraordinary concordance between two completely orthogonal workflows further underscores the quantitative nature of tRNA abundance measurements by mim-tRNAseq[21,22]. Whereas Pol III ChIP–Seq requires highly specific antibodies that are unavailable for most organisms, profiling mature tRNA repertoires with mim-tRNAseq is much more broadly applicable and we anticipate that it will help uncover other fundamental aspects of tRNA regulation.

The distinct A and B boxes and 5′ flanking sequence motifs of housekeeping tRNA genes may favour Pol III recruitment or facilitate its recycling at these loci, enabling their escape from MAF1-mediated repression during differentiation. A similar mechanism could account for the protection of some highly transcribed tRNA genes from stress-induced MAF1 inhibition in yeast[81], mice[82] and human fibroblasts[83]. The broader tRNA repertoires we found transcribed in cells with high mTORC1 activity, which is a hallmark of pluripotency but also of many cancers[29], result in tRNA pools with a more diverse isodecoder composition. However, tRNA isodecoder diversity has surprisingly minor effects on decoding speed. We instead found that the relative abundance of tRNA anticodon families, which remains largely unchanged across cell types, determines translation elongation rates at different codons. In physiological contexts, a stable tRNA anticodon supply during development—maintained by housekeeping tRNA gene transcription—would minimize the potential for ribosome errors and protein misfolding that could result from decoding rate fluctuations[84,85].

Why are active tRNA gene sets restricted during differentiation? Given that tRNAs are highly abundant, their synthesis is energetically costly and restriction of Pol III to housekeeping tRNA genes via MAF1 may help maintain tRNA anticodon pools in differentiated cells while conserving resources. In line with this, *Maf1*[−/−] mice are viable but have a lean phenotype and increased energy expenditure[86]. MAF1 plays a role in mouse adipogenesis[87] and osteoblast differentiation[88], but whether

this is through tRNA repertoire reprogramming is difficult to dissect, given that the protein also inhibits Pol III-mediated transcription of 5S ribosomal RNA[5]. This coupling of rRNA and tRNA biogenesis may also serve to maintain the overall stoichiometry between ribosomes and tRNA molecules in cell types with distinct global translation demands.

Despite the strong correlation between H3K4me3 and RPC1 ChIP signals at tRNA genes in our datasets and previous studies[46], we found no clear association of tRNA gene activity with Pol II transcription of nearby coding genes. However, in very rare cases—such as we propose for *tRNA-Arg-TCT-4-1*—an overlap with an enhancer element may boost the expression of an individual tRNA gene in specific cell contexts. Long-range regulatory DNA interactions, rather than linear distance to Pol II genes, could thus modulate the expression of specific tRNA genes in defined cell types. We found some evidence for a similar mode of regulation for *tRNA-Lys-TTT-3-1* and *tRNA-Lys-TTT-3-2* in NPC and neurons, and it remains possible that other tRNA loci we found to overlap with predicted enhancers may be differentially expressed in cellular contexts where these enhancers are active.

## Online content

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

## Methods

### Cell culture and hiPSC differentiation

HEK293T/17 (American Type Culture Collection, CRL-11268) and Lenti-X 293T (Takara Bio, 632180) cells were cultured in DMEM high-glucose medium supplemented with 10% fetal calf serum (FCS) at 37 °C with 5% $CO_2$. The HPSI0214i-kucg_2 and HPSI0214i-wibj_2 reference hiPSC lines[23] were obtained from the European Collection of Authenticated Cell Cultures (catalogue number 77659901) and cultured in mTeSR Plus on Geltrex-coated plates at 37 °C with 5% $CO_2$. Neurons and NPC were derived from HPSI0214i-kucg_2 cells using small molecules as previously described[32,33]. For NPC differentiation, hiPSC were cultured to 90% confluency and cut by scratching a chequered pattern into the dish with a cannula, followed by incubation with collagenase IV for 10–15 min at 37 °C. Cell clusters were carefully scratched off the plate, transferred to a 15 ml tube containing Neurobasal (N2B27) medium (Gibco, 21103049)−DMEM/F12 (Gibco, 21331020) 50:50, 0.5×N2 (Thermo Fisher Scientific, 17502048), 0.5×B27 (Thermo Fisher Scientific, 12587010), 2 mM GlutaMAX (Gibco, 35050061)−and pelleted by gravity. Cell clusters were washed once with N2B27 medium and transferred into NPC-induction medium (N2B27 with 200 μM ascorbic acid (Sigma-Aldrich, A4403), 3 μM CHIR99021 (Axon Medchem, Axon1386), 0.5 μM purmorphamine (Santa Cruz Biotechnology, sc-202785A), 150 nM dorsomorphin (Absource, S7306) and 10 μM SB431542 (Biomol, Cay12031)) with 5 μM ROCK inhibitor (Y-27632; Stemcell Technologies, 72305) in a sterile dish without coating, to allow embryoid body formation, and incubated at 37 °C with 5% $CO_2$. The medium was exchanged every 2 d with NPC-induction medium without Y-27632. On day six, the embryoid bodies were dissociated into single cells by pipetting and plated into a Geltrex-coated well in NPC expansion medium (N2B27 with 200 μM ascorbic acid, 3 μM CHIR99021 and 0.5 μM purmorphamine). The medium was changed every other day. To remove non-NPC cells, a sequential digest was performed during the first passages using Accutase. Standard passaging was performed as for hiPSC single-cell passaging every 5 d at a ratio of 1:10.

For the differentiation of NPC to neurons[33], cells were singularized with Accutase and $1 \times 10^6$ cells were seeded into a six-well plate containing patterning medium (N2B27 with 200 μM ascorbic acid, 1 μM retinoic acid (Sigma-Aldrich, R2625), 0.5 μM purmorphamine and 10 ng ml⁻¹ of both GDNF and BDNF (Peprotech, 450-10 and 450-02)). The cells were cultured for 6 d with a medium change every other day. On day six, the medium was changed to maturation medium (MM; N2B27 with 200 μM ascorbic acid, 100 μM dbcAMP (Sigma-Aldrich, D0627), 5 ng ml⁻¹ GDNF and BDNF, and 1 ng ml⁻¹ TGF-β3 (Peprotech, AF-100-36E)) with 5 ng μl⁻¹ Activin A (Life Technologies, PHG9014). After 2 d, the medium was exchanged with MM without Activin A. The cells were maintained in plates for another 10 d, with medium exchanges every 2–3 d. On day 16, the cells were detached with Accutase, resuspended in MM, pelleted by centrifugation for 5 min at 200g and transferred to a new plate. CompE (0.1 μM; Merck, 565790) was added to the medium on day 19 to enhance neuronal maturation and the cells were harvested on day 21.

Cardiomyocytes were derived from HPSI0214i_kucg-2 hiPSC as previously described[30,31], with some modifications. Accutase was used to dissociate hiPSC into single cells, which were then seeded in day 0 differentiation medium (KO-DMEM (Gibco, 10829-018); 2 mM ʟ-glutamine (Gibco, 25030-024); insulin, transferrin and selenious acid (5 μg ml⁻¹ each; ITS; Corning, 354351); 10 ng ml⁻¹ FGF2 (Peprotech, 100-18B-250); 1 μM CHIR 9920 (Axon, 1386); 1 ng ml⁻¹ BMP-4 (R&D, 314-BP-010); 5 ng ml⁻¹ Activin A (Life Technologies, PHG9014) and 10 μM Y-27632 on Matrigel-coated plates. The medium was changed to transferrin/selenium medium (KO-DMEM, 2 mM ʟ-glutamine, 5.5 μg ml⁻¹ human transferrin (Sigma-Aldrich, TS8158-100mg), 6.7 ng ml⁻¹ sodium selenite (Sigma-Aldrich, 214485) and 250 μM ascorbic acid (Sigma-Aldrich, A4403-100mg)) after 1 d. On days 2 and 3, the medium was replaced with transferrin/selenium medium supplemented with 0.2 μM WNT-inhibitor C59 (Tocris, 5148). The medium was exchanged daily

until day 9. To enrich for CM, the cells were then starved of glucose for 1 d in transferrin/selenium medium minus glucose medium (DMEM without glucose (Gibco, A13320-01), 2 mM ʟ-glutamine, 5.5 μg ml⁻¹ human transferrin, 6.7 ng μl⁻¹ sodium selenite, 250 μM ascorbic acid and 4 mM lactic acid (Sigma L4263-100ml)[31]. On day 10, the cells were trypsinized with Accutase and plated in CM-MM medium (KO-DMEM, 2% FCS (Gibco, 16000-044), 2 mM ʟ-glutamine and 10 μM Y-27632 on Matrigel-coated wells. The following day the medium was replaced with fresh CM-MM without Y-27632, which was exchanged every 2 d until the cells were harvested on day 15.

HEK293T/17 and Lenti-X 293T cells were cultured (at 37 °C with 5% $CO_2$) in DMEM high-glucose medium supplemented with 10% FCS and passaged using 0.25% trypsin in EDTA every other day at a ratio of 1:10–1:20.

### Generation of an inducible CRISPRi hiPSC line

HPSI0214i_kucg-2 cells were engineered to express KRAB-dCas9 from a doxycycline-inducible promoter at the *AAVS1* locus[71] using pAAVS1-PDi-CRISPRn (a gift from B. Conklin; Addgene, plasmid 73500; http://n2t.net/addgene:73500; RRID: Addgene_73500). The cultures were selected with 100 μg ml⁻¹ G418 until stable colonies originating from single cells formed. The colonies were picked and screened for heterozygous insertion by PCR using two primers flanking the *AAVS1* locus (5′-CGAGAGCTCAGCTAGTCTTC-3′ and 5′-CTCTCCCTCCCAGGATCC-3′) and an additional primer binding the insert (5′-GTTCATTCAGGGCACCGGAC-3′). KRAB-dCas9 expression in positive clones was assessed by flow cytometry and immunoblotting after the addition of 2 μM doxycycline. Genome integrity was verified by G-band analysis of expanded clones.

### CRISPR−Cas9 genome editing

CRISPR RNA and single-stranded oligodeoxynucleotide templates were obtained from IDT. Guide RNAs were assembled by annealing the CRISPR RNA (5′-UGUGGGCCAAGGCUAGGGAGGUUUUAGAGCUAUGCU-3′ for the Pro-TGG-2 gene body edit and 5′-UUGCUCAGCAGAUGGCU CGUGUUUUAGAGCUAUGCU-3′ for the Pro-TGG-2 upstream region edit) with *trans*-activating CRISPR RNA (IDT) in equimolar ratios at 95 °C for 5 min. Ribonucleoprotein complex was assembled by mixing 100 pmol guide RNA with 50 pmol Alt-R HiFi Cas9 (IDT) and incubated at room temperature for 20 min. HPSI0214i_kucg-2 cells were dissociated into single cells using Accutase and Nucleofected with ribonucleoprotein complex and HDR donor oligonucleotide in P3 solution (Lonza) using the CA137 programme in a Nucleocuvette strip. The cells were re-plated in mTeSR Plus supplemented with 1:10 CloneR (Stemcell Technologies, 05888). The medium was exchanged with mTeSR Plus every 2 d. Colonies were picked and expanded, and homozygous edited clones were identified by PCR amplification of genomic DNA and Sanger sequencing.

### CRISPRi sgRNA design

Single guide RNAs (sgRNAs) were designed to target the TSS of genes in the GENCODE v19 annotation using an adapted workflow of the CRISPRiaDesign protocol (https://github.com/mhorlbeck/CRISPRia-Design). To incorporate information about single nucleotide polymorphisms (SNPs) in the HPSI0214i_kucg-2 genome, we used GATK haplotype calls for the cell line (ftp://ftp.sra.ebi.ac.uk/vol1/ERZ447/ERZ447992/) and extracted variant sites only using gvcftools extract_variants v0.17.0. The resulting genomic variant call format (VCF) file was indexed and genotypes were called using GATK GenotypeGVCFs v4.1.0.0. From this, only SNPs were retained so as to preserve genomic context and position information between GRCh37 and our custom genome. This was achieved using GATK SelectVariants v4.1.0.0 with the -select-type SNP parameter. We then replaced nucleotides in the reference GRCh37 genome with the called genotype SNPs by generating a sequence dictionary from the reference genome using Picard

CreateSequenceDictionary v2.17.10 and supplying the SNP VCF to GATK FastaAlternateReferenceMaker v4.1.0.0. To train an elastic net linear regression model for sgRNA activity predictions, this custom genome was used in combination with other supplied training data from the CRISPRia pipeline, including sgRNA activity scores and TSS predictions (https://github.com/mhorlbeck/CRISPRiaDesign/tree/master/data_files) and our own ATAC–Seq data from HPSI0214i-kucg_2 cells as a proxy for chromatin accessibility. After activity score prediction, off-targets were predicted per sgRNA as described[89].

## CRISPRi knockdown
*POLR3G*- and *MAF1*-targeting sgRNAs (5′-GGACTCGCCGGAGC GCTCTG-3′ and 5′-GGTGCCGGCCGGCAAGGAAA-3′) were cloned in pU6-sgRNA EF1α-Puro-T2A-GFP by Gibson assembly. This plasmid was constructed by replacing BFP with GFP in pU6-sgRNA EF1α-Puro-T2A-BFP (a gift from J. Weissman; Addgene plasmid 60955; http://n2t.net/addgene:60955; RRID:Addgene_60955). Lentivirus stocks were produced by co-transfection of the resulting plasmid with packaging plasmids (gifts from D. Trono; pMDLg/pRRE, Addgene, plasmid 12251, http://n2t.net/addgene:12251, RRID:Addgene_12251; pRSV-Rev, Addgene, plasmid 12253, http://n2t.net/addgene:12253, RRID:Addgene_12253; and pMD2.G, Addgene, plasmid 12259, http://n2t.net/addgene:12259, RRID:Addgene_12259) into Lenti-X 293T cells with TransIT-Lenti transfection reagent (Mirus, MIR6603) following the manufacturer's instructions. Viral supernatant was harvested 48–72 h after transfection, filtered through a 0.45 μm polyvinylidene fluoride syringe filter and then precipitated overnight with Lentivirus precipitation solution (Alstembio, VC125) at 4 °C. Virus stocks were concentrated tenfold in cold PBS, aliquoted and stored at −80 °C.

Lentiviral transduction of hiPSC was performed by adding thawed lentivirus stock mixed with fresh medium to plates, followed by an incubation of 10 min at 37 °C with 5% $CO_2$ and the addition of trypsinized cells. The cells were incubated with lentivirus for 2 d before splitting and selection with 2.5 μg ml$^{-1}$ puromycin for 2–3 d. NPC were transduced as per the protocol for hiPSC, except that the incubation with lentivirus was reduced to 1 d and performed in the absence of doxycycline. The cells were then selected with 2.5 μg ml$^{-1}$ puromycin in the presence or absence of doxycycline until >80% of the cells were GFP-positive.

## RNA isolation
Cells were lysed in lithium dodecyl sulfate (LiDS)/LET buffer (5% LiDS in 20 mM Tris, 100 mM LiCl, 2 mM EDTA, 5 mM dithiothreitol (DTT) pH 7.4 and 100 μg ml$^{-1}$ proteinase K). The lysates were incubated at 60 °C for 10 min, pushed ten times through a 1 ml syringe with a 26 G needle and mixed by vortexing. Two volumes of cold acid phenol (pH 4.3), 1/10 volume 1-bromo-3-chloropropane and 50 μg glycogen (Thermo Fisher Scientific, AM9510) were added. The samples were mixed vigorously by vortexing, followed by centrifugation at 10,000g and 4 °C. The aqueous phase was transferred to a new tube and the phenol and 1-bromo-3-chloropropane extraction was repeated. RNA was then precipitated from the aqueous phase by the addition of three volumes of 100% ethanol and incubation at −20 °C for 30 min. The pellets were washed with 80% ethanol, air-dried and resuspended in RNase-free water. The RNA concentration was measured using a Nanodrop system and the samples were stored at −80 °C.

## Northern blotting
For each sample, 0.5 μg total RNA was separated on denaturing gels (10% polyacrylamide in 7 M urea and 1×TBE). The RNA was transferred to Immobilon NY+ membranes (Millipore) in 1×TBE at 4 mA cm$^{-2}$ for 40 min using a TransBlot Turbo apparatus (Bio-Rad) and crosslinked at 0.04 J in a Stratalinker ultraviolet light crosslinker. The membranes were incubated at 80 °C for 1 h and pre-hybridized at 55 °C for 4 h in hybridization buffer (20 mM $Na_2HPO_4$ pH 7.2, 5×SSC, 7% SDS, 2×Denhardt's solution and 40 μg ml$^{-1}$ sheared

salmon sperm DNA). This was followed by overnight hybridization with 10 pmol 5′-end $^{32}$P-labelled probes (tRNA-Gly-CCC-2, 5′-CGGGTCGCAAGAATGGGAATCTTGCATGATAC-3′; tRNA-Arg-UCU-4, 5′-CGGAACCTCTGGATTAGAAGTCCAGCGCGCTCGTCC-3′ and tRNA-Asn-GUU-1, 5′-CGTCCCTGGGTGGGATCGAACC-3′) in hybridization buffer. Finally, the membranes were washed three times in 25 mM $Na_2HPO_4$ pH 7.5, 3×SSC, 5% SDS and 10×Denhardt's solution, washed once in 1×SSC and 10% SDS, and exposed to PhosphorImager screens scanned on a Typhoon FLA 9000 (GE Healthcare). The band intensity was quantified using ImageJ.

## Immunoblotting
Cells were lysed in RIPA buffer (20 mM Tris pH 7.5, 150 mM NaCl, 1% NP-40, 0.5% sodium deoxycholate and 0.1% SDS) supplemented with 10 μg ml$^{-1}$ aprotinin, 20 μM leupeptin, 2.5 μM pepstatin A, 0.5 mM 4-(2-aminoethyl)benzenesulfonyl fluoride hydrochloride and 1×phosphatase inhibitor cocktail (Cell Signaling Technologies, 5870). The protein concentration was determined using a Pierce BCA protein assay kit (Thermo Fisher Scientific, 23225). For each sample, 20 μg total protein was resolved by SDS–PAGE on 10% gels supplemented with 0.5% 2,2,2-trichloroethanol (Sigma, T54801) or on pre-cast 4–12% bis-Tris polyacrylamide gels (Life Technologies) in Bolt MES SDS running buffer (Invitrogen, B0002). Total protein stained with 2,2,2-trichloroethanol was imaged by ultraviolet light illumination on a ChemiDoc system (Bio-Rad). The proteins were then transferred to a nitrocellulose membrane (Amersham, 10600015). For visualizing total protein in pre-cast gels, the membranes were stained with Ponceau S solution (0.5% Ponceau S and 1% acetic acid) for 3 min at room temperature with gentle shaking and imaged on a ChemiDoc system (Bio-Rad) after rinsing with distilled water. The membranes were blocked in 5% milk in PBS-0.1% Tween-20 for 1 h, followed by overnight incubation at 4 °C with primary antibodies. The primary antibodies used for immunoblotting were anti-phospho-p70 S6 kinase (1:1,000; Cell Signaling Technologies, 9206S), rabbit anti-phospho-4E-BP1 (1:1,000; Cell Signaling Technologies, 2855T), mouse anti-Pol III RPC32/RPC7α (1:1,000; Santa Cruz Biotechnology, sc-21754), mouse anti-MAF1 (1:1,000; Santa Cruz Biotechnology, sc-515614 X), mouse anti-POLR3B/RPC2 (1:1,000; Santa Cruz Biotechnology, sc-515362), rabbit anti-POLR3A/RPC1 (1:1,000; Cell Signaling Technologies, 12825) and rabbit anti-vinculin (1:1,000; Cell Signaling Technologies, 13901). The membranes were then incubated with horseradish peroxidase (HRP)-labelled secondary antibodies (1:4,000; anti-rabbit IgG–HRP or anti-mouse IgG–HRP; Dianova, 111-035-003 and 115-035-003, respectively) at room temperature for 1 h. Proteins were visualized by chemiluminescence using SuperSignal west pico PLUS (Thermo Fisher Scientific, 34577) and imaged on an iBright system (Thermo Fisher Scientific).

For S6K1 and 4E-BP1 immunoblotting, the membranes were first probed with phospho-specific antibodies (1:1,000; anti-phospho-4E-BP1 and anti-phospho-p70 S6 kinase (T389); Cell Signaling Technologies, 2855T and 9206S, respectively). The membranes were stripped by two rounds of incubation in 25 ml Restore western blot stripping buffer (Thermo Fisher Scientific, 21059) for 15 min at room temperature with gentle shaking. This was followed by another round of blocking and the membranes were re-probed with anti-4E-BP1 (1:2,000; Cell Signaling Technologies, 9644) and anti-p70 S6 kinase (1:2,000; Cell Signaling Technologies, 2708T).

For Phos-tag immunoblotting, 20 μg total protein from each sample was mixed with 4×Laemmli sample buffer (Bio-Rad, 161-0747) supplemented with 25 mM DTT and boiled for 10 min at 95 °C. The denatured samples were run on Phos-tag gels (8% acrylamide in bis solution 29:1; 0.375 M Tris–HCl, pH 8.8, 20 μM Phos-tag (Wako, AAL-107) and 40 μM $MnCl_2$) in 1×Tris/glycine/SDS running buffer (Bio-Rad, 1610732). The gels were washed twice with gentle shaking in transfer buffer (25 mM Tris, 192 mM glycine and 10% methanol) containing 1 mM EDTA (10 min each wash), followed by two washes (10 min each

wash) in transfer buffer without EDTA. The proteins were transferred to polyvinylidene fluoride membranes (Amersham, 10600021) overnight in 25 mM Tris, 192 mM glycine and 10% methanol at 35 V and room temperature. The membrane was blocked in 5% milk in PBS-0.1% Tween-20 for 1 h, followed by overnight incubation with mouse anti-MAF1 (1:1,000; Santa Cruz Biotechnology, sc-515614 X) at 4 °C and anti-mouse IgG–HRP (1:4,000; Dianova, 115-035-003) at room temperature for 1 h. The proteins were visualized by chemiluminescence using SuperSignal West Femto Maximum Sensitivity Substrate (Thermo Fisher Scientific, 34094) and imaged on an iBright system.

#### Immunostaining
Cells were cultured on glass-bottomed dishes (ibidi, 80827). For staining, the cells were washed with PBS and fixed in 3.7% formaldehyde for 10 min at room temperature. The formaldehyde was exchanged stepwise with PBS-0.02% Tween-20, followed by three complete washes with PBS. NPC and hiPSC were permeabilized with 0.5% Triton X-100 in PBS0.02% Tween-20 for 10 min and blocked for 1 h in blocking solution (3% BSA and 0.1% Triton X-100 in PBS). The cells were incubated overnight at 4 °C with the primary antibody diluted in blocking solution (POU5F1 C-10, Santa Cruz Biotechnology, sc-5279, 1:400; SOX2 E-4, Santa Cruz Biotechnology, sc-365823, 1:200; NANOG P1-2D8, deposited to the Developmental Studies Hybridoma Bank (DSHB) by Common Fund Protein Capture Reagents Program (DSHB Hybridoma Product PCRP-NANOGP1-2D8), 1:200; PAX6, Abcam ab5790, 1:200; Nestin, R&D Systems, MAB1259, 1:200). After three washes with PBS-0.02% Tween-20, the cells were incubated with secondary antibody diluted in blocking solution for 1 h at room temperature (goat anti-mouse–Alexa Fluor 488, 1:2,000; goat anti-rabbit–Alexa Fluor 488, 1:2,000 or goat anti-mouse–Alexa Fluor 633, 1:500; Thermo Fisher Scientific, A-11001, A-11034 and A-21052, respectively). The cells were washed another three times in PBS-0.02% Tween-20 before imaging, and 4,6-diamidino-2-phenylindole (DAPI) was added during the second wash step (1:1,000). Neurons were permeabilized for 10 min in PBS-0.7% Tween-20 and blocked for 1 h in neuron blocking solution (1% BSA, 0.1% Triton X-100 and 10% FCS in PBS). The cells were washed once in 0.1% BSA in PBS and incubated overnight with the primary antibody diluted in PBS containing 1% BSA at 4 °C (anti-MAP2, 1:1,000 and anti-CHAT, 1:200; Abcam, ab92434 and ab6168, respectively). After three washes in PBS containing 0.1% BSA, the cells were incubated with the secondary antibody diluted in PBS containing 1% BSA for 1 h at room temperature (goat anti-rabbit A633, 1:500 and goat anti-chicken A488, 1:2,000; Thermo Fisher Scientific, A-21070 and A-11039, respectively), followed by another three washes with 1% BSA in PBS-0.05% Tween-20, with DAPI added during the second wash step (1:1,000). Cardiomyocytes were blocked and permeabilized in blocking solution (3% BSA and 0.1% Triton X-100 in PBS) for 1 h at room temperature. After three washes with PBS-0.1% Tween-20, the cells were incubated overnight at 4 °C with primary antibody (anti-α-actinin-2, Sigma-Aldrich, A7811, 1:800 and anti-cardiac troponin T, CT3, deposited to the DSHB by Lin, J. J. -C., 1:5) diluted in staining solution (1% BSA and 0.1% Tween in PBS). After three washes with PBS0.1% Tween-20, the cells were incubated with the secondary antibody (goat anti-mouse–Alexa Fluor 488, Thermo Fisher Scientific, A-11001, 1:2,000) and DAPI (1:1,000) diluted in staining solution for 1 h at room temperature in the dark. The cells were washed three times in PBS-0.1% Tween-20 and imaged in PBS.

#### RNA-Seq library construction
A total of 250 ng of the same total RNA used for mim-tRNAseq library preparation was used for mRNA-Seq library construction with a Zymo-Seq RiboFree total RNA library kit (Zymo Research, R3000). The libraries were quantified using a Qubit dsDNA HS assay, fragment size was determined on an Agilent TapeStation and the libraries were sequenced for 120 cycles on an Illumina NovaSeq platform, generating >21 × 10^6 reads per library.

#### tRNA-Seq library construction
The tRNASeq libraries were prepared using the mim-tRNAseq workflow[21,22]. Briefly, total RNA from two biological replicates for each cell line was mixed with synthetic *Escherichia coli* tRNA-Lys-UUU-CCA and *E. coli* tRNA-Lys-UUU-CC at a 3:1 ratio, followed by dephosphorylation with T4 PNK (NEB, M0201S) and ethanol precipitation. The RNA samples were resolved on denaturing 10% polyacrylamide, 7 M urea and 1×TBE gels. RNA of 60–100 nt in length was recovered by gel excision and elution from gel slices, followed by ethanol precipitation. The gel-purified tRNA was then ligated to pre-adenylated, barcoded 3′-adaptors[22] in 1×T4 RNA ligase buffer, 25% PEG-8000, 20 U Superase In (Thermo Fisher Scientific, AM2696) and 1 μl T4 RNA ligase 2, truncated KQ (NEB, M0373S). The mix was incubated for 3 h at 25 °C and the ligation products were purified by size selection on a 10% polyacrylamide, 7 M urea and 1×TBE gel. Adaptor-ligated tRNA (100 ng) was annealed with 1 μl of 1.25 μM RT primer (5′-pRNAGATCGGAAGAGCGTCGTGTAGGGAAAGAG/iSp18/GTGACTGGAGTTCAGACGTGTGCTC-3′, where iSp18 is a 18-atom hexa-ethyleneglycol spacer) at 82 °C for 2 min, followed by incubation at 25 °C for 5 min. Reverse transcription was performed with 500 nM TGIRT (InGex, TGIRT50) in 50 mM Tris–HCl pH 8.3, 75 mM KCl, 3 mM MgCl$_2$, 5 mM DTT (from a freshly prepared 100 mM stock), 1.25 mM dNTPs and 20 U Superase In at 42 °C for 16 h. After reverse transcription, NaOH was added to a final concentration of 0.1 M and the RNA was hydrolysed by incubating the samples for 5 min at 90 °C. Complementary DNA products were separated from unextended primer on a 10% polyacrylamide, 7 M urea and 1×TBE gel. Regions corresponding to cDNAs that were >10 nt longer than the RT primer were excised after SYBR Gold staining. Gel-purified and ethanol-precipitated cDNA was incubated for 3 h at 60 °C with CircLigase ssDNA ligase (Lucigen) in 1×reaction buffer supplemented with 1 mM ATP, 50 mM MgCl$_2$ and 1 M betaine. Following enzyme inactivation for 10 min at 80 °C, one-fifth of the circularized cDNA was used directly for library construction PCR with a common forward (5′-AATGATACGGCGACCACCGAGATCTACACTCTTTCCCTACACGACGCT*C-3′) and unique indexed reverse primers (5′-CAAGCAGAAGACGGCATACGAGATNNNNNNGTGACTGGAGTTCAGACGTGT*G-3′; NNNNNN, the reverse complement of an Illumina index sequence; asterisk, phosphorothioate bond) with KAPA HiFi DNA polymerase (Roche) in 1×GC buffer with initial denaturation at 95 °C for 3 min, followed by five cycles of 98 °C for 20 s, 62 °C for 30 s and 72 °C for 30 s at a ramp rate of 3 °C s$^{-1}$. The PCR products were purified using a DNA Clean and Concentrator 5 kit (Zymo Research), quantified with a Qubit dsDNA HS kit (Thermo Fisher Scientific, Q32851) and sequenced for 150 cycles on an Illumina NextSeq 500 platform, generating >2.5 × 10^6 reads per library.

#### ChIP–Seq library construction
Cells cultured in six-well plates were fixed with 0.8% methanol-free formaldehyde (Thermo Fisher Scientific, 28906) in DMEM medium for 10 min at room temperature with gentle shaking, followed by quenching with 0.125 M glycine for 5 min. All buffers in the subsequent steps of the protocol were supplemented with cOmplete EDTA-free protease inhibitor cocktail (Roche, 1187358000). The cells were washed twice with ice-cold PBS and resuspended in Farnham buffer (5 mM PIPES pH 8.0, 85 mM KCl and 0.5% IGEPAL-CA 630), followed by snap freezing in liquid nitrogen.

Chromatin was isolated and sheared following the NEXSON protocol[90]. Frozen cell pellets were thawed on ice and sonicated for 2 min in 1 ml tubes (Covaris, 520130) on a Covaris S220 ultrasonicator at peak power = 75 W, duty factor = 2% and cycles per burst = 200. Isolated nuclei were washed once with Farnham buffer and resuspended in shearing buffer (10 mM Tris–HCl pH 8.0, 0.1% SDS and 1 mM EDTA). Chromatin was sheared on a Covaris S220 system by sonication for 9 min (for BRF1 ChIP) or 18 min (for all other ChIP) in 1 ml tubes at peak power = 140 W, duty factor = 5% and cycles per burst = 200. The sheared chromatin was clarified by centrifugation for 10 min at 16,000*g*. DNA

isolated from 10 µl sheared chromatin was used for size analysis on an Agilent TapeStation system; a DNA fragment-size distribution of 100–800 bp was considered suitable for ChIP. The sheared chromatin was snap-frozen and stored at −80 °C. A 10 µl aliquot was used to determine the DNA concentration using a Qubit dsDNA HS assay. For this, crosslinks were reversed by overnight incubation with 0.2 M NaCl at 65 °C, followed by incubation with 50 µg ml$^{-1}$ RNase A (Thermo Fisher Scientific, EN0531) at 37 °C for 30 min, after which the samples were incubated with 200 µg ml$^{-1}$ proteinase K (Sigma-Aldrich, P2308) at 65 °C for 1 h. The DNA was purified using a DNA ChIP clean and concentrator kit (Zymo Research, D5205) and eluted in 10 µl of 10 mM Tris pH 8.5 with 0.1 mM EDTA.

The sheared chromatin was thawed on ice. For the RPC1 ChIP, 5 µg chromatin was diluted 1:8 with ChIP Dilution buffer (23 mM Tris–HCl pH 8.0, 200 mM NaCl, 2.3 mM EDTA and 1.3% Triton X-100). Magna ChIP protein A + G magnetic beads (Merck, 16-663) were blocked with 5 mg ml$^{-1}$ BSA in PBS for 2 h at room temperature on a rotating platform and resuspended in ChIP dilution buffer. *Drosophila melanogaster* spike-in chromatin (Active Motif, 53083) was added to a final concentration of 0.5% and the chromatin was pre-cleared by incubation with 10 µl blocked magnetic beads for 1 h at 4 °C on a rotating platform. The pre-cleared chromatin was incubated with 5 µg anti-POLR3A/RPC1 (Cell Signaling Technologies, 12825) and 0.2 µg *Drosophila* antibody (Active Motif, 61686) overnight at 4 °C on a rotating platform. For the H3K4me3 and H3K27me3 ChIP, 2 µg pre-cleared chromatin and 5 µl anti-H3K4me3 (Active Motif, 39159) or anti-H3K27me3 (Millipore, 07-449) was used per ChIP and spike-in chromatin was omitted. The samples were then incubated with 60 µl blocked magnetic beads for 2 h at 4 °C on a rotating platform. The beads were washed sequentially with low-salt buffer (0.1% SDS, 1% Triton X-100, 2 mM EDTA pH 8.0, 20 mM Tris–HCl pH 8.0 and 150 mM NaCl), high-salt buffer (0.1% SDS, 1% Triton X-100, 2 mM EDTA pH 8.0, 20 mM Tris–HCl pH 8.0 and 500 mM NaCl), lithium chloride buffer (0.25 M LiCl, 1% IGEPAL-CA 630, 1% sodium deoxycholate, 1 mM EDTA and 10 mM Tris–HCl pH 8.0) and Tris–EDTA buffer (10 mM Tris–HCl and 1 mM EDTA pH 8.0). Each wash was performed twice at 4 °C for 10 min on a rotating platform. The DNA was eluted through two incubations with ChIP elution buffer (1% SDS and 50 mM NaHCO$_3$) for 30 min at room temperature on a rotating platform. The crosslinking was reversed and DNA was purified as for the input chromatin.

For the BRF1 and RPC7α/POLR3G ChIP, 5 µg chromatin was diluted 1:8 with ChIP RIPA buffer (50 mM Tris–HCl pH 8.0, 150 mM NaCl, 2 mM EDTA pH 8.0, 1% NP-40, 0.5% sodium deoxycholate and 0.1% SDS) and pre-cleared as described above. Magnetic beads were blocked with PBS containing BSA as for the RPC1 ChIP but resuspended in ChIP RIPA buffer. The pre-cleared chromatin (5 µg) was incubated overnight with 10 µl anti-BRF1 (Abcam, ab264191) or 20 µl anti-RPC7α/POLR3G (Santa Cruz Biotechnology, sc-21754) at 4 °C with rotation. The samples were then incubated with 60 µl blocked magnetic beads with rotation for 6 h at 4 °C. The beads were washed three times with low-salt buffer and once with high-salt buffer for 10 min at 4 °C with rotation. DNA was eluted from the beads by two sequential 30 min incubations with RIPA elution buffer (1% SDS and 100 mM NaHCO$_3$) at room temperature with rotation.

H3K9me3 ChIP was performed following the Ren laboratory ChIP protocol (http://bioinformatics-renlab.ucsd.edu/RenLabChipProtocolV1.pdf). Dynabeads M-280 sheep anti-rabbit IgG (50 µl; Thermo Fisher Scientific, 11203D) were washed three times with 5 mg ml$^{-1}$ BSA in PBS (BSA/PBS) and resuspended in 100 µl BSA/PBS. Anti-H3K9me3 (5 µl; Cell Signaling Technologies, 13969) was added to 900 µl BSA/PBS and then combined with the magnetic beads. The mixture was incubated overnight on a rotating platform at 4 °C. The beads were then washed three times with 1 ml BSA/PBS and resuspended in 100 µl BSA/PBS. ChIP reactions were set up by taking 5 µg chromatin and adjusting the volume to 1 ml with TE buffer (10 mM Tris–HCl and 1 mM EDTA pH 8.0). A 300 µl volume of STOCK solution (1% Triton

X-100 and 0.1% sodium deoxycholate, prepared in Tris–EDTA buffer) was added to each reaction, followed by mixing with the resuspended 100 µl antibody–beads mixture. The mixture was incubated overnight on a rotating platform at 4 °C. The beads were washed eight times with 1 ml RIPA2 buffer (50 mM HEPES pH 8.0, 1 mM EDTA, 1% NP-40, 0.7% sodium deoxycholate and 0.5 M LiCl), followed by one wash with 1 ml Tris–EDTA. After removing the TE buffer using a magnetic rack, the beads were centrifuged for 1 min at 4,000 r.p.m. and the remaining liquid was removed. The protocol for DNA elution from beads was performed as for the RPC1 ChIP. Crosslinking reversal and DNA clean-up were performed as for input chromatin.

Sequencing libraries from ChIP-eluted DNA samples were prepared using an Ovation ultralow V2 DNA-Seq library preparation kit (Tecan, 0344NB) and SPRIselect beads (Beckman Coulter, B23318) according to the manufacturer's protocol. The library concentration was determined using a Qubit dsDNA HS assay (Thermo Fisher Scientific) and fragment-size distribution was assessed on an Agilent TapeStation system. An Illumina NovaSeq platform was used to perform 110-bp paired-end sequencing, generating >30 × 10$^6$ reads per library.

### ATAC–Seq library construction

ATAC–Seq was performed using an ATAC–Seq kit (Active Motif, 53150) according to the manufacturer's instructions. Briefly, two biological replicates of 50,000 cells from each cell type were tagmented at 37 °C for 60 min. After verifying a nucleosomal banding pattern in the resulting libraries on an Agilent Tapestation, they were quantified using a KAPA library quantification kit (catalogue number, KK4854) and sequenced in a 75-bp paired-end run on an Illumina NextSeq 500 platform, generating 21.5–46 × 10$^6$ reads per library.

### Ribosome profiling library construction

Ribosome footprint libraries were prepared essentially as described previously[40,91] with minor modifications. The cell medium was changed 2 h before harvesting. The cells were quickly washed with ice-cold PBS supplemented with 100 µg ml$^{-1}$ cycloheximide (Sigma-Aldrich, C1988) and snap-frozen. For libraries prepared with cycloheximide in the lysis buffer, plates were thawed on ice and the cells were scraped off the plate in 400 µl polysome lysis buffer (20 mM Tris pH 7.4, 150 mM NaCl, 5 mM MgCl$_2$, 1% Triton X-100, 1 mM DTT, 100 µg ml$^{-1}$ cycloheximide, 25 U ml$^{-1}$ Turbo DNase (Thermo Fisher Scientific, AM2238), 0.1% NP-40, 10 µg ml$^{-1}$ aprotinin, 20 µM leupeptin, 2.5 µM pepstatin A, 0.5 mM 4-(2-aminoethyl)benzenesulfonyl fluoride hydrochloride and 1×phosphatase inhibitor cocktail). The samples were vortexed vigorously, triturated through a 26 G gauge needle and spun down for 7 min at 16,000g and 4 °C. The RNA concentration in the supernatant was measured using a Qubit RNA HS kit. RNA (20 µg) in 200 µl polysome lysis buffer was digested with 50 U RNase I (Thermo Fisher Scientific, AM2295) for 45 min at 2,000 r.p.m. and 22 °C.

For libraries prepared with cycloheximide and tigecycline in the lysis buffer, plates were thawed and cells from a 10 cm dish were lysed in 15 ml polysome lysis buffer supplemented with 0.1% NP-40 and 100 µg ml$^{-1}$ tigecycline (Sigma-Aldrich, PZ0021). After incubation on ice for 5 min, extracts were pre-cleared by centrifugation for 5 min at 3,000g and 4 °C. Ribosomes were pelleted through 3 ml of a sucrose cushion (1 M sucrose, 20 mM Tris pH 8.0, 140 mM KCl, 5 mM MgCl$_2$ and 1 mM DTT) in a Type 70 Ti rotor for 120 min at 50,000 r.p.m. and 4 °C. The ribosome pellets were rinsed once, dissolved in 200 µl drug-free polysome lysis buffer and incubated with 200 (hiPSC) or 300 U (NPC) RNase I for 45 min at 2,000 r.p.m. and 22 °C.

The RNase I digestion was stopped by the addition of 100 U Superase In (Thermo Fisher Scientific, AM2694), and the extracts were loaded on a 0.9 ml sucrose cushion (1 M sucrose in polysome lysis buffer), followed by centrifugation for 75 min at 120,000 r.p.m. and 4 °C in a S120AT2 rotor (Thermo Fisher Scientific). The pellet was dissolved in 400 µl LiDS/LET lysis buffer and RNA was extracted as described

for total RNA isolation. The RNA (3 µg) was loaded on 15% polyacrylamide, 7 M urea and 1×TBE gels. Fragments in the range of 19–32 nt were excised from the gel and crushed with a pestle. The RNA was eluted in 400 µl gel elution buffer (0.3 M sodium acetate pH 4.5, 0.25% SDS, 1 mM EDTA pH 8.0) by heating (65 °C for 10 min), followed by snap freezing on dry ice for 10 min, thawing for 5 min at 65 °C and overnight incubation on a rotating wheel at room temperature. The gel debris was removed by centrifuging the samples through a Spin-X filter (Corning) and the RNA was purified by ethanol precipitation. The size-selected RNA was dephosphorylated for 45 min at 37 °C using T4 PNK (NEB, M0201S).

The dephosphorylated RNA was mixed with pre-adenylated adaptors containing five random nucleotides at their 5′ ends[91] in 1×T4 RNA ligase buffer, 25% PEG-8000, 20 U Superase In and 1 µl T4 RNA ligase 2, truncated KQ (NEB, M0373S). The mix was incubated for 3 h at 25 °C and the ligation products were purified by size selection on a 12% polyacrylamide, 7 M urea and 1×TBE gel. The linker-ligated sample (50 ng) was used for rRNA depletion using a Ribo-Seq riboPOOL h/m/r depletion kit (siTOOLs) for the cycloheximide-only samples, and legacy RiboZero Gold kit (Illumina) for the cycloheximide + tigecycline samples following the manufacturer's instructions. The rRNA-depleted footprints were annealed with RT primer (5′-pRNAGATCGGAAGAGCGTCGTGTAGGGAAAGAG/ iSp18/GTGACTGGAGTTCAGACGTGTGCTC-3′) at 65 °C for 5 min and reverse transcribed for 30 min at 50 °C in an RT master mix containing 1×Protoscript II Buffer, 0.5 mM dNTPs, 10 mM DTT, 20 U Superase In and 200 U Protoscript II (NEB, E6560S). After reverse transcription, NaOH was added to a final concentration of 0.1 M and the RNA was hydrolysed by incubating the samples for 5 min at 90 °C. The cDNA products were purified by size selection on a 12% polyacrylamide, 7 M urea and 1×TBE gel, followed by ethanol precipitation. For cDNA circularization, a 20 µl reaction was prepared containing the gel-purified RT product mixed with 3 µM recombinant TS2126 RNA ligase 1 in circularization buffer (50 µM ATP, 2.5 mM MnCl$_2$, 50 mM MOPS pH 7.5, 10 mM KCl, 5 mM MgCl$_2$, 1 mM DTT and 1 mM betaine) and incubated for 3 h at 60 °C, followed by heat inactivation for 10 min at 80 °C. Libraries were constructed from circularized cDNA using the same primers as for tRNASeq. Amplification was performed using KAPA HiFi DNA polymerase (Roche) in 1×HiFi buffer with an initial denaturation at 95 °C for 3 min, followed by 6–10 cycles of 98 °C for 20 s, 62 °C for 30 s and 72 °C for 15 s at a ramp rate of 3 °C s$^{-1}$. The PCR products were purified by size selection on an 8% polyacrylamide and 1×TBE gel. Excised gel slices were crushed with a pestle and DNA was eluted by rotating samples overnight in 300 µl DNA elution buffer (300 nM NaCl, 10 mM Tris–HCl pH 7.5 and 0.2% Triton X-100). After ethanol precipitation, the libraries were quantified using a Qubit dsDNA high sensitivity kit and 75–86-bp single-end sequencing was performed on an Illumina NextSeq 500 platform, generating >19 × 10$^6$ reads per library.

## Analysis of RNA-Seq data

RNA-Seq datasets were pre-processed to remove potential 3′ adaptors using Trim Galore v0.6.4 with default settings, retaining reads with a length of ≥20. The reads were aligned to the GRCh38 human genome using STAR v2.6.1c with the following parameters: --outSAMtype BAM SortedByCoordinate --outFilterMultimapNmax 1 --outFilterMismatchNmax 1 --quantMode TranscriptomeSAM GeneCounts. The featureCounts v1.6.2 software was used to count reads overlapping a filtered set of protein-coding gene annotations from the GENCODE basic gene annotation. Differential gene expression analysis was performed using DESEq2 v1.38.1. Gene expression heatmaps were generated using ComplexHeatmap v2.14.0 (ref. [92]) by combining standardized gene counts and significant log$_2$-transformed fold-change values from DESeq2 ($P$adj ≤ 0.05).

## Analysis of tRNASeq data

Demultiplexing and 3′ sequencing adaptor removal was performed using cutadapt v3.5. Indels were disallowed (--no-indels) and both read ends were quality trimmed with a quality score of 30 (-q 30,30).

As sequencing was performed with more cycles than the length of any sequenced fragment, all reads were expected to contain adaptors and only trimmed reads were retained with --trimmed-only. The reads were further trimmed to remove the two 5′-RN nucleotides introduced by circularization from the RT primer with -u 2. In both processing steps, reads <10 nt were discarded using -m 10. Analysis of tRNA expression and modification was performed with v1.2 of the mim-tRNAseq computational package (https://mim-trnaseq.readthedocs.io/ en/latest/index.html)[21]. Briefly, the full set of 619 predicted tRNA genes for the hg38 human genome assembly were downloaded from GtRNAdb[93] and the 22 mitochondrially encoded human tRNA genes were fetched from mitotRNAdb[94]. After intron removal and the addition of 5′-G (for tRNA-His) and 3′-CCA (for nuclear-encoded transcripts), a curated set of 599 nuclear-encoded tRNA sequences (excluding tRNAs with non-canonical secondary structure alignments or undetermined anticodons) and 22 mitochondrially encoded tRNA sequences was compiled as an alignment reference (--species Hsap). The reads were aligned to this reference with a cluster ID of 0.97, maximum mismatch tolerance at a number of nucleotides equal to 7.5% read length for the first alignment round and 5% read length for realignment, a deconvolution coverage ratio of 0.4 at mismatch sites to allow accurate cluster deconvolution and a minimum coverage threshold of 0.05% total reads per transcript for low coverage transcript filtering. The following command was used: mimseq --species Hsap --cluster-id 0.97 --threads 40 --min-cov 0.0005 --max-mismatches 0.075 --control-condition kiPSC --deconv-cov-ratio 0.4 -n hg38_diff --out-dir hg38_WTdiff_2rep_deconv0.4_ID0.97_0.075_remap0.05_v12/ --max-multi 6 –remap --remap-mismatches 0.05 sampleData_ht_ diff_2rep.txt.

In addition, DESeq2 was run on tRNA transcripts with single-transcript resolution by first removing those still in clusters from the counts table (evidenced by the presence of multiple transcripts in the name, separated by '/') and repeating DESeq2 analysis on these. Isotype counts, generated by aggregating anticodon counts for the same tRNA isotype were also generated, and DESeq2 was additionally run on this count data.

## Codon usage analysis

We used RSEM v1.3.1 to calculate coding-gene expression in TPM. First, we built a custom reference transcriptome annotation, which was defined on the basis of APPRIS annotations[95]. From these we extrapolated the MANE-annotated transcript for each gene and retained transcripts with a coding sequence beginning with an AUG codon and ending with a UAG/UAA/UGA codon, a nucleotide length that was a multiple of three and no unidentified bases. Sequences without a perfect match with a protein sequence in UniProtKB/SwissProt were removed, yielding a reference containing 16,731 transcripts.

An RSEM reference for read alignment using STAR was built using rsem-prepare-reference with the --star option enabled. For TPM calculation, we used this reference and adaptor-trimmed RNA-Seq reads with rsem-calculate-expression for each sample.

To calculate the codon usage in each sample, we weighted the 61 sense codon frequencies of each transcript in our custom annotation by the TPM expression of the transcript in that sample. We separately counted start AUG codons from coding sequence AUG codons for the distinction between dynamics at start and coding sequence methionine codons. These raw codon usages were additionally summed across all transcripts to generate aggregated codon usage per codon. For normalization, these values were divided by the sum of all codon usages per sample, representing proportional codon usage.

For comparison to tRNA anticodon abundance, we utilized raw mim-tRNAseq read counts summed by anticodon and converted to proportions of total tRNA-aligned reads. Where no perfect match between anticodon and codon was available due to wobble pairing, we duplicated the anticodon abundance of tRNAs that are known to

wobble pair to such codons, such that all 61 sense codons had corresponding tRNA anticodon abundance values.

### Ribosome profiling data analysis

Sequencing libraries were demultiplexed and adaptor-trimmed using Cutadapt v3.5 (ref. [96]) as described for the tRNA-Seq. Trimmed reads >10 nt were aligned to a human rRNA reference using Bowtie v1.2.2 (ref. [97]) with the following options: -p 40 -S --best. Ribosomal RNA-filtered reads were aligned to GRCh38 using STAR v2.6.1c[98] with the following options: --outFilterMultimapNmax 1 --outSAMtype BAM SortedByCoordinate --outFilterMismatchNmax 0 --alignEndsType Local --seedSearchStartLmax 14 --alignIntronMax 10000 --sjdbOverhang 28 --outFilterIntronMotifs RemoveNoncanonicalUnannotated --quantMode TranscriptomeSAM --outSAMattributes NH HI AS nM NM MD. Between $5.3 \times 10^6$ and $21.9 \times 10^6$ pre-processed reads were aligned to coding regions in the GRCh38 transcriptome.

We identified the A- and P-site codon in each open reading frame-mapped read using Scikit-ribo[99], which uses a random forest with recursive feature selection for accurate A-site prediction and a generalized linear model for codon dwell time estimation based on matched ribosome profiling and RNA-Seq datasets. Kallisto 0.44.0 with the parameters -b 100 --single -l 180 -s 20 -t 40 was used to quantify transcript abundances in TPM from RNA-Seq data based on the reference set of MANE-annotated transcripts. To avoid memory errors due to the large size of the human genome and the presence of multiple transcript isoforms, all RNAfold dependencies in Scikit-ribo were omitted and the index was built separately for each chromosome. To make the hg38 Gene Transfer Format file compatible with Scikit-ribo, transcript/untranslated region annotations were removed. For each transcript, the start codon in the first exon and the stop codon in the last exon were adjusted to represent transcript start and end coordinates, taking into account the gene strand. To calculate relative codon dwell times (defined as the difference between the dwell time of each codon and the median of all codon dwell times[99]), short (20–22 nt) and long (28–33 nt) ribosome footprints were analysed separately.

### ChIP–Seq read alignment and multimapping analysis

ChIP–Seq and ATAC–Seq datasets were pre-processed to remove potential 3′ adaptors using Trim Galore v0.6.4 with default settings, retaining reads with a length of ≥20. Given the high frequency of tRNA gene duplication, which can include flanking sequences[4,100], we first analysed the extent of multimapping for RPC1 ChIP–Seq reads mapping to predicted tRNA genes. First, 2 × 110-bp paired-end reads from RPC1 ChIP–Seq libraries were aligned to the human GRCh38 reference genome using STAR v2.6.1c, allowing up to one mismatch per read (--outFilterMismatchNmax 1), up to ten alignment positions (--outFilterMultimapNmax 10) and in end-to-end alignment mode with prohibited introns in reads (--alignEndsType EndToEnd --alignIntronMax 1). Read duplicates were then removed using Picard Tools MarkDuplicates v2.17.10, with REMOVE_DUPLICATES = true to enable direct filtering of duplicates in the output binary alignment map (bam) file. The mmquant v1.3 (ref. [101]) tool was used to count reads overlapping the 619 predicted tRNA genes, with each gene extended by 125 bp of upstream and downstream sequence. Using a custom Python script, the mmquant output was parsed such that for each library input, an output was produced consisting of tRNA genes as rows and one column each for uniquely mapping read counts, multimapping read counts and the proportion of total reads per tRNA represented by multimapping reads. We defined tRNA genes that are not distinguishable in ChIP–Seq data by finding the consensus list of tRNA genes with ≥25% multimapping reads and ≥50 total aligned reads in RPC1 ChIP–Seq libraries (n = 61 from 27 isodecoders and 16 anticodon families; Supplementary Table 4). As expected, 20 of the 23 tRNA genes in the four tandem repeats of a cluster of tRNA genes on chromosome 1 (Glu-CTC, Gly-TCC, Asp-GTC, Leu-CAG and Gly-GCC)[100] fall within

this group. These 61 tRNA genes were excluded from all gene-level analyses of ChIP–Seq and ATAC–Seq datasets. Given that nearly all multi-mapped reads aligned to identical gene copies coding for the same tRNA transcript, one alignment position was randomly chosen and reported for such reads in Pol III occupancy and chromatin accessibility analysis aggregated by tRNA transcript.

### ChIP–Seq and ATAC–Seq peak calling and annotation

Adaptor-trimmed ChIP–Seq and ATAC–Seq libraries were aligned to the GRCh38 reference genome using STAR with the following settings: up to one mismatch per read, a maximum of ten alignment positions, end-to-end alignment, prohibited introns and only one alignment reported per read (--outFilterMismatchNmax 1 --outFilterMultimapNmax 10 --alignEndsType EndToEnd --alignIntronMax 1 --outSAMmultNmax 1). Reads from spike-in-containing libraries were also aligned to the *D. melanogaster* r6.39 genome with the same settings, except only uniquely mapped reads were retained (--outFilterMultimapNmax 1). Read duplicates were then removed using Picard Tools MarkDuplicates v2.17.10 as described earlier, and for ATAC–Seq libraries, alignments to the mitochondrial genome were also filtered. To account for dimerization of the transposon before insertion[49], filtered ATAC–Seq reads were additionally shifted by +4 bp and −5 bp for positive and negative strand alignments, respectively, using deepTools alignmentSieve v3.4.0, which was simultaneously used to split fragments with a maximum length of 100 nt representing NFRs. Both operations were performed simultaneously using the --ATACshift and --maxFragmentLength 100 parameters. The ATAC–Seq NFR alignments were then converted to BEDPE format for peak calling using alignmentSieve --BED.

Peaks were called using MACS callpeak v2.2.6, supplying ChIP input samples from HPSI0214i-kucg_2 for the *kucg-2* hiPSC and CM datasets, HPSI0214i-wibj_2 for the *wibj-2* hiPSC datasets and from HPSI0214i-kucg_2-derived NPC for NPC and neuron datasets, specifying the fragment sizes (--extsize) with shifting model building disabled (--nomodel). The small region size used to calculate dynamic lambda was reduced to 500 bp (--slocal 500) and peak summits were also reported (--call-summits). MACS peak calling was performed on all reads without duplicate removal (--keep-dup all), as these had previously been filtered for duplicates using Picard Tools. For the ATAC–Seq peak calling, the BEDPE files generated above were used (--f BEDPE) without the corresponding control input samples, shifting model building was not disabled and fragment sizes were not specified. For the H3K27me3 ChIP–Seq peak calling, the --broad parameter was additionally specified to call broad peaks for this mark. For both data types, significant peaks were called if the FDR-adjusted Poisson distribution *P* value was ≤0.05. Predicted peaks were filtered using the ENCODE project unified GRCh38 blacklist regions bed file (https://www.encodeproject.org/files/ENCFF356LFX/) by identifying overlaps using bedtools intersect v2.29.2. The blacklist-filtered peak region summits were then annotated by searching for their nearest predicted tRNA locus with bedtools closest using the filtered set of tRNA genes excluding those with significant 'within isodecoder' multimapping reads in hiPSC, as defined above. Peaks with tRNA 'hits' were then defined for each sample as those within 125 bp of an annotated tRNA gene, whereas tRNA hits shared by both biological replicates of an experimental condition were used to define consensus tRNA peaks for that condition. Using RPC1 tRNA peak datasets, we defined housekeeping tRNAs as those that are shared between consensus sets for all cell types and those that were absent from all consensus lists constitute persistently inactive tRNAs. Repressed tRNA genes were defined as the difference between the union of all tRNA peaks in all cells and the housekeeping set.

### ChIP–Seq coverage normalization and visualization

For visual analysis of ChIP–Seq datasets, duplicate-filtered bam files were converted into normalized bigWig signal tracks using deepTools

v3.5.1. To calculate the required normalization factors for individual libraries, mmquant was used to count ChIP–Seq reads that overlapped the annotated hg38 tRNA genes extended by 125 bp at both ends. Using edgeR v3.34.1, normalization factors were calculated using these counts as input for the calcNormFactors function using the 'RLE' method. Relative library sizes were taken as the sum of reads assigned to tRNA features, scaled per million reads. The edgeR normalization factors were further multiplied by these library-size factors and the reciprocal of this product was used for normalized signal generation. DeepTools bamCoverage with a normalization bin size of 1 bp (--binSize 1), the previously calculated scale factors (--scaleFactor) and read extension using fragment lengths that were previously estimated by Phantompeakqualtools (--extendReads) was implemented to generate normalized signal files. Plotting of this signal was performed with deep-Tools computeMatrix (in reference-point mode) and plotHeatmap, using the tRNA gene start as a reference (--referencePoint TSS), bed files of housekeeping, repressed and inactive tRNAs as regions (-R), and either 500 bp or 1,000 bp flanking the tRNA gene start (-a 500 -b 500 or -a 1000 -b 1000, respectively).

### Differential occupancy analysis with DiffBind

For the differential occupancy analysis, we utilized DiffBind v3.2.7 and specified the set of human tRNA genes filtered for <25% multimapping reads (as described earlier) for inclusion in the analysis. This enabled us to obtain occupancy analysis results for all tRNA genes regardless of the presence or absence of a ChIP peak. Briefly, we first generated a bed file of these tRNA genes extended by 200 bp on either end to capture all ChIP signals around each tRNA. To avoid peak merging by DiffBind, overlapping regions in these extended features (for tRNAs separated by less than 200 bp) were determined, using bedtools intersect, and subtracted from the extended features using bedtools subtract. Sample sheets specifying duplicate-filtered bam files for alignments to the human ('BamReads' column) and *D. melanogaster* ('SpikeIn' column) genomes, extended and processed tRNA regions ('Peaks' column) as well as metadata such as condition and replicate were supplied for DiffBind analysis. After read counting (dba.count), blacklisted regions were not filtered, as this had previously been done after peak calling, but non-redundant sets of greylist regions were determined and excluded from analysis using dba.blacklist with blacklist = FALSE. Normalization and differential occupancy analysis were performed using dba.normalize with RLE normalization from DESeq2 (Benjamini–Hochberg-adjusted Wald test *P* value) combined with spike-in normalization (normalize = DBA_NORM_RLE, spikein = TRUE) and dba.analyse. Finally, the results for individual contrasts were retrieved using the DiffBind dba.report function, and annotation information (that is, tRNA gene name) was restored using the annotatePeakInBatch function of ChIPpeakAnno v3.26.4.

### Whole-genome bisulfite sequencing analysis

Public whole-genome bisulfite sequencing data for the human H1 human embryonic stem cell line were obtained from ENCODE project number ENCSR617FKV (Gene Expression Omnibus (GEO): GSE80911) by downloading the processed bed files of methylation state at CpG nucleotides for both biological replicates (ENCFF434CNG and ENCFF573YXL). A custom tRNA annotation was generated by extending each tRNA gene with 125 bp upstream for the filtered set of tRNA genes without significant multimapping in the RPC1 ChIP–Seq datasets. The CpG methylation data were then matched to these annotations using bedtools intersect v2.29.2. The proportion of CpG methylation, present in column 11 of the bed files, was plotted per biological replicate separated by tRNA gene activity defined by RPC1 occupancy in the four cell types ('ChIP–Seq and ATAC–Seq peak calling and annotation' section).

### Sequence motif analysis

To compare A- and B-box sequences in the three activity classes of tRNAs defined from the RPC1 ChIP–Seq data, we first generated multiple sequence alignments of all human hg38 tRNA genes to tRNA covariance models using the cmalign command from Infernal v1.1.2. We then extracted A- and B-box sequences from these alignments corresponding to positions 9–21 and 75–85, respectively. Sequence logos were then generated for these subsequences, separated by tRNA activity class, using the Python package logomaker v0.8.

To define genome-wide motifs for A- and B-box promoter sequences, we used the online MEME prediction tool[102] (https://meme-suite.org/meme/tools/meme) and uploaded all 619 predicted tRNAs in the hg38 genome from GtRNAdb[93]. Motif prediction was run in classic mode, with one occurrence per sequence allowed per motif, as is expected for A and B boxes in tRNA sequences. The search was limited to two motifs with a width of 9–11 nt, based on previous predictions of A- and B-box consensus motif lengths. Finally, motif searching was limited to the given strand only, as the supplied sequences were mature tRNA and not DNA. MEME found exactly two motifs in the input sequences and the consensus for each corresponded to known A- and B-box consensus sequences.

The results in XML format were downloaded, imported into R v4.2.2 using the read_meme function and plotted using view_motifs from universalmotif v1.16.0. These were converted into position weight matrices using universalmotif convert_type for each motif instance. Motif densities were calculated using a customized version of the seqPattern function plotMotifDensityMap that returns the motif densities in each sequence. Briefly, motifScanHits is called using the imported position weight matrices to return motif hits in each sequence passing a minimum motif counting score of 90% (minScore = 90%). Two-dimensional binned kernel density estimates were then calculated on the motif hits using bkde2D from KernSmooth v2.23 with a bandwidth of 1 bp in both coordinate directions. For each sequence, the maximum density score was extracted and used for comparing distributions of motif densities in each tRNA activity class.

### tRNet architecture

The tRNet CNN is a multi-class CNN implemented in keras v2.2.4 (Tensorflow v1.15.5 backend) to predict the class of tRNA gene as housekeeping, repressed or inactive from genomic input sequence in one-hot-encoded format (A = [1,0,0,0], C = [0,1,0,0], G = [0,0,1,0] and T = [0,0,0,1]). Conceptually, the architecture is based on that described for BPNet[56] with minor adjustments to the size of the receptive field of the network and the output (Extended Data Fig. 5g). Briefly, tRNet consists of an initial convolutional layer with 128 filters and a width of 20 bp, followed by eight consecutive dilated convolutional layers with 128 filters and a width of 10 bp, where the dilation rate is doubled at each layer. Such exponential dilation rates double the number of skip positions in the convolutional filter, effectively increasing the complexity of pattern learning and the receptive field in sequence space that is visible to the network. Each convolutional layer is followed by a rectified linear activation ($f(x) = \max(0, x)$). A global max pooling follows the convolutional layers and precedes the fully connected hidden layer, which contains 32 neurons. The tRNet output consists of a final fully connected layer with softmax activation to three outputs, each representing the probability of a tRNA gene belonging to each class based on the input sequence.

### tRNet transfer learning approach

During the training of tRNet we utilized a transfer learning approach from a network trained for a binary classification task. In this network the architecture is identical to that of tRNet, except that the final output layer consists of a single sigmoid activated output to predict whether the input sequence belongs to a housekeeping tRNA or not. Inputs for this model also only consisted of sequences from the subset of tRNA genes in housekeeping and inactive classes. Given the more distinct sequence difference between these two classes, this is a simpler classification problem from which learned features are exploited for

better generalization in the final multi-class model. Transfer learning was achieved by training the modified model on input sequences from housekeeping and inactive genes, and their gene class labels obtained from called peaks from ChIP–Seq data. Next, all layers were frozen to prevent retraining of already trained layers and the model architecture was updated to replace the output layer with one producing three softmax activated outputs, as described earlier. This model was then retrained on one-hot-encoded sequence data from all three classes and their corresponding tRNA gene class labels. Finally, the last convolutional layer of the network was unfrozen and the model trained once again to optimize the weights of this layer for the new multi-class model.

### CNN training and evaluation

All networks were trained with the same approach on 80% of the input data, and validated on 20% held-out data. To evaluate model performance, K-fold cross-validation ($k = 5$) was implemented on training data, and performance in the form of validation accuracy and loss across the five folds was compared. Training of the initial binary classification model, from which learning was transferred, was implemented using the Adam optimizer (learning rate = 0.00025, as determined by parameter hypertuning), binary cross-entropy loss function and early stopping with patience of ten epochs. For final model training, after transfer learning, the same training parameters were specified, except that a categorical cross-entropy loss function more tailored to the multi-class output of this model was used. Final model performance was evaluated on held-out testing data and the accuracy the of model predictions for each class was assessed using the AUROC by plotting the one-versus-rest macro-average scores.

### Nucleotide contribution score calculation and motif analysis with TF-Modisco

To calculate the contribution scores of each nucleotide in each input sequence to the final prediction, we employed the SHAP DeepExplainer module, an extension of DeepLift for calculating SHAP contribution scores. These contribution scores, one for every nucleotide in every input sequence, are based on the difference in output between the model given a set of shuffled input sequences and the output of the model on actual tRNA upstream sequence. Ten dinucleotide-shuffled sequences for every input sequence were supplied for the contribution score calculation. The resulting DeepExplainer hypothetical contribution scores were multiplied by the one-hot encoded matrix for each sequence to derive the final contribution scores for each sequence. The hypothetical and final contribution scores were calculated separately for every output, or task, of the model, corresponding to classification of sequences as housekeeping, repressed or inactive tRNAs.

TF-Modisco v0.5.14.1 was then run on the contribution scores from SHAP DeepExplainer for each task separately to find sequence enrichment or motifs among nucleotides with high contribution to model output. Significant high-importance windows in the sequences, or seqlets, were detected using a sliding window size of 15 bp, a flanking sequence of 5 bp and seqlet FDR threshold of 0.01 (TfModiscoWorkflow(sliding_window_size = 15, flank_size = 5, target_seqlet_fdr = 0.01)). Final patterns were assembled from detected seqlets with a window size of 20 bp, flaking sequence of 10 bp and a minimum of 20 seqlets per cluster (TfModiscoSeqletsToPatternsFactory(trim_to_window_size = 20, initial_flank_to_add = 10, final_min_cluster_size = 20).

### Enhancer analysis

Enhancer elements for the GRCh38 genome were obtained from the UCSC GeneHancer Double Elite regulatory elements table, fetching elements whose identification and association to target genes are derived from more than one information source. First, enhancer elements per chromosome were downloaded with the table browser using a filter for 'Enhancer' in elementType (accessed 14 April 2023);

these were merged to obtain all Double Elite GeneHancer enhancers in the GRCh38 genome. To find tRNA genes that overlap with this set of enhancers, bedtools closest v2.29.2 was used to obtain the closest enhancer to each of the tRNA loci with gene-resolution RPC1 ChIP occupancy data ($n = 558$); those overlapping tRNAs (distance of 0 bp) were retained ($n = 55$). As an additional source of evidence for enhancer activity, overlaps with FANTOM5 CAGE data were obtained from https://fantom.gsc.riken.jp/5/datafiles/reprocessed/hg38_latest/extra/enhancer/F5.hg38.enhancers.bed.gz and CAGE peaks overlapping all enhancer elements were identified using bedtools intersect v2.29.2. From this set of FANTOM5 CAGE-overlapping enhancers, those that also overlapped tRNAs were found with a combination of bedtools closest and filtering for a distance of 0 bp, as above. A table containing tRNA and overlapping enhancer position and identity information, FANTOM5 CAGE peak overlap, tissue and/or cell type specificity of each of these tRNA-associated enhancers as well as their predicted target genes was compiled (Supplementary Table 3). Tissue/cell type and target information were obtained by request from the GeneCards database (https://www.genecards.org/Guide/Datasets) for GeneHancer v5.16. Co-regulation with overlapping enhancers was assessed for tRNA genes that (1) were in the repressed and housekeeping class, (2) overlapped a Double Elite enhancer with FANTOM5 CAGE support and (3) demonstrated evidence of tissue/cell-type specificity in a cell type related to those used in this study. If an enhancer overlapped more than one tRNA, only those where the tRNAs are of the same activity status were retained for analysis.

### Statistics and reproducibility

No statistical method was used to pre-determine sample size. The experiments were not randomized and the investigators were not blinded to allocation during experiments and outcome assessment. No data were excluded from the analyses. Information on the statistical tests used for each analysis and reproducibility is included in the relevant sections describing the method as well as in the figure legends.

### Reporting summary

Further information on research design is available in the Nature Portfolio Reporting Summary linked to this article.

## Data availability

High-throughput sequencing data have been deposited in the GEO database (GSE227928). The public genome-wide bisulfite sequencing data used here are available through ENCODE project ENCSR617FKV (GEO: GSE80911). Source data are provided with this paper. All other data supporting the findings of this study are available from the corresponding author on reasonable request.

## Code availability

The mim-tRNAseq data analysis pipeline is available at https://github.com/nedialkova-lab/mim-tRNAseq. The tRNet code is available at https://github.com/nedialkova-lab/tRNet. Customized Scikit-ribo v0.2.4b1 for use on the human genome is available at https://github.com/nedialkova-lab/scikit-ribo-ext.

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

## Acknowledgements

We thank the Protein Production and Imaging Core Facilities and the NGS Facility in the Department of Totipotency at the Max Planck Institute of Biochemistry for technical support; C. Wu, N. Sinha, J. Gagneur, P. T. da Silva and M. Colomé-Tatché for advice; J. Sterneckert and B. Greber for help with establishing differentiation protocols; M. M. R. Pinto and S. Dodgson for manuscript feedback; and F. Bonneau and E. Conti for sharing resources. D.D.N. acknowledges the Wellcome Trust Sanger Institute as the source of the HPSI0214i-kucg_2 and HPSI0214i-wibj_2 cell lines, which were generated under the Human Induced Pluripotent Stem Cell Initiative supported by the Wellcome Trust (grant number WT098051) and the National Institute for Health Research/Wellcome Trust Clinical Research Facility, and acknowledges Life Science Technologies Corporation as the provider of Cytotune. A.B., G.R. and S.W. were supported by the International Max Planck Research School for Molecular Life Sciences. S.F. was supported by a postdoctoral fellowship from the Alexander von Humboldt Foundation. This work was funded by the Max Planck Society, the European Research Council under the European Union's Horizon 2020 Research and Innovation Programme (ERC Starting Grant number 803825-TransTempoFold to D.D.N.) and the EMBO Young Investigator Program (grant number 4833 to D.D.N.).

## Author contributions

Conceptualization: D.D.N. Experimental methodology: L.G., A.B., G.R., S.W. and D.D.N. Investigation: L.G., G.R., S.W., K.S. and D.D.N. Formal analysis: A.B., S.F., L.G. and D.D.N. Software: A.B. and S.F. Writing (original draft): A.B., L.G. and D.D.N. Writing (review and editing): L.G., A.B., G.R., S.F., S.W., K.S. and D.D.N. Supervision and funding acquisition: D.D.N.

## Funding

## Competing interests

A.B., G.R. and D.D.N. are listed as inventors on a patent application (WO2021EP72902) filed by the Max Planck Society pertaining to the mim-tRNAseq technology. The other authors declare no competing interests.

## Additional information

**Extended data** is available for this paper at https://doi.org/10.1038/s41556-023-01317-3.

**Correspondence and requests for materials** should be addressed to Danny D. Nedialkova.

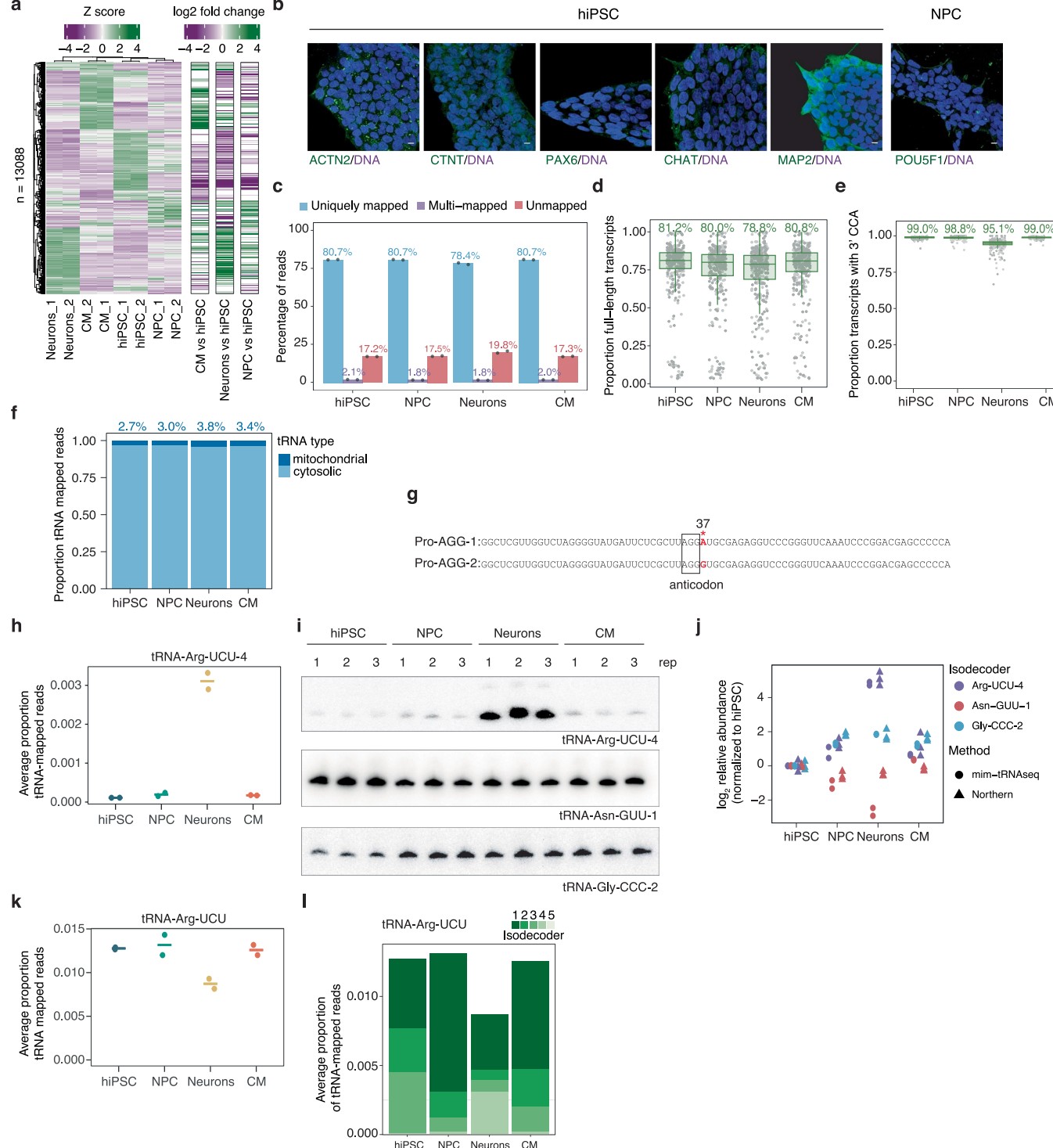

**Extended Data Fig. 1 | See next page for caption.**

**Extended Data Fig. 1 | mim-tRNAseq accurately captures changes in mature tRNA pool composition upon differentiation. a**, Heatmap of differentially expressed mRNAs (Benjamini–Hochberg-adjusted Wald test; *Padj* ≤ 0.05) in at least one cell type relative to hiPSC. Left: hierarchically clustered heatmap of scaled Z score of normalized transcript counts in hiPSC, NPC, neurons and CM ($n = 2$). Right: log$_2$ fold changes for NPC, neurons and CM relative to hiPSC. **b**, Representative fluorescence microscopy images (from at least two independent experiments) of immunostaining for cell type-specific marker proteins (green) and DAPI (blue) in cells with undetectable or substantially lower (*MAP2* in hiPSC) marker gene expression based on RNA-Seq. Scale bar, 10 µm. **c**, Alignment statistics for mim-tRNAseq reads. Bars and percentages: mean values per cell type, dots: individual sample values ($n = 2$ biological replicates). **d-e**, Box plots of full-length read fraction (**d**) and full 3′-CCA end fraction (**e**) per tRNA transcript ($n = 2$ biological replicates; centre line and label: median; box limits: upper and lower quartiles; whiskers: 1.5×interquartile range). **f**, Bar plot of cytosolic and mitochondrial tRNA read fractions per cell type. Bars: mean ($n = 2$ biological

replicates), percentages: mean mitochondrial fraction. **g**, Full transcript sequences for human tRNA-Pro-AGG-1 and tRNA-Pro-AGG-2. Box indicates anticodon, highlighted in bold is the single mismatch that coincides with m$^1$G37 in tRNA-Pro-AGG-2. **h**, Mean expression of neuron-specific tRNA-Arg-UCU-4 in human cell lines (proportions of tRNA-aligned reads from mim-tRNAseq; $n = 2$ biological replicates). **i**, Northern blotting of tRNA-Arg-UCU-4, tRNA-Asn-GUU-1 and tRNA-Gly-CCC-2 ($n = 3$ biological replicates). **j**, Relative abundance of tRNA-Arg-UCU-4, tRNA-Asn-GUU-1 and tRNA-Gly-CCC-2 measured by mim-tRNAseq (Fig. 1f) or Northern blotting (**i**), normalized to the mean value for hiPSC ($n = 2$ biological replicates for mim-tRNAseq; matched total RNA samples). **k**, Abundance of tRNA-Arg-UCU in hiPSC and differentiated cells (proportions of tRNA-aligned reads from mim-tRNAseq). Line is mean, dots represent individual sample values ($n = 2$ biological replicates). **l**, Proportional isodecoder composition changes for tRNA-Arg-UCU upon differentiation of hiPSC. Values are mean proportions of tRNA-mapped reads per isodecoder ($n = 2$ biological replicates). Source numerical data and unprocessed blots are provided.

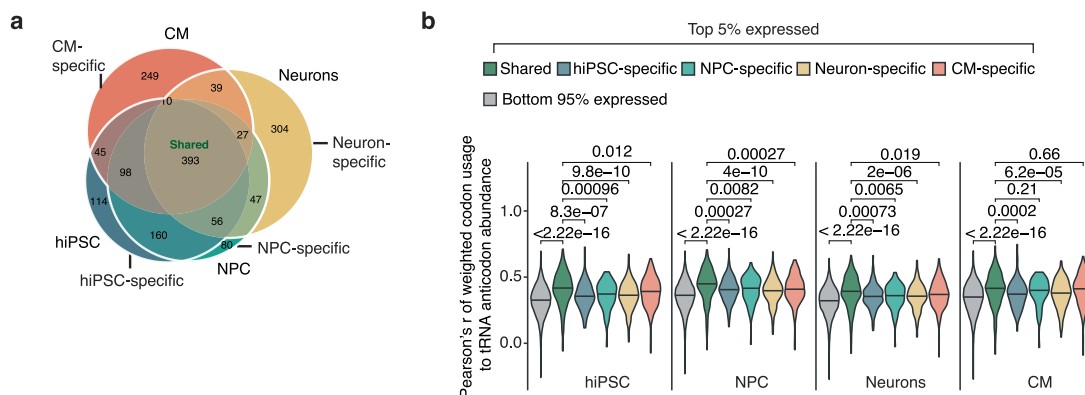

**Extended Data Fig. 2 | Codon usage correlates with tRNA abundance in human cells. a**, Size and overlap of top 5% expressed gene sets in each cell type based on mean TPM values ($n = 2$ biological replicates for each cell type). Cell-type or state-specific sets, and shared sets are indicated. **b**, Violin plots indicating distributions of Pearson's correlation coefficients between mean weighted codon usage and mean tRNA anticodon abundance per transcript defined in (**a**) (centre line: median). $P$-values were calculated using Wilcoxon tests. Source numerical data are provided.

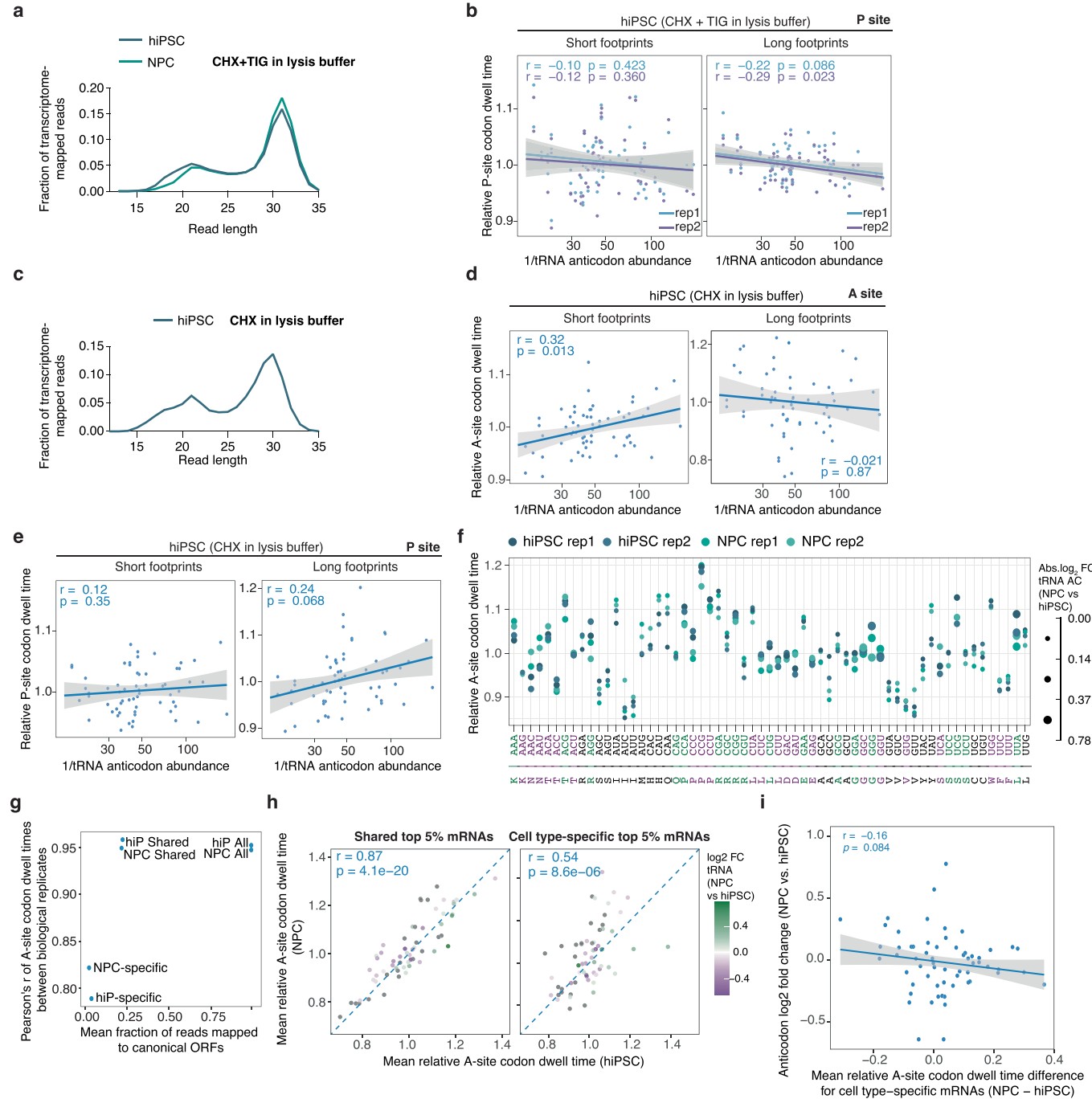

**Extended Data Fig. 3 | Decoding rates correlate with tRNA abundance in human cells. a-b**, Representative length distributions of ORF-mapped reads in ribosome footprint libraries from hiPSC and NPC extracts supplemented with cycloheximide (CHX) and tigecycline (TIG) (**a**) and correlation of 1/tRNA anticodon abundance to hiPSC P-site codon dwell times for short (20–22 nt) and long (28–33 nt) footprints (**b**). **c**, As in (**a**) for ribosome footprint libraries from hiPSC extracts supplemented with CHX only. **d-e**, Correlation of 1/tRNA anticodon abundance to A-site (**d**) and P-site (**e**) codon dwell times for short (20–22 nt) and long (28–32 nt) footprints in ribosome footprint libraries from hiPSC extracts supplemented with CHX only. Solid blue lines: linear regression model; shaded grey: 95% confidence interval (CI); Pearson's correlation coefficient. **f**, A-site codon dwell times estimated from long footprints (28–33 nt) from hiPSC and NPC lysates treated with both CHX and TIG (*n* = 2 biological replicates). Dot size: absolute log₂ fold change in tRNA anticodon abundance in NPC relative to hiPSC (Benjamini–Hochberg-adjusted *Padj* ≤ 0.05); green:

upregulation, purple: downregulation of cognate tRNA anticodon in NPC. **g**, Plot of mean fraction of reads mapped to canonical ORFs relative to cell-type matched analysis for full gene set and corresponding Pearson's correlation coefficient of relative ribosome dwell times between biological replicates calculated for transcript sets as in Extended Data Fig. 2a. **h**, Correlation between mean relative A-site codon dwell time in hiPSC vs NPC (from Fig. 2c,d; long footprints) for shared highly-expressed mRNAs (left panel) and cell type-specific highly expressed mRNAs (right panel). Dots are coloured by log₂ fold changes in tRNA anticodon abundance in NPC relative to hiPSC (*Padj* ≤ 0.05; grey dots denote non-significant changes). Dashed line represents *y* = *x*. Pearson's correlation coefficient displayed. **i**, Correlation between log₂ fold changes in tRNA anticodon abundance in NPC relative to hiPSC (*Padj* ≤ 0.05) and differences in mean codon dwell time between NPC and hiPSC for cell type-specific highly expressed mRNAs. Solid line: linear regression model; shaded grey: 95% confidence interval; Pearson's correlation coefficients. Source numerical data are provided.

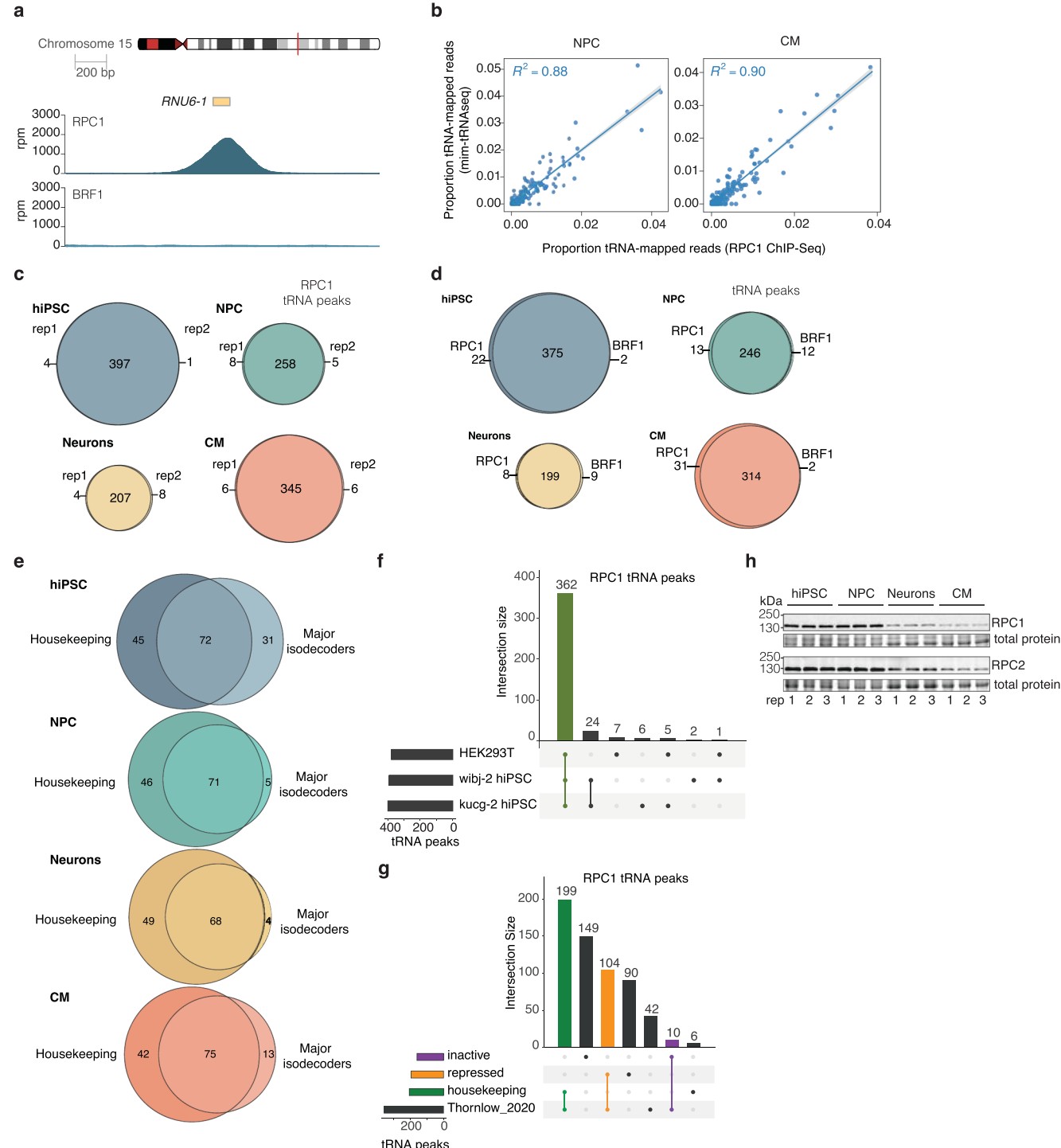

**Extended Data Fig. 4 | See next page for caption.**

**Extended Data Fig. 4 | High-resolution RPC1 and BRF1 ChIP-Seq reveal
Pol III dynamics at tRNA genes during differentiation. a**, Representative
normalized ChIP-Seq signal at a U6 RNA gene (*RNU6-1*) on chromosome 15 of
the human genome for RPC1 and BRF1 ChIP-Seq in one biological replicate of
hiPSC. *y*-axis values: genome-wide ChIP signal normalized to the estimated
library sizes generated from counts over extended tRNA features (±125 bp) and
scaled to reads-per-million (rpm). **b**, Correlation of mean tRNA abundance per
deconvoluted unique transcript (*n* = 373) estimated by mim-tRNAseq to mean
RPC1 ChIP-Seq reads aligned to extended tRNA features (±125 bp) for NPC and CM
(*n* = 2 biological replicates). Both metrics are scaled to proportions of total tRNA-
mapped reads for each dataset and method. Solid blue lines: linear regression
model; shaded grey: 95% confidence interval; Pearson's correlation coefficients.
**c**, Venn diagram indicating overlap of tRNA peaks from replicates of RPC1 ChIP-
Seq datasets. Shared peaks represent the consensus set for each cell type. **d**,
Venn diagram indicating overlap of tRNA peaks between RPC1 and BRF1 ChIP-Seq

consensus sets per cell type (*n* = 2 biological replicates for each cell type). **e**,
Venn diagram indicating overlap between housekeeping tRNA gene set (RPC1
consensus tRNA peak in all four cell types from *n* = 2 biological replicates) and
major isodecoders (contributing 90% to anticodon pool from mim-tRNAseq).
Anticodon families with no detectable expression are excluded. Housekeeping
tRNA genes were collapsed according to identical transcripts to enable matching
to transcript-level data for major isodecoders. **f-g**, UpSet plots of (**f**) significant
consensus RPC1 peaks within 125 bp of annotated tRNA genes per cell type
(FDR-adjusted *P* ≤ 0.05, *n* = 2 biological replicates), and (**g**) comparison of tRNA
gene numbers in three tRNA activity classes and predicted active tRNA genes
in Thornlow et al.[43]. Lower left: barplot of total detected consensus tRNA peaks
for each cell type/group/publication. Right: barplot of intersection set size of
consensus tRNA peaks (upper) in the given intersection (lower). **h**, Immunoblots
of RPC1 and RPC2 in hiPSC, NPC, neurons, and CM (*n* = 3 biological replicates).
Source numerical data and unprocessed blots are provided.

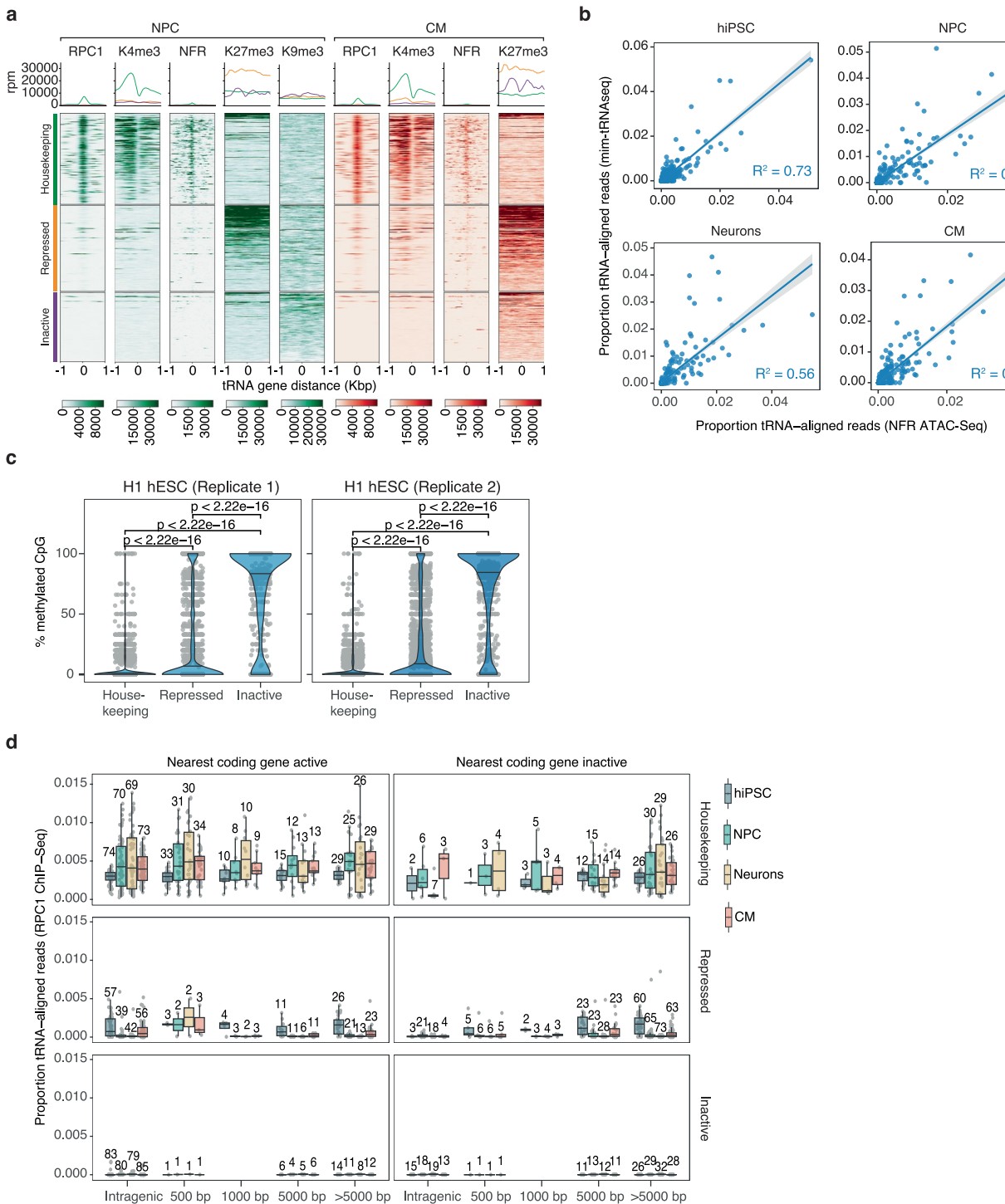

**Extended Data Fig. 5 | Relationship between Pol III occupancy at tRNA genes with chromatin status and nearby gene activity. a**, Heatmaps showing ChIP-Seq signal (RPC1, H3K4me3, H3K27me3, H3K9me3) and nucleosome-free regions (NFR) from ATAC–Seq around tRNA gene start sites (±1 kbp) for single replicates of NPC and CM. Normalized signal, accounting for estimated library sizes generated from counts over extended tRNA features (±125 bp) is scaled to reads-per-million (rpm). tRNA genes are separated into housekeeping, repressed and inactive based on significant peaks in RPC1 ChIP-Seq data (FDR-adjusted $P \leq 0.05$), and sorted in descending order based on mean value per region. **b**, Correlation of mean ATAC–Seq NFR reads aligned to extended tRNA features (±125 bp) to tRNA abundance per deconvoluted unique transcript ($n = 373$) estimated by mim-tRNAseq ($n = 2$ biological replicates for each cell type; Pearson's correlation coefficients). Both metrics are scaled to proportions of

total tRNA-mapped reads for each dataset and method. Solid blue lines: linear regression model; shaded grey: 95% confidence interval. **c**, Violin plots of CpG methylation proportions separated by tRNA activity (centre line: median) from ENCODE. *P* values are from Wilcoxon tests. **d**, Boxplot showing the distribution of mean RPC1 ChIP-Seq reads aligned to extended tRNA features (±125 bp) as a function of the distance between tRNA genes in different activity classes to their nearest neighbouring coding gene (centre line and label: median; box limits: upper and lower quartiles; whiskers: 1.5 × interquartile range). Data are separated by activity class of tRNA, and whether the neighbouring coding gene is predicted to be active by presence of upstream H3K4me3 and ATAC–Seq NFR peaks ($n = 1$ for H3K4me3 in NPC, $n = 2$ biological replicates for all others). Source numerical data are provided.

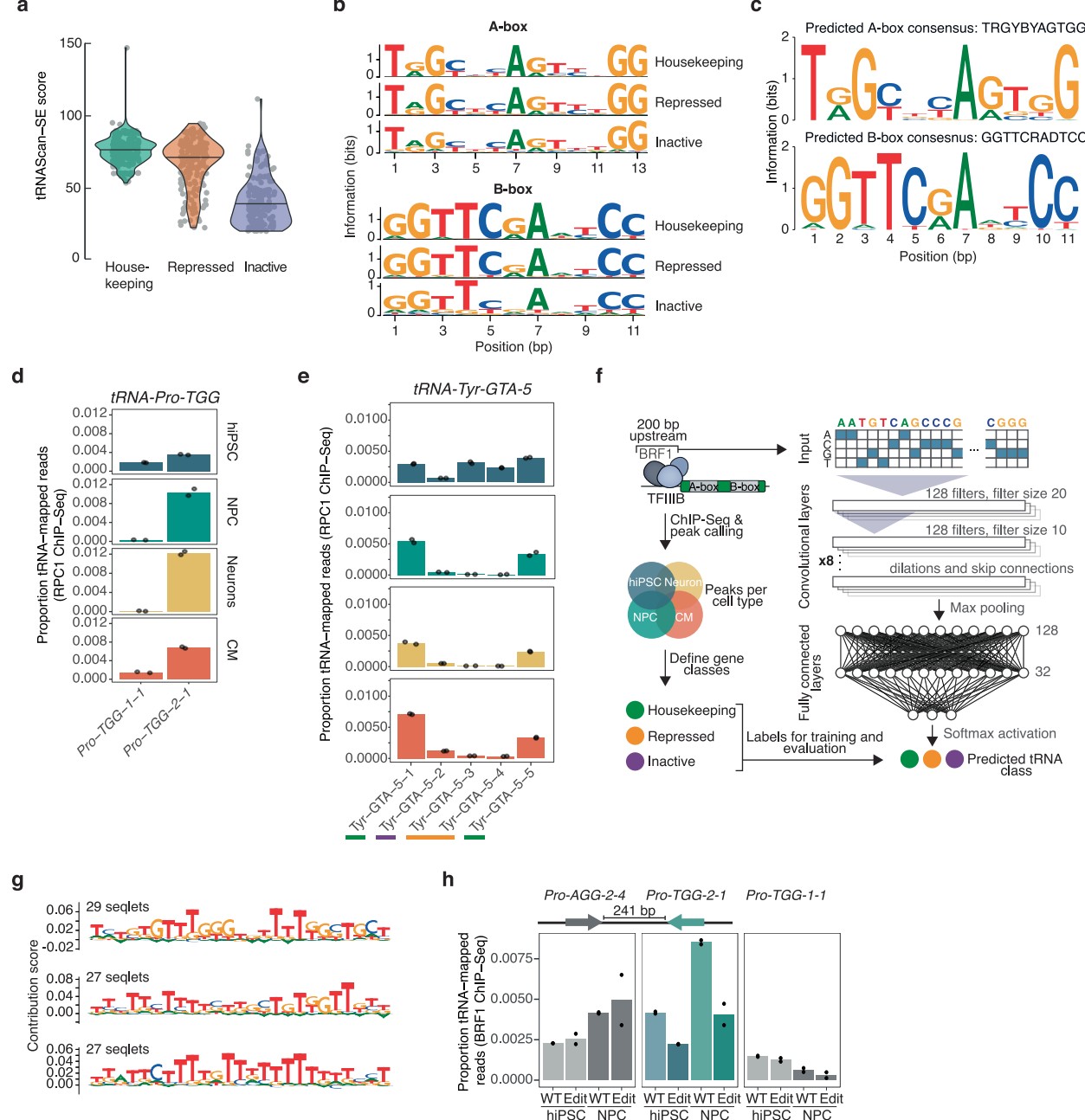

**Extended Data Fig. 6 | Sequence determinants of selective tRNA gene expression in human cells. a**, Violin plot showing the distribution of tRNAScan-SE score per tRNA gene (*n* = 558) separated by tRNA activity (centre line: median). **b**, Sequence logos of human A- and B-box promoter sequences from aligned mature hg38 tRNA, separated by tRNA activity. **c**, MEME-predicted human A- and B-box consensus sequences across the complete hg38 tRNA gene set (*n* = 619). **d**, Mean RPC1 ChIP-Seq reads aligned to extended tRNA features (±125 bp) for *tRNA-Pro-TGG-1-1* and *tRNA-Pro-TGG-2-1* across different cell types. Individual sample values indicated by dots (*n* = 2 biological replicates). **e**, Mean fraction of RPC1 ChIP-Seq reads aligned to extended tRNA features (±125 bp) for *tRNA-Tyr-GTA-5* gene copies across different cell types. Colour bars under gene names indicate

activity class (green: housekeeping; orange: repressed; purple: inactive). Bars represent the mean (*n* = 2 biological replicates); dots indicate individual sample values. **f**, Schematic of tRNet architecture and training. **g**, Top three significant TF-Modisco-generated sequence motif patterns for prediction in inactive tRNA task (FDR-adjusted *P* ≤ 0.01). Displayed are the number of seqlets contributing to the given motif pattern. **h**, Fraction of tRNA-mapped BRF1 ChIP-Seq reads at wild-type ('WT') and CRISPR-edited *tRNA-Pro-TGG-2-1* locus ('Edit', upstream sequence insertion) in hiPSC and NPC (*n* = 2 biological replicates; bar: median); read fractions at the neighbouring *tRNA-Pro-AGG-2-4* and at *tRNA-Pro-TGG-1-1* is shown for comparison. Source numerical data are provided.

**a**

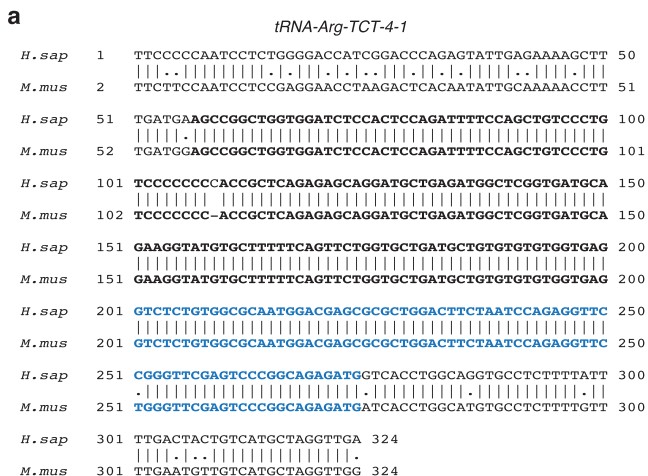

**b**

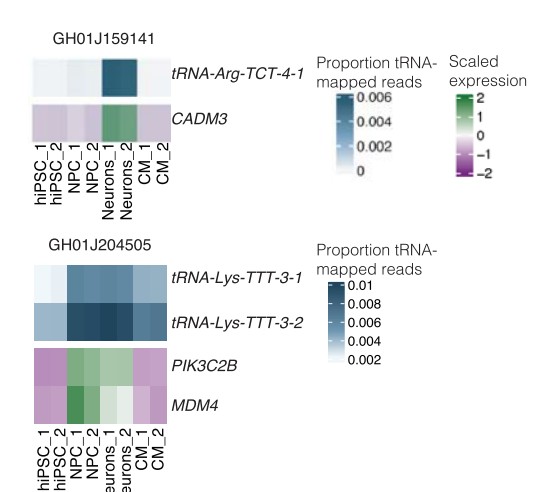

**Extended Data Fig. 7 | *tRNA-Arg-TCT-4-1* overlaps an enhancer and is co-regulated with *CADM3* in neurons. a**, Pairwise sequence alignment of human (hg38) and mouse (mm39) *tRNA-Arg-TCT-4-1* loci, including 200 bp upstream from tRNA gene start and 100 bp downstream of tRNA gene end. Highlighted in blue are *tRNA-Arg-TCT-4-1* gene sequences. Bold black indicates -140-bp

upstream sequence with near-complete sequence identity. **b**, Heatmaps showing proportion of tRNA-mapped RPC1 ChIP reads and scaled *Z* score of normalized transcript counts (from DESeq2) for elite enhancer target genes from RNA-Seq data in hiPSC, NPC, neurons and CM (*n* = 2) for *tRNA-Arg-TCT-4-1* (top) and *tRNA-Lys-TTT-3-1* and *tRNA-Lys-TTT-3-2* (bottom). Source numerical data are provided.

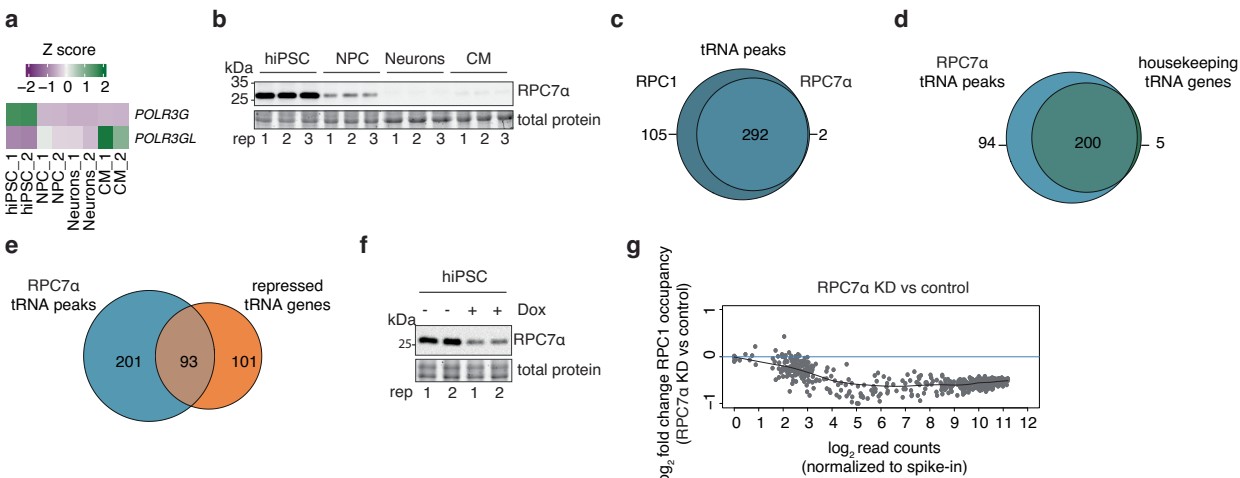

**Extended Data Fig. 8 | Impact of changes in Pol III composition on tRNA gene expression during differentiation. a**, Gene expression heatmaps for *POLR3G* and *POLR3GL* in hiPSC, NPC, neurons and CM (*n* = 2). Scale represents standardized *Z* score calculated using RNA-Seq raw gene counts across samples. **b**, Immunoblots of RPC7α in hiPSC, NPC, neurons, and CM (*n* = 3 biological replicates). **c**–**e**, Venn diagrams depicting overlap between consensus RPC7α ChIP-Seq tRNA peaks in *kucg-2* hiPSC and (**c**) consensus RPC1 tRNA ChIP-Seq peaks in *kucg-2* hiPSC, (**d**) housekeeping tRNA genes, and (**e**) tRNA genes repressed during differentiation. **f**, Immunoblot of RPC7α hiPSC CRISPRi cells carrying a sgRNA against *POLR3G* (*n* = 2 biological replicates). Gene knockdown was induced by addition of 2 μM doxycycline for 2 d. **g**, MA plot generated by DiffBind of spike-in normalized RPC1 counts over tRNA features (±125 bp) vs. log₂ fold-change for (**f**) doxycycline-induced hiPSC carrying a sgRNA against *POLR3G* relative to uninduced controls (*n* = 2 biological replicates). Grey dots: not significant (FDR-adjusted *P* > 0.05). Source numerical data and unprocessed blots are provided.

# Reporting Summary

## Statistics

For all statistical analyses, confirm that the following items are present in the figure legend, table legend, main text, or Methods section.

| n/a | Confirmed | |
|---|---|---|
| ☐ | ☒ | The exact sample size (*n*) for each experimental group/condition, given as a discrete number and unit of measurement |
| ☐ | ☒ | A statement on whether measurements were taken from distinct samples or whether the same sample was measured repeatedly |
| ☐ | ☒ | The statistical test(s) used AND whether they are one- or two-sided *Only common tests should be described solely by name; describe more complex techniques in the Methods section.* |
| ☐ | ☒ | A description of all covariates tested |
| ☐ | ☒ | A description of any assumptions or corrections, such as tests of normality and adjustment for multiple comparisons |
| ☐ | ☒ | A full description of the statistical parameters including central tendency (e.g. means) or other basic estimates (e.g. regression coefficient) AND variation (e.g. standard deviation) or associated estimates of uncertainty (e.g. confidence intervals) |
| ☐ | ☒ | For null hypothesis testing, the test statistic (e.g. *F*, *t*, *r*) with confidence intervals, effect sizes, degrees of freedom and *P* value noted *Give P values as exact values whenever suitable.* |
| ☒ | ☐ | For Bayesian analysis, information on the choice of priors and Markov chain Monte Carlo settings |
| ☒ | ☐ | For hierarchical and complex designs, identification of the appropriate level for tests and full reporting of outcomes |
| ☐ | ☒ | Estimates of effect sizes (e.g. Cohen's *d*, Pearson's *r*), indicating how they were calculated |

*Our web collection on statistics for biologists contains articles on many of the points above.*

## Software and code

Policy information about availability of computer code

| Data collection | Western blot images were collected with iBiright Analysis Software (Thermo Fisher). Norther blot images were collected on a Typhoon FLA 9000 (GE Healthcare). |
|---|---|
| Data analysis | R v4.2.2 and Python v3.7 were used for analysis of NGS data, in addition to the following command-line software: mimseq v1.2 (https://github.com/nedialkova-lab/mim-tRNAseq/tree/master/mimseq) Customized scikit-ribo v0.2.4b1 for use on human genome (https://github.com/nedialkova-lab/scikit-ribo-ext). STAR v2.6.1.c cutadapt v3.5 Trim Galore v0.6.4 RSEM v1.3.1 Kallisto v0.44.0 Picard Tools MarkDuplicates v2.17.10 mmquant v1.3 deepTools alignmentSieve v3.4.0 deepTools v3.5.1 MACS v2.2.6 bedtools v2.29.2 Infernal v1.1.2 Phantompeakqualtools v1.2.2 tRNet CNN model (https://github.com/nedialkova-lab/tRNet) |

R packages:
DESEq2 v1.38.1
edgeR v3.34.1
ComplexHeatmap v2.14.0
DiffBind v3.2.7
ChIPpeakAnno v3.26.4
universalmotif v1.16.0
phantompeakqualtools v1.2.2

Python packages:
logomaker v0.8
keras v2.2.4
SHAP v0.29.3
Tensorflow v1.15.5
TF-Modisco v0.5.14.1

Online tools:
MEME v5.5.4 (https://meme-suite.org/meme/tools/meme)

For manuscripts utilizing custom algorithms or software that are central to the research but not yet described in published literature, software must be made available to editors and reviewers. We strongly encourage code deposition in a community repository (e.g. GitHub). See the Nature Portfolio guidelines for submitting code & software for further information.

## Data

Policy information about availability of data

All manuscripts must include a data availability statement. This statement should provide the following information, where applicable:
- Accession codes, unique identifiers, or web links for publicly available datasets
- A description of any restrictions on data availability
- For clinical datasets or third party data, please ensure that the statement adheres to our policy

High-throughput sequencing data has been deposited in the Gene Expression Omnibus Database (GSE227928). Public genome-wide bisulfite sequencing data used here is available through ENCODE project ENCSR617FKV (GEO: GSE80911). Source data have been provided in Source Data. All other data supporting the findings of this study are available from the corresponding author on reasonable request.

## Research involving human participants, their data, or biological material

Policy information about studies with human participants or human data. See also policy information about sex, gender (identity/presentation), and sexual orientation and race, ethnicity and racism.

| Reporting on sex and gender | N/A |
| Reporting on race, ethnicity, or other socially relevant groupings | N/A |
| Population characteristics | N/A |
| Recruitment | N/A |
| Ethics oversight | N/A |

Note that full information on the approval of the study protocol must also be provided in the manuscript.

# Field-specific reporting

Please select the one below that is the best fit for your research. If you are not sure, read the appropriate sections before making your selection.

☒ Life sciences   ☐ Behavioural & social sciences   ☐ Ecological, evolutionary & environmental sciences

For a reference copy of the document with all sections, see nature.com/documents/nr-reporting-summary-flat.pdf

# Life sciences study design

All studies must disclose on these points even when the disclosure is negative.

| Sample size | No statistical method was used to determine appropriate sample sizes. For all sequencing datasets where comparative statistical analysis was performed, a sample size of two was chosen to allow such statistical tests at an affordable cost. Sample sizes are indicated in the figure legends. |

| | |
|---|---|
| Data exclusions | No data was excluded. |
| Replication | For tRNA-seq, ATAC-seq, RNA-seq and ChIP-seq experiments, 2 biological replicates (independent differentiations) were performed, and correlation analysis were conducted to ensure the consistency between replicates. For H3K27me3 ChIP-seq, H3K9me3 ChIP-Seq, and H3K4me3 ChIP-Seq in NPC, a single replicate per cell type were performed. For ribosome profiling where cycloheximide (CHX) and tigecycline (TIG) were present in lysis buffer, 2 biological replicates for hiPSC and NPC were performed, while only one replicate was performed for the CHX-only sample. For CRISPRi and following functional studies, 2 clones were selected for each sgRNA. All attempts of replications were successful. |
| Randomization | No randomization was performed. This study was carried out in the kucg_2 hiPSC line and its differentiated counterparts, as well as in wibj_2 hiPSC cells and HEK293T/17. Covariates control is not applicable due to the small number of cell lines used. |
| Blinding | The investigators were not blinded to the group as no human subjects or clinical samples were involved and no subjective measurements were taken. |

# Reporting for specific materials, systems and methods

We require information from authors about some types of materials, experimental systems and methods used in many studies. Here, indicate whether each material, system or method listed is relevant to your study. If you are not sure if a list item applies to your research, read the appropriate section before selecting a response.

### Materials & experimental systems

| n/a | Involved in the study |
|---|---|
| ☐ | ☒ Antibodies |
| ☐ | ☒ Eukaryotic cell lines |
| ☒ | ☐ Palaeontology and archaeology |
| ☒ | ☐ Animals and other organisms |
| ☒ | ☐ Clinical data |
| ☒ | ☐ Dual use research of concern |
| ☒ | ☐ Plants |

### Methods

| n/a | Involved in the study |
|---|---|
| ☐ | ☒ ChIP-seq |
| ☒ | ☐ Flow cytometry |
| ☒ | ☐ MRI-based neuroimaging |

## Antibodies

| | |
|---|---|
| Antibodies used | Anti-POU5F1 C-10 (1:400; Santa Cruz, #sc-5279)<br>Anti-SOX2 E-4 (1:200; Santa Cruz, #sc-365823)<br>Anti-NANOG P1-2D8 (1:200; DSHB Hybridoma Product PCRP-NANOGP1-2D8)<br>Anti-PAX6 (1:200; Abcam #ab5790)<br>Anti-Nestin (1:200; R&D Systems, #MAB1259)<br>Anti-MAP2 (1:1000; Abcam, #ab92434)<br>Anti-CHAT (1:200; Abcam, #ab6168)<br>Anti-cTNT (1:5; CT3, deposited to the DSHB by Lin, J.J-C.)<br>Anti-ACTN2 (1:800; Sigma-Aldrich #A7811)<br>Goat anti-mouse Alexa Fluor 488 (1:2000; Thermo Fisher Scientific, #A-11001)<br>Goat anti-rabbit Alexa Fluor 488 (1:2000; Thermo Fisher Scientific, #A-11034)<br>Goat anti-mouse Alexa Fluor 633 (1:500; Thermo Fisher Scientific, #A-21052)<br>Anti-POLR3A/RPC1 (1:1000 for immunoblotting and 5μg for ChIP; Cell Signaling Technology, #12825)<br>Drosophila spike-in antibody (0.2 μg for ChIP; Active Motif, #61686)<br>Anti-H3K4me3 (1:200; Active Motif, #39159)<br>Anti-H3K27me3 (1:200; Millipore, #07-449)<br>Anti-H3K9me3 (1:200; Cell Signaling, #13969)<br>Anti-POLR3G/RPC7α (1:1000 for immunoblotting and 1:100 for ChIP; Santa Cruz, #sc21754)<br>Anti-POLR3B/RPC2 (1:1000; Santa Cruz; #sc-515362)<br>Anti-BRF1 (1:100; Abcam, #ab264191)<br>Anti-MAF1 (1:1000; Santa Cruz; #sc-515614 X)<br>Anti-phospho-4E-BP1 (1:1000; Cell Signaling, #2855T)<br>Anti-phospho-p70 S6 Kinase (T389) (1:1000; Cell Signaling, #9206S)<br>Anti-4E-BP1 (1:1000; Cell Signaling, #9644)<br>Anti-p70 S6 Kinase (1:1000; Cell Signaling, #2708T)<br>Anti-vinculin (1:1000; Cell Signaling; #13901)<br>Anti-mouse IgG-HRP, 1:4000; Dianova, #115-035-003<br>Anti-rabbit IgG-HRP (1:4000; Dianova, #111-035-003) |
| Validation | Anti-POU5F1 C-10 (Santa Cruz, #sc-5279): according to the manufacturer, this mouse monoclonal antibody is raised against amino acids 1-134 of Oct-3/4 of human origin recommended for is recommended for detection of Oct-3/4 of mouse, rat and human origin by immunofluorescence; non cross-reactive with Oct-3/4 isoform B; cited in >2450 publications (https://www.scbt.com/p/oct-3-4- |

antibody-c-10?productCanUrl=oct-3-4-antibody-c-10&_requestid=526329)

Anti-SOX2 E-4 (Santa Cruz, #sc-365823): according to the manufacturer, this mouse monoclonal antibody is specific for an epitope mapping between amino acids 170-201 within an internal region of Sox-2 of human origin and is recommended for is recommended for detection of Sox-2 of mouse, rat and human origin by immunofluorescence; cited in >260 publications (https://www.scbt.com/p/sox-2-antibody-e-4?requestFrom=search).

Anti-NANOG P1-2D8 (DSHB Hybridoma Product PCRP-NANOGP1-2D8): according to the manufacturer, this monoclonal antibody was raised against amino acids 1-127 of the human NANOG protein and is recommended for detecting human NANOG by immunofluorescence (https://dshb.biology.uiowa.edu/PCRP-NANOGP1-2D8).

Anti-PAX6 (Abcam #ab5790): according to the manufactirer, this polyclonal antibody is suitable for detecting PAX6 from mouse, human, rat, and monkey by immunofluorescence; cited in >100 publications (https://www.abcam.com/products/primary-antibodies/pax6-antibody-ab5790.html?productWallTab=ShowAll)

AAnti-Nestin (R&D Systems, #MAB1259): according to the manufactirer, this monoclonal antibody detects human nestin by immunofluorescence; cited in >100 publications (https://www.rndsystems.com/products/human-nestin-antibody-196908_mab1259?gclid=Cj0KCQjw4NujBhC5ARIsAF4Iv6eJ-nyTOhCazimeegmTVBXGL9pW6doy4o12apC7F5i93ffSmtPhOxUaAtgNEALw_wcB&gclsrc=aw.ds#product-details)

Anti-MAP2 (Abcam, #ab92434): according to the manufactirer, this polyclonal antibody detects human MAP2 by immunofluorescence; cited in >45 publications (https://www.abcam.com/products/primary-antibodies/map2-antibody-ab92434.html)

Anti-CHAT (Abcam, #ab6168): according to the manufactirer, this polyclonal antibody was raised against a peptide corresponding to amino acids 168-189 of Choline Acetyltransferase and reacts with the protein in immunohistochemitry; cited in >13 publications (https://www.abcam.com/products/primary-antibodies/choline-acetyltransferase-antibody-ab6168.html)

Anti-cTNT (CT3, deposited to the DSHB by Lin, J.J-C.): initially published in initially published in: Jin, J., Lin, J.-C., and Lin, J.J.-C. (1990). Troponin T isoform switching during heart development. Ann. NY Acad. Sci. 588, 393-396. CT3 recognizes the embryonic and adult cardiac isforms [PMID: 2358124]. CT3 cross-react with slow skeletal muscle TnT but doesn't recognize fast skeletal muscle TnT [PMID: 12732643]. Cited in >99 publications (https://dshb.biology.uiowa.edu/CT3)

Anti-ACTN2 (Sigma-Aldrich #A7811): according to the manufacturer, this monoclonal antibody is suitable for detecting human α-Actinin by immunofluorescence; cited in >970 publications (https://www.sigmaaldrich.com/DE/en/product/sigma/a7811).

Anti-POLR3A/RPC1 (Cell Signaling Technology, #12825): according to the manufacturer, this monoclonal antibody is produced by immunizing animals with a synthetic peptide corresponding to residues surrounding Val613 of human POLR3A protein and has been validated for use in ChIP-Seq (https://www.cellsignal.com/products/primary-antibodies/polr3a-d5y2d-rabbit-mab/12825). Additional validation we performed included Western blotting (single band at the expected MW of POLR3A/RPC1) and extensive characterization of ChIP-Seq datasets (presence of clearly defined strong peaks at Pol III target genes and absence of ChIP signal from other genomic regions).

Drosophila spike-in antibody (Active Motif, #61686): according to the manufacturer, the spike-in antibody recognizes a histone variant that is specific to the species of the Spike-in Chromatin (Drosophila); cited in >25 publications (https://www.activemotif.com/catalog/1091/chip-normalization)

Anti-H3K4me3 (Active Motif, #39159): according to the manufacturer, this Histone H3 trimethyl Lys4 (H3K4me3) polyclonal antibody was raised against a peptide including trimethyl-lysine 4 of histone H3 and its specificity was confirmed by dot bot analysis (https://www.activemotif.com/catalog/details/39159/histone-h3-trimethyl-lys4-antibody-pab); it has been validated for ChIP by modENCODE (https://compbio.med.harvard.edu/antibodies/antibodies/84).

Anti-H3K27me3 (Millipore, #07-449): according to the manufacturer, this polyclonal antibody is dot blot tested for trimethylated lysine 27 specificity and validated in immunoprecipitation (https://www.merckmillipore.com/DE/de/product/Anti-trimethyl-Histone-H3-Lys27-Antibody,MM_NF-07-449?ReferrerURL=https%3A%2F%2Fwww.google.com%2F#); it has been validated for ChIP-Seq by modENCODE (https://compbio.med.harvard.edu/antibodies/antibodies/57).

Anti-H3K9me3 (Cell Signaling, #13969): according to the manufacturer, this monoclonal antibody detects endogenous levels of histone H3 when tri-methylated on Lys9. It shows some cross-reactivity with histone H3 that is di-methylated on Lys9, but does not cross-react with non-methylated or mono-methylated histone H3 Lys9. This antibody does not detect tri-methyl histone H3 Lys9 when the adjacent Ser10 residue is phosphorylated during mitosis. In addition, this antibody does not cross-react with methylated histone H3 Lys4, Lys27, Lys36, or Lys79. This antibody has been validated using SimpleChIP® Enzymatic Chromatin IP Kits. (https://www.cellsignal.com/products/primary-antibodies/tri-methyl-histone-h3-lys9-d4w1u-rabbit-mab/13969); cited in >112 publications.

Anti-POLR3G/RPC7α (Santa Cruz, #sc21754): according to the manufacturer, this monoclonal antibody raised against recombinant human RPC32/POLR3G/RPC7α subunit of RNA polymerase III and cited in >6 publications (https://www.scbt.com/p/pol-iii-rpc32-antibody-c32-1). Additional validation we performed included Western blotting (single band at the expected MW of POLR3G and its substantial decrease upon POLR3G knockdown by CRISPRi) and extensive characterization of ChIP-Seq datasets (presence of clearly defined strong peaks at Pol III target genes and absence of ChIP signal from other genomic regions).

Anti-BRF1 (Abcam, #ab264191): according to the manufacturer, this polyclonal antibody was raised against a synthetic peptide within human BRF1 aa 627-677 (https://www.abcam.com/products/primary-antibodies/brf1-antibody-ab264191.html). Additional validation we performed included Western blotting (single band at the expected MW of BRF1 and its substantial decrease upon BRF1 knockdown by CRISPRi) and extensive characterization of ChIP-Seq datasets (presence of clearly defined strong peaks at Pol III target genes and absence of ChIP signal from other genomic regions, inlcuding RNU6-1, at which Pol III is assembled via BRF2).

Anti-MAF1 (Santa Cruz; #sc-515614 X): according to the manufacturer, this monolconal antibody specific for an epitope mapping between amino acids 99-122 within an internal region of MAF1 of human origin; cited in 3 publications (https://www.scbt.com/p/maf1-antibody-h-2). Additional validation we performed included Western blotting (single band or smear at the expected MW of MAF1 depending on phosphorylation status and its substantial decrease upon MAF1 knockdown by CRISPRi).

Anti-phospho-4E-BP1 (Cell Signaling, #2855T): according to the manufacturer, this monoclonal antibody detects endogenous levels of 4E-BP1 only when phosphorylated at Thr37 and/or Thr46. This antibody may cross-react with 4E-BP2 and 4E-BP3 when phosphorylated at equivalent sites; cited in >1680 publications (https://www.cellsignal.com/products/primary-antibodies/phospho-4e-bp1-thr37-46-236b4-rabbit-mab/2855). Additional validation we performed included Western blotting (several bands at the expected MW of phoshorylated 4E-BP1 and the disappearance of a subset of those upon mTORC1 inhibition by Torin 1 treatment).

Anti-phospho-p70 S6 Kinase (T389) (Cell Signaling, #9206S): according to the manufacturer, this monoclonal antibody detects endogenous levels of p70 S6 kinase only when phosphorylated at Thr389. This antibody also detects p85 S6 kinase when phosphorylated at the analogous site (Thr412) and possibly S6KII phosphorylated at Thr388; cited in >530 publications (https://www.cellsignal.com/products/primary-antibodies/phospho-p70-s6-kinase-thr389-1a5-mouse-mab/9206?site-search-type=Products&N=4294956287&Ntt=%239206s&fromPage=plp&_requestid=2582991). Additional validation we performed included Western blotting (bands at the expected MW of phoshorylated p70 and p85 S6 kinases and the disappearance upon mTORC1 inhibition by Torin 1 treatment).

Anti-4E-BP1 (Cell Signaling, #9644): according to the manufacturer, this monoclonal antibody detects endogenous levels of total 4E-BP1 protein from human origin; cited in > 1116 publications (https://www.cellsignal.com/products/primary-antibodies/4e-bp1-53h11-rabbit-mab/9644?site-search-type=Products&N=4294956287&Ntt=%239644%29&fromPage=plp&_requestid=2583398).

Anti-p70 S6 Kinase (Cell Signaling, #2708T): according to the manufacturer, this monoclonal antibody detects endogenous levels of total p70 S6 kinase protein. The antibody also recognizes p85 S6 kinase; cited in >1500 publications (https://www.cellsignal.com/products/primary-antibodies/p70-s6-kinase-49d7-rabbit-mab/2708?site-search-type=Products&N=4294956287&Ntt=%232708t%29%3A&fromPage=plp&_requestid=2583783).

Anti-vinculin (Cell Signaling; #13901): according to the manufacturer, this monoclonal antibody recognizes endogenous levels of total vinculin protein. This antibody also reacts with metavinculin, a 145 kDa splice variant of vinculin; cited in >390 publications (https://www.cellsignal.com/products/primary-antibodies/vinculin-e1e9v-xp-rabbit-mab/13901?site-search-type=Products&N=4294956287&Ntt=%2313901%29%3A&fromPage=plp&_requestid=2584007).

# Eukaryotic cell lines

Policy information about cell lines and Sex and Gender in Research

| | |
|---|---|
| Cell line source(s) | The hiPSC HPSI0214i-kucg_2 (male) and HPSI0214i-wibj_2 (female) cell lines were sourced from the HipSci Consortium (https://www.hipsci.org/) through the European Collection of Authenticated Cell Cultures (ECACC). HEK 293T/17 cells were obtained from ATCC (CRL-11268). Lenti-X™ 293T cells were obtained from Takara Bio (#632180). |
| Authentication | hiPSC, NPC, neurons, and cardiomyocytes were authenticated by the analysis of marker gene expression in RNA-seq datasets and the presence of the respective proteins by fluorescence microscopy. HEK 293T/17 and Lenti-X™ 293T cells were not authenticated. |
| Mycoplasma contamination | All cell lines used in this study were tested negative for mycoplasma contamination. |
| Commonly misidentified lines (See ICLAC register) | No commonly misidentified lines were used. |

# Plants

| | |
|---|---|
| Seed stocks | *Report on the source of all seed stocks or other plant material used. If applicable, state the seed stock centre and catalogue number. If plant specimens were collected from the field, describe the collection location, date and sampling procedures.* |
| Novel plant genotypes | *Describe the methods by which all novel plant genotypes were produced. This includes those generated by transgenic approaches, gene editing, chemical/radiation-based mutagenesis and hybridization. For transgenic lines, describe the transformation method, the number of independent lines analyzed and the generation upon which experiments were performed. For gene-edited lines, describe the editor used, the endogenous sequence targeted for editing, the targeting guide RNA sequence (if applicable) and how the editor was applied.* |
| Authentication | *Describe any authentication procedures for each seed stock used or novel genotype generated. Describe any experiments used to assess the effect of a mutation and, where applicable, how potential secondary effects (e.g. second site T-DNA insertions, mosiacism, off-target gene editing) were examined.* |

# ChIP-seq

## Data deposition

☒ Confirm that both raw and final processed data have been deposited in a public database such as GEO.

☒ Confirm that you have deposited or provided access to graph files (e.g. BED files) for the called peaks.

| | |
|---|---|
| Data access links<br>*May remain private before publication.* | https://www.ncbi.nlm.nih.gov/geo/query/acc.cgi?acc=GSE227928<br>(reviewer token: mdmzwugkhputxkz) |
| Files in database submission | Too many files to list, please see GEO accession. |
| Genome browser session<br>(e.g. UCSC) | NA |

## Methodology

| | |
|---|---|
| Replicates | All ChIP-seq experiments were performed on two biological replicates per cell type examined in this study, with the exception of H3K27me3 ChIP-seq experiments and H3K4me3 ChIP-Seq in NPC, which were performed as single replicates. |
| Sequencing depth | ChIP-seq library sequencing was performed on an Illumina NovaSeq platform, with 110bp paired-end reads. All libraries had > 30 million reads and 60-76% uniquely mapped reads per library. Full details below:<br><br>Sample Reads Uniquely mapped number Uniquely mapped %<br>WT_RPC1_k_hiPSC_rep1 80,515,257 54,829,218 68.10%<br>WT_RPC1_k_hiPSC_rep2 83,868,794 57,410,834 68.45%<br>WT_RPC1_w_hiPSC_rep1 56,632,856 37,844,794 66.82%<br>WT_RPC1_w_hiPSC_rep2 55,982,708 37,859,040 67.63%<br>WT_RPC1_k_NPC_rep1 101,693,829 71,746,094 70.55%<br>WT_RPC1_k_NPC_rep2 81,257,191 57,657,985 70.96%<br>WT_RPC1_k_neurons_rep1 98,243,649 65,871,592 67.05%<br>WT_RPC1_k_neurons_rep2 126,844,321 87,688,888 69.13%<br>WT_RPC1_k_CM_rep1 84,380,879 58,631,896 69.48%<br>WT_RPC1_k_CM_rep2 77,730,654 55,015,449 70.78%<br>WT_RPC1_HEK293T_rep1 75,085,959 51,484,833 68.57%<br>WT_RPC1_HEK293T_rep2 69,435,742 50,403,901 72.59%<br>WT_BRF1_k_hiPSC_rep1 69,651,303 42,859,623 61.53%<br>WT_BRF1_k_hiPSC_rep2 65,541,911 47,193,717 72.01%<br>WT_BRF1_k_NPC_rep1 62,636,853 43,866,556 70.03%<br>WT_BRF1_k_NPC_rep2 60,150,585 42,949,150 71.40%<br>WT_BRF1_k_neurons_rep1 61,790,148 43,711,076 70.74%<br>WT_BRF1_k_neurons_rep2 66,923,872 45,695,831 68.28%<br>WT_BRF1_k_CM_rep1 56,767,496 38,857,672 68.45%<br>WT_BRF1_k_CM_rep2 70,591,953 48,701,750 68.99%<br>WT_H3K4me3_k_hiPSC_rep1 79,476,847 54,218,854 68.22%<br>WT_H3K4me3_k_hiPSC_rep2 66,452,058 43,025,129 64.75%<br>WT_H3K4me3_k_NPC_rep1 70,304,132 50,679,818 72.09%<br>WT_H3K4me3_k_neurons_rep1 87,149,242 61,670,227 70.76%<br>WT_H3K4me3_k_neurons_rep2 62,893,308 44,984,001 71.52%<br>WT_H3K4me3_k_CM_rep1 84,122,956 61,107,086 72.64%<br>WT_H3K4me3_k_CM_rep2 92,281,152 66,120,972 71.65%<br>WT_H3K27me3_k_hiPSC_rep1 78,287,550 48,793,806 62.33%<br>WT_H3K27me3_k_NPC_rep1 128,691,861 94,942,665 73.78%<br>WT_H3K27me3_k_neurons_rep1 86,560,571 59,314,104 68.52%<br>WT_H3K27me3_k_CM_rep1 100,608,444 75,254,256 74.80%<br>WT_H3K9me3_k_hiPSC_rep1 74,537,018 54,404,208 72.99%<br>WT_H3K9me3_k_NPC_rep1 80,831,019 58,753,343 72.69%<br>WT_H3K9me3_k_neurons_rep1 75,643,034 53,672,033 70.95%<br>MAF1_ctrl_RPC1_k_hiPSC_rep1 75,815,702 51,798,493 68.32%<br>MAF1_ctrl_RPC1_k_hiPSC_rep2 72,318,018 50,548,859 69.90%<br>MAF1_KD_RPC1_k_hiPSC_rep1 72,099,050 52,361,187 72.62%<br>MAF1_KD_RPC1_k_hiPSC_rep2 86,229,444 60,336,854 69.97%<br>MAF1_ctrl_RPC1_k_NPC_rep1 86,098,574 58,337,615 67.76%<br>MAF1_ctrl_RPC1_k_NPC_rep2 31,700,275 22,666,980 71.50%<br>MAF1_KD_RPC1_k_NPC_rep1 70,440,589 47,449,928 67.36%<br>MAF1_KD_RPC1_k_NPC_rep2 77,231,700 54,075,167 70.02%<br>MAF1_ctrl_RPC1_k_NPCderived_rep1 80,812,139 58,324,605 72.17%<br>MAF1_ctrl_RPC1_k_NPCderived_rep2 111,879,883 78,916,068 70.54%<br>MAF1_KD_RPC1_k_NPCderived_rep1 96,711,579 66,940,061 69.22%<br>MAF1_KD_RPC1_k_NPCderived_rep2 142,919,671 100,032,182 69.99%<br>ProTGG_bodyEdit_RPC1_k_hiPSC_rep1 37,535,143 22,576,642 60.15%<br>ProTGG_bodyEdit_RPC1_k_hiPSC_rep2 26,027,666 16,797,930 64.54%<br>ProTGG_bodyEdit_RPC1_k_NPC_rep1 46,713,583 33,263,504 71.21% |

ProTGG_bodyEdit_RPC1_k_NPC_rep2 58,655,074 41,317,943 70.44%
ProTGG_upstreamEdit_RPC1_k_hiPSC_rep1 28,084,522 18,838,232 67.08%
ProTGG_upstreamEdit_RPC1_k_hiPSC_rep2 30,599,758 20,104,269 65.70%
ProTGG_upstreamEdit_RPC1_k_NPC_rep1 30,287,901 22,374,858 73.87%
ProTGG_upstreamEdit_RPC1_k_NPC_rep2 34,271,870 24,755,841 72.23%
ProTGG_bodyEdit_BRF1_k_hiPSC_rep1 81,161,914 60,397,498 74.42%
ProTGG_bodyEdit_BRF1_k_hiPSC_rep2 96,259,734 70,116,314 72.84%
ProTGG_bodyEdit_BRF1_k_NPC_rep1 49,297,297 37,661,883 76.40%
ProTGG_bodyEdit_BRF1_k_NPC_rep2 62,869,462 46,707,897 74.29%
ProTGG_upstreamEdit_BRF1_k_hiPSC_rep1 65,353,550 47,632,397 72.88%
ProTGG_upstreamEdit_BRF1_k_hiPSC_rep2 68,648,862 51,492,865 75.01%
ProTGG_upstreamEdit_BRF1_k_NPC_rep1 64,534,453 48,190,454 74.67%
ProTGG_upstreamEdit_BRF1_k_NPC_rep2 44,920,325 28,873,475 64.28%
WT_RPC7a_k_hiPSC_rep1 28,173,190 18,642,449 66.17%
WT_RPC7a_k_hiPSC_rep2 33,137,234 24,040,817 72.55%
POLR3G_ctrl_RPC1_k_hiPSC_rep1 58,112,790 39,861,878 68.59%
POLR3G_ctrl_RPC1_k_hiPSC_rep2 65,067,136 46,919,952 72.11%
POLR3G_KD_RPC1_k_hiPSC_rep1 62,249,136 43,208,968 69.41%
POLR3G_KD_RPC1_k_hiPSC_rep2 57,939,450 42,281,264 72.97%
POLR3G_ctrl_Rapa_RPC1_k_hiPSC_rep1 35,674,600 24,354,478 68.27%
POLR3G_ctrl_Rapa_RPC1_k_hiPSC_rep2 37,339,132 25,634,855 68.65%
POLR3G_KD_Rapa_RPC1_k_hiPSC_rep1 40,067,445 25,751,532 64.27%
POLR3G_KD_Rapa_RPC1_k_hiPSC_rep2 44,752,449 29,952,037 66.93%
w_hiPSC_Input_ChIP 74,182,549 54,018,380 72.82%
NPC_Input_ChIP 91,246,822 69,375,429 76.03%
CM_Input_ChIP 100,695,245 75,124,740 74.61%
HEK293T_Input_ChIP 72,935,763 54,048,987 74.10%

| Antibodies | Anti-POLR3A/RPC1 antibody (Cell Signaling Technology, #12825)<br>Anti-BRF1 (Abcam, #ab264191)<br>Drosophila spike-in antibody (Active Motif, #61686)<br>Anti-H3K4me3 (Active Motif, #39159)<br>Anti-H3K27me3 (Millipore, #07-449)<br>Anti-H3K9me3 (Cell Signaling, #13969)<br>Anti-POLR3G/RPC7α (Santa Cruz, #sc21754) |
|---|---|

**Peak calling parameters**

Reads were aligned to the GRCh38 reference genome using STAR with allowing up to one mismatch per read, a maximum of ten alignment positions, end-to-end alignment, prohibited introns, only one alignment reported per read.
Peak calling was performed on duplicate-filtered alignments using MACS callpeak v2.2.6, supplying ChIP input samples from HPSI0214i-kucg_2 for kucg-2 hiPSC and CM datasets, HPSI0214i-wibj_2 for wibj_2 hiPSC datasets, and from HPSI0214i-kucg_2-derived NPC for NPC and neuron datasets with the following parameters:
-g hs --slocal 500 -B --keep-dup all --nomodel --extsize {FRAG} --SPMR
where {FRAG} corresponds to the estimated fragment size from cross-strand correlation analysis. For H3K4me3, H3K27me3 and H3K9me3 the --broad parameter was additionally specified to call broad peaks for these chromatin marks.
Finally, peaks were filtered for blacklist regions as specified by the ENCODE GRCh38 blacklist file (https://www.encodeproject.org/files/ENCFF356LFX/).

**Data quality**

FastQC was run on all FASTQ files to assess general sequencing quality. Mapping stats were generated automatically by STAR and assessed. For all ChIP-seq samples except H3K27me3, narrow peaks were called with an FDR < 0.05. Called peaks were validated for RPC1, RPC7, and BRF1 by assessing overlaps with predicted Pol III targets, specifically tRNAs, revealing high congruency with previous reports, and near-complete overlap of tRNA peaks between biological replicates. Full details of number of peaks given below:

Sample Peaks
WT_RPC1_k_hiPSC_rep1 1,113
WT_RPC1_k_hiPSC_rep2 979
WT_RPC1_w_hiPSC_rep1 666
WT_RPC1_w_hiPSC_rep2 940
WT_RPC1_k_NPC_rep1 348
WT_RPC1_k_NPC_rep2 340
WT_RPC1_k_neurons_rep1 341
WT_RPC1_k_neurons_rep2 293
WT_RPC1_k_CM_rep1 537
WT_RPC1_k_CM_rep2 542
WT_RPC1_HEK293T_rep1 882
WT_RPC1_HEK293T_rep2 690
WT_BRF1_k_hiPSC_rep1 1,088
WT_BRF1_k_hiPSC_rep2 733
WT_BRF1_k_NPC_rep1 334
WT_BRF1_k_NPC_rep2 326
WT_BRF1_k_neurons_rep1 325
WT_BRF1_k_neurons_rep2 378
WT_BRF1_k_CM_rep1 457
WT_BRF1_k_CM_rep2 511
WT_H3K4me3_k_hiPSC_rep1 25,807
WT_H3K4me3_k_hiPSC_rep2 23,820
WT_H3K4me3_k_NPC_rep1 19,780

WT_H3K4me3_k_neurons_rep1 18,340
WT_H3K4me3_k_neurons_rep2 18,553
WT_H3K4me3_k_CM_rep1 25,575
WT_H3K4me3_k_CM_rep2 24,835
WT_H3K27me3_k_hiPSC_rep1 34,934
WT_H3K27me3_k_NPC_rep1 19,160
WT_H3K27me3_k_neurons_rep1 110,582
WT_H3K27me3_k_CM_rep1 76,439
WT_H3K9me3_k_hiPSC_rep1 455,378
WT_H3K9me3_k_neurons_rep1 342,506
WT_H3K9me3_k_NPC_rep1 377,453
MAF1_ctrl_RPC1_k_hiPSC_rep1 2,063
MAF1_ctrl_RPC1_k_hiPSC_rep2 1,761
MAF1_KD_RPC1_k_hiPSC_rep1 2,148
MAF1_KD_RPC1_k_hiPSC_rep2 2,635
MAF1_ctrl_RPC1_k_NPC_rep1 740
MAF1_ctrl_RPC1_k_NPC_rep2 414
MAF1_KD_RPC1_k_NPC_rep1 1,019
MAF1_KD_RPC1_k_NPC_rep2 884
MAF1_ctrl_RPC1_k_NPCderived_rep1 628
MAF1_ctrl_RPC1_k_NPCderived_rep2 677
MAF1_KD_RPC1_k_NPCderived_rep1 967
MAF1_KD_RPC1_k_NPCderived_rep2 839
ProTGG_bodyEdit_RPC1_k_hiPSC_rep1 1,338
ProTGG_bodyEdit_RPC1_k_hiPSC_rep2 2,180
ProTGG_bodyEdit_RPC1_k_NPC_rep1 282
ProTGG_bodyEdit_RPC1_k_NPC_rep2 356
ProTGG_upstreamEdit_RPC1_k_hiPSC_rep1 893
ProTGG_upstreamEdit_RPC1_k_hiPSC_rep2 999
ProTGG_upstreamEdit_RPC1_k_NPC_rep1 314
ProTGG_upstreamEdit_RPC1_k_NPC_rep2 383
ProTGG_bodyEdit_BRF1_k_hiPSC_rep1 722
ProTGG_bodyEdit_BRF1_k_hiPSC_rep2 681
ProTGG_bodyEdit_BRF1_k_NPC_rep1 253
ProTGG_bodyEdit_BRF1_k_NPC_rep2 123
ProTGG_upstreamEdit_BRF1_k_hiPSC_rep1 658
ProTGG_upstreamEdit_BRF1_k_hiPSC_rep2 742
ProTGG_upstreamEdit_BRF1_k_NPC_rep1 272
ProTGG_upstreamEdit_BRF1_k_NPC_rep2 102
WT_RPC7a_k_hiPSC_rep1 425
WT_RPC7a_k_hiPSC_rep2 175
POLR3G_ctrl_RPC1_k_hiPSC_rep1 906
POLR3G_ctrl_RPC1_k_hiPSC_rep2 737
POLR3G_KD_RPC1_k_hiPSC_rep1 725
POLR3G_KD_RPC1_k_hiPSC_rep2 672
POLR3G_ctrl_Rapa_RPC1_k_hiPSC_rep1 734
POLR3G_ctrl_Rapa_RPC1_k_hiPSC_rep2 842
POLR3G_KD_Rapa_RPC1_k_hiPSC_rep1 650
POLR3G_KD_Rapa_RPC1_k_hiPSC_rep2 719
w_hiPSC_Input_ChIP NA
NPC_Input_ChIP NA
CM_Input_ChIP NA
HEK293T_Input_ChIP NA

Software

Software used to analyze ChIP-seq data includes: STAR, MACS, Trim Galore, Picard Tools, mmquant, deepTools, bedtools, ComplexHeatmap, DiffBind and ChIPpeakAnno. Software versions stated above.

