## [Peer Review File · Nature Cell Biology]

Peer Review Information

Journal: Nature Cell Biology

Manuscript Title: Selective gene expression maintains human tRNA anticodon pools during differentiation

Corresponding author name(s): Professor Danny Nedialkova

Reviewer Comments & Decisions:

Decision Letter, initial version:

Dear Professor Nedialkova,

Your manuscript, "Selective gene expression maintains human tRNA anticodon pools during differentiation", has now been seen by 3 referees, who are experts in tRNA Biology (referee 1); protein translation (referee 2); and gene expression regulation (referee 3). As you will see from their comments (attached below) they find this work of potential interest, but have raised some concerns, which in our view would need to be addressed with suitable revisions before we can consider publication in Nature Cell Biology.

Nature Cell Biology editors discuss the referee reports in detail within the editorial team, including the chief editor, to identify key referee points that should be addressed with priority, and requests that are overruled as being beyond the scope of the current study. To guide the scope of the revisions, I have listed these points below. We are committed to providing a fair and constructive peer-review process, so please feel free to contact me if you would like to discuss any of the referee comments further.

In particular, it would be essential to:

A: Add BRFF1 ChIP-seq data to strengthen the 5'-flanking sequence swapping experiment, and H3K9me3 and/or DNA methylation data to explain why inactive tRNA genes are not expressed, whether tRNA gene expression depends on altered enhancer activity (Reviewer#3)

B: compare the dataset with the published datasets and discuss any differences (Reviewer#1 pts 2,4, Reviewer#2 pt1)

C: Perform separate analysis of codon usage/Ribo-seq data for cell type-/tissue-specific genes (separate from house keeping genes) (Reviewer#1 pt3)

D: All other referee concerns pertaining to strengthening existing data, providing controls, methodological details, clarifications and textual changes, [EDITOR DELETE AS APPROPRIATE] should also be addressed.

E: Finally please pay close attention to our guidelines on statistical and methodological reporting (listed below) as failure to do so may delay the reconsideration of the revised manuscript. In particular please provide:

- a Supplementary Figure including unprocessed images of all gels/blots in the form of a multi-page pdf file. Please ensure that blots/gels are labeled and the sections presented in the figures are clearly indicated.
- a Supplementary Table including all numerical source data in Excel format, with data for different figures provided as different sheets within a single Excel file. The file should include source data giving rise to graphical representations and statistical descriptions in the paper and for all instances where the figures present representative experiments of multiple independent repeats, the source data of all repeats should be provided.

We would be happy to consider a revised manuscript that would satisfactorily address these points, unless a similar paper is published elsewhere, or is accepted for publication in Nature Cell Biology in the meantime.

- ensure that it conforms to our format instructions and publication policies (see below and www.nature.com/nature/authors/).
- provide a point-by-point rebuttal to the full referee reports verbatim, as provided at the end of this letter.
- provide the completed Editorial Policy Checklist (found here <https://www.nature.com/authors/policies/Policy.pdf>), and Reporting Summary (found here <https://www.nature.com/authors/policies/ReportingSummary.pdf>). This is essential for reconsideration of the manuscript and these documents will be available to editors and referees in the event of peer review. For more information see <http://www.nature.com/authors/policies/availability.html> or contact me.

Nature Cell Biology is committed to improving transparency in authorship. As part of our efforts in this direction, we are now requesting that all authors identified as 'corresponding author' on published papers create and link their Open Researcher and Contributor Identifier (ORCID) with their account on the Manuscript Tracking System (MTS), prior to acceptance. ORCID helps the scientific community achieve unambiguous attribution of all scholarly contributions. You can create and link your ORCID from the home page of the MTS by clicking on 'Modify my Springer Nature account'. For more information please visit www.springernature.com/orcid.

[redacted]

We would like to receive a revised submission within six months. We would be happy to consider a revision even after this timeframe, however if the resubmission deadline is missed and the paper is eventually published, the submission date will be the date when the revised manuscript was received.

We hope that you will find our referees' comments, and editorial guidance helpful. Please do not hesitate to contact me if there is anything you would like to discuss.

Best wishes,

Sabrya Carim

Sabrya Carim, PhD
(she/her/hers)
Associate Editor, Nature Cell Biology
Nature Portfolio

Springer Nature
The Campus, 4 Crinan Street, London N1 9XW, UK
sabrya.carim@springernature.com
<https://orcid.org/0000-0001-9485-1938>

Reviewers' Comments:

Reviewer #1:

Remarks to the Author:

Mammalian tRNA expression and its regulation are long standing questions that have not been extensively addressed in part due to technical difficulties. Gao et al combine the recently developed mim-tRNA-seq for tRNA quantitation, RNA Pol3 ChiP-seq for RNAPIII occupation, ATAC-seq for chromatic accessibility, and mTOR phosphorylation of the RNAPIII inhibitor MAF1 to gain its understanding in human iPSC differentiation. They found that tRNA genes can be divided into three groups: house-keeping, repressed, and inactive. House-keeping genes are expressed before and after differentiation, inactive not expressed, and repressed are regulated by MAF1 phosphorylation. Cellular tRNA abundance is primarily determined by the level of PolIII transcription through multiple tRNA gene sequence features. The technical quality of the data is superb, and they validated a few results using CRISPR knock-in of PolIII promoter's internal and upstream of tRNA genes. The thorough design, execution, and multi-variant consideration for understanding the regulation of mammalian tRNA expression make this work an excellent model to follow for future studies in the field.

Comments:

1. Fig. 1, 2. Arg-UCU-4-1: this gene has been the flag ship case of tRNA isodecoder tissue specific

expression in the field. How much change does the expression of all other Arg-UCU isodecoders in neurons versus other cell types occur? Does the ribo-seq data show differences among the AGA codons in the hiPSC and differentiated cells?

2. Fig. 1: constant tRNA anticodon pool: numerous literature showed tRNA reprogramming for differentiation versus proliferation (e.g. Pilpel lab). These results should be compared with the findings of this work to reveal differences and commonalities.

3. Fig. 2, codon usage analysis: in many cases, most translation activity in a cell can be for house-keeping proteins, e.g. actin, ribosomal proteins etc. tRNA anticodon expression needs to balance the high translation level for these same protein genes in all cells versus cell type or tissue specific genes. These different groups of genes can also have different codon usages. The ribo-seq data should be analyzed separately for these groups. The low difference shown here may simply be dominated by the house keeping protein genes.

4. Fig. 4, "house-keeping" tRNA genes: previous work by other groups (e.g. Odom, Collier labs) also showed a common group of tRNA genes expressed among mammalian tissues. How does the result of this work compare to those in the literature? How common or different and possible reasons?

Reviewer #2:

Remarks to the Author:

Referee report to Manuscript "Selective gene expression maintains human RNA anticodon pools during differentiation." by Gao, Behrens et al., submitted to Nature Cell Biology.

A quantitative, mechanistic and functional understanding of how tRNA profiles are shaped in mammalian cells, across cell types and during differentiation, continues to be a crucial and challenging frontier in contemporary research in gene expression regulation. A major hindrance to progress has been the inaccessibility of the highly modified, structured and sequence-wise frequently near-identical tRNA species to the chemistry of conventional RNA-seq methodology. The authors' lab presented an important technical solution to these challenges a few years ago when reporting on the mim-tRNAseq technology and its associated data analysis pipeline (Molecular Cell 2021) – a major advance towards making tRNA profiling more accessible.

The current manuscript builds on the mim-tRNAseq method to assess how tRNA profiles evolve and are maintained across cellular differentiation. It further analyses the downstream effects of changed tRNA expression on translation, as well as upstream regulatory mechanisms responsible for the reshaping of the tRNA landscape. The questions that are addressed are thus all very fundamental in nature. A particular strength of the study is the setup and use of a clean, well-controlled cellular differentiation system, relying on hiPSCs that can be differentiated to isogenic NPCs and further to neurons, or to cardiac cells. The integration of the additional datasets (ChIP-Seq, ATAC-Seq, ribo-seq) is exemplary, too.

Altogether, the study presents a coherent model for how tRNA genes are broadly expressed in undifferentiated cells, with a considerable reduction in the expressed tRNA repertoires upon differentiation due to reduced mTORC1 activity, which in turn engenders dephosphorylation of Maf1, which will act to repress PolIII. Of note, the reduction in the tRNA repertoire is happening principally on the tRNA gene level, without significantly manifesting on the level of anticodon pools that remain

relatively stable due to the redundancy in tRNA genes.

This study is rigorously executed, the manuscript is well-written, and it reports on important new insights. I am therefore enthusiastic about this piece and have only few additional, minor comments and suggestions.

(1) Figure 2: The strong correlation between A-site abundance for CHX/TIG-treated samples and tRNA anticodon abundance is impressive. In CHX-treated extracts, this correlation is not observable anymore. Most labs using ribo-seq, and most published datasets, are CHX-treated rather than CHX/TIG-treated. Therefore, I think it could be very interesting and helpful for researchers in the field to understand better the general relationship between CHX vs. CHX/TIG data. With such knowledge, it could be possible to “transform” available CHX data in a way that would potentially allow predicting the abundance of cognate tRNA anticodon families in the biological samples the footprints came from. Based on the molecular mechanisms that CHX and TIG act on, one could naively think, for example, that the A-site correlation from CHX/TIG data will convert to a P-site correlation in CHX data. Is this the case in the authors’ datasets? Is there a cross-correlation between CHX and CHX/TIG monosome peaks that would allow formulating rules for the relationship between the datasets?

This would be more of a methodological addition to the manuscript and I certainly do not expect a full-fledged analysis as it would distract from the main thrust of the piece. However, it would be nice to explore and slightly expand on such questions, as they would be of great interest for the field.

(2) Figure 1 - minor comment to panel b: it could be nice to include negative controls in these images to demonstrate specificity of the signal (i.e., not only show presence of signal for the markers in the respective cell types where we expect the marker to be expressed, but also show panels with cells where the markers are not expressed).

(3) Page 5, lines 18-19: “Using this workflow, we obtained single-transcript resolution data for 373 of the 413 predicted nuclear-encoded tRNA transcripts...”: how does this number, 413, relate to the other number “432 unique tRNA transcripts” found on page 2, line 18? Should this be the same?

(4) Figure 3a and page 10 line 9: The figure panel shows 26 categories, but the results section mentions “between 1 and 25”. Please clarify.

(5) Page 12, line 11: For the sentence “Differences in Pol III occupancy thus explain nearly all of the variation in mature tRNA abundance in human cells, suggesting that post-transcriptional mechanisms contribute little to controlling the levels of individual tRNAs in physiological settings.” I would recommend removing the last three words “in physiological settings” because it invokes a general principle that is maybe too overstated given the singular differentiation system that is used in this study.

(6) Figure 5e, h: The system presented to evaluate the elements involved in tRNA expression is a great setup. However, the two effects – basal expression difference and inducibility – are difficult to discern the way the data is shown. It would (if we take the right panels in Figure 5e as an example) be good to show the quantified fold-change between NPC-wt and hiPSC-wt for Pro-TGG-2-1 and compare it to the FC for the edited version (I can see the latter is lower in basal state, but the precise FC numbers are difficult to estimate from the graph). Moreover, I am wondering if it would be nice to have another panel next to the existing ones that shows the quantification of endogenous Pro-TGG-1-

1. This would give the possibility to compare the response for the endogenous/real gene vs. the edited copy. In the end, this could help to give the reader a better idea of what is actually most strongly affected by the edits: basal expression or inducibility? Can the ensemble of intragenic and upstream sequence features explain the full effect?

(7) Page 19, line 5 (from bottom): "AIM2 [...] and CADM3 [...], both of which are conserved in vertebrates." – it should be stated more clearly if this means that the position/constellation is conserved or if the genes themselves are conserved genes.

(8) Page 20/21, in the section on RPC7a loss, I was wondering if it would be better for the flow of the presentation to move up the last part (Extended Data Fig. 6, f-g) and present it first, before the other data in this section?

(9) Discussion, Page 25: I am not quite convinced by the "energy expenditure" argument. Why would it be much more costly to have more diverse repertoires of tRNAs as compared to a more restricted housekeeping set (which, in compensation, would also need to be expressed at higher levels).

Reviewer #3:

Remarks to the Author:

Gao et al. submitted a manuscript describing RNA polymerase (Pol) III-driven expression patterns of human tRNAs during cell differentiation. The authors used human induced pluripotent stem cells for differentiation into isogenic cardiomyocytes and neuronal cells. Using their recently published mim-tRNAseq technology, they determined genome-wide tRNA expression before, during, and after differentiation. They found that a subset of tRNAs is expressed in all cell types, subsequently termed "housekeeping", a subset of genes is "inactive", and another is "repressed", representing tRNA genes expressed in iPSCs but downregulated in at least one of the differentiated cell types. ChIP-seq with anti-RPC1 antibodies showed that tRNA transcription is well reflected by the presence of Pol III. Consequently, only housekeeping genes of Pol III were recognized after differentiation. In neurons, a particular case of tRNA expression was reported, namely the tRNA-Arg-TCT-4-1 gene located near the neuron-specific CADM3 gene. The expression of this tRNA gene was dramatically upregulated in neurons. Although strong changes in tRNA expression were observed during differentiation, the overall abundance of anticodon families remained generally stable. Nevertheless, individual cell types could be determined based on tRNA expression patterns. Codon usage and translation speed did not change with altered tRNA expression. Actively transcribed tRNA genes were nucleosome-free and surrounded by trimethylated H3K4-containing nucleosomes (NFRs) in all cell types before and after differentiation. Repressed genes lost H3K4me3 and NFRs after differentiation but gained H3K27me3 instead, the latter being absent at housekeeping tRNA genes. Inactive tRNA genes were not associated with either H3K4me3 or H3K27me3. Moreover, they were embedded in nucleosome-containing chromatin, as expected. Expression of tRNA genes correlated with A- and B-box consensus sequences. In addition, sequences upstream of the transcription start site influenced differentiation-dependent expression patterns. Differentiation-dependent downregulation of embryonic stem cell-specific POLR3G subunit expression of Pol III did not explain the changes in tRNA transcription. Finally, the authors showed that TORC1-regulated MAF1 activity correlated well with the expression of the repressed tRNAs.

The data presented in this manuscript are important for understanding a possible influence of tRNA transcription on cell differentiation. The cellular system is well chosen and the data are convincing.

The authors did not find any obvious influences of changes in tRNA transcription on differentiation, as anticodon family expression remains stable. However, the notion that different cell types can be identified by their respective tRNA pools suggests that additional mechanisms, such as tRNA modification, may be required to generate tRNA pools adapted to cell type-specific translation requirements. The manuscript presented here contains a wealth of data that will be of great importance for future studies of this type. Therefore, I recommend publishing these data after considering some minor changes.

Page 14, line 7: The authors state that RPC1 is not bound to one third of the predicted tRNAs in immortalized cell lines. This should be toned down to HEK cells, which is the only immortalized cell line analyzed here.

Page 16, line 2: Although most tRNA genes are either intragenic or near coding genes..... I wonder if the assumption that most tRNA genes are intragenic is correct? The authors are certainly right about the phrase "or near coding genes," especially since "near" was not defined, but the wording of the sentence is misleading.

The tRNA and 5'-flanking sequence swapping experiments would be strengthened by BRF1 ChIP results. Moreover, the combined swapping of the corresponding tRNA genes with their 5'-flanking sequences could reveal whether they act independently or synergistically.

On page 20, Gao et al. discuss the possibility that tRNA expression is regulated simply by being in proximity to enhancers. For the tRNA Arg-4-1 gene, they show data that support this theory. The authors should also mention whether the expression of the 37 housekeeping and 14 repressed tRNA genes also depends on enhancer activities. In particular, does regulation of the repressed tRNA genes coincide with altered enhancer activity?

The paragraph on "global restructuring of tRNA genes upon differentiation" would benefit from the inclusion of H3K9me3 data and/or DNA methylation data. Without such data, it remains unexplained why "inactive tRNA genes" are not transcribed.

METHODS – Nature Cell Biology publishes methods online. The methods section should be provided as a separate Word document, which will be copyedited and appended to the manuscript PDF, and

incorporated within the HTML format of the paper.

Methods should be written concisely, but should contain all elements necessary to allow interpretation and replication of the results. As a guideline, Methods sections typically do not exceed 3,000 words. The Methods should be divided into subsections listing reagents and techniques. When citing previous methods, accurate references should be provided and any alterations should be noted. Information must be provided about: antibody dilutions, company names, catalogue numbers and clone numbers for monoclonal antibodies; sequences of RNAi and cDNA probes/primers or company names and catalogue numbers if reagents are commercial; cell line names, sources and information on cell line identity and authentication. Animal studies and experiments involving human subjects must be reported in detail, identifying the committees approving the protocols. For studies involving human subjects/samples, a statement must be included confirming that informed consent was obtained. Statistical analyses and information on the reproducibility of experimental results should be provided in a section titled "Statistics and Reproducibility".

All Nature Cell Biology manuscripts submitted on or after March 21 2016 must include a Data availability statement at the end of the Methods section. For Springer Nature policies on data availability see <http://www.nature.com/authors/policies/availability.html>; for more information on this particular policy see <http://www.nature.com/authors/policies/data/data-availability-statements-data-citations.pdf>. The Data availability statement should include:

- Accession codes for primary datasets (generated during the study under consideration and designated as "primary accessions") and secondary datasets (published datasets reanalysed during the study under consideration, designated as "referenced accessions"). For primary accessions data should be made public to coincide with publication of the manuscript. A list of data types for which submission to community-endorsed public repositories is mandated (including sequence, structure, microarray, deep sequencing data) can be found here <http://www.nature.com/authors/policies/availability.html#data>.
- Unique identifiers (accession codes, DOIs or other unique persistent identifier) and hyperlinks for datasets deposited in an approved repository, but for which data deposition is not mandated (see here for details <http://www.nature.com/sdata/data-policies/repositories>).
- At a minimum, please include a statement confirming that all relevant data are available from the authors, and/or are included with the manuscript (e.g. as source data or supplementary information), listing which data are included (e.g. by figure panels and data types) and mentioning any restrictions on availability.
- If a dataset has a Digital Object Identifier (DOI) as its unique identifier, we strongly encourage including this in the Reference list and citing the dataset in the Methods.

We recommend that you upload the step-by-step protocols used in this manuscript to the Protocol Exchange. More details can be found at www.nature.com/protocolexchange/about.

All imaging data should be accompanied by scale bars, which should be defined in the legend. Cropped images of gels/blots are acceptable, but need to be accompanied by size markers, and to retain visible background signal within the linear range (i.e. should not be saturated). The boundaries of panels with low background have to be demarked with black lines. Splicing of panels should only be considered if unavoidable, and must be clearly marked on the figure, and noted in the legend with a statement on whether the samples were obtained and processed simultaneously. Quantitative comparisons between samples on different gels/blots are discouraged; if this is unavoidable, it should only be performed for samples derived from the same experiment with gels/blots were processed in parallel, which needs to be stated in the legend.

Regardless of format, all figures must be vector graphic compatible files, not supplied in a flattened

raster/bitmap graphics format, but should be fully editable, allowing us to highlight/copy/paste all text and move individual parts of the figures (i.e. arrows, lines, x and y axes, graphs, tick marks, scale bars etc.). The only parts of the figure that should be in pixel raster/bitmap format are photographic images or 3D rendered graphics/complex technical illustrations.

The total number of Supplementary Figures (not including the “unprocessed scans” Supplementary Figure) should not exceed the number of main display items (figures and/or tables (see our Guide to Authors and March 2012 editorial <http://www.nature.com/ncb/authors/submit/index.html#suppinfo>; <http://www.nature.com/ncb/journal/v14/n3/index.html#ed>). No restrictions apply to Supplementary Tables or Videos, but we advise authors to be selective in including supplemental data.

Each Supplementary Figure should be provided as a single page and as an individual file in one of our

accepted figure formats and should be presented according to our figure guidelines (see above). Supplementary Tables should be provided as individual Excel files. Supplementary Videos should be provided as .avi or .mov files up to 50 MB in size. Supplementary Figures, Tables and Videos must be accompanied by a separate Word document including titles and legends.

GUIDELINES FOR EXPERIMENTAL AND STATISTICAL REPORTING

REPORTING REQUIREMENTS – To improve the quality of methods and statistics reporting in our papers we have recently revised the reporting checklist we introduced in 2013. We are now asking all life sciences authors to complete two items: an Editorial Policy Checklist (found here <https://www.nature.com/authors/policies/Policy.pdf>) that verifies compliance with all required editorial policies and a reporting summary (found here <https://www.nature.com/authors/policies/ReportingSummary.pdf>) that collects information on experimental design and reagents. These documents are available to referees to aid the evaluation of the manuscript. Please note that these forms are dynamic 'smart pdfs' and must therefore be downloaded and completed in Adobe Reader. We will then flatten them for ease of use by the reviewers. If you would like to reference the guidance text as you complete the template, please access these flattened versions at <http://www.nature.com/authors/policies/availability.html>.

Author Rebuttal to Initial comments**Point-by-Point Response**

We thank the editors and reviewers for their time and efforts on our manuscript and for their positive reviews and constructive feedback. We have added new data and clarifications that address all the points raised below. **The relevant sections of the manuscript are highlighted in yellow.**

Reviewer #1:

Mammalian tRNA expression and its regulation are long standing questions that have not been extensively addressed in part due to technical difficulties. Gao et al combine the recently developed mim-tRNA-seq for tRNA quantitation, RNA Pol3 ChIP-seq for RNAPIII occupation, ATAC-seq for chromatic accessibility, and mTOR phosphorylation of the RNAPIII inhibitor MAF1 to gain its understanding in human iPSC differentiation. They found that tRNA genes can be divided into three groups: house-keeping, repressed, and inactive. House-keeping genes are expressed before and after differentiation, inactive not expressed, and repressed are regulated by MAF1 phosphorylation. Cellular tRNA abundance is primarily determined by the level of PolIII transcription through multiple tRNA gene sequence features. The technical quality of the data is superb, and they validated a few results using CRISPR knock-in of PolIII promoter's internal and upstream of tRNA genes. The thorough design, execution, and multi-variant consideration for understanding the regulation of mammalian tRNA expression make this work an excellent model to follow for future studies in the field.

We thank the reviewer for the positive feedback and appreciation of our work. Comments:

1. Fig. 1, 2. Arg-UCU-4-1: this gene has been the flag ship case of tRNA isodecoder tissue specific expression in the field. How much change does the expression of all other Arg-UCU isodecoders in neurons versus other cell types occur? Does the ribo-seq data show differences among the AGA codons in the hiPSC and differentiated cells?

Thank you for this comment; we realize that our description of tRNA-Arg-UCU regulation in Fig. 1 lacked sufficient detail. We have now added **new figure panels** to clarify that the levels of tRNA-Arg-UCU or the expression-weighted usage of its matching AGA codon do not increase in neurons (page 8 in the revised manuscript):

*“Notably, despite the strong and specific upregulation of tRNA-Arg-UCU-4-1, which comprises ~40% of mature tRNA-Arg-UCU in neurons, the abundance of this anticodon family decreased by ~1.4-fold in this cell type because of the downregulation of other isodecoders (**Extended data***

Fig. 1, k-l: Supplementary Table 1).”

“Of note, despite the upregulation of the *tRNA-Arg-UCU-4-1* isodecoder, the expression-weighted usage of its matching AGA codon did not differ between neurons and other cell types (Fig. 2a).”

Since its discovery by Susan Ackerman’s lab as a CNS-enriched transcript, *tRNA-Arg-UCU-4-1* has indeed been a prominent example for the potential of individual tRNA genes to exhibit strong tissue-specific expression. This discovery was based on the identification of an intronic mutation in mouse *tRNA-Arg-TCT-4-1* that impaired tRNA processing, which led to a decrease in the levels of mature *tRNA-Arg-UCU* in the CNS and to neurodegeneration when the rescue of ribosomes stalled at AGA codons is impaired by mutations in *GTPBP2* (Ishimura et al., 2014 PMID: 25061210). The mechanistic basis for the strong tissue-specific expression of *tRNA-Arg-UCU-4-1*, however, has remained unclear, as its A- and B-box sequences are identical to those of other isodecoders. Careful subsequent work by the Ackerman lab showed that the phenotypes caused by the *tRNA-Arg-UCU-4-1* processing defect can be rescued by overexpression of other *tRNA-Arg-UCU* isodecoders (Kapur et al., 2020 PMID: 32853550), indicating that these phenotypes stem from depletion of the *tRNA-Arg-UCU* anticodon pool and not from isodecoder-specific functions of *tRNA-Arg-UCU-4-1*.

We now classify human *tRNA-Arg-TCT-4-1* as a housekeeping tRNA gene in our datasets based on the presence of a significant RPC1 peak in consensus sets for all cell types we tested (**Supplementary Table 2**). We show that its increased Pol III occupancy in neurons goes against the general trend of decreased expression for all other tRNA genes (**Fig. 4f**). Our identification of a neuron-specific enhancer overlapping with *tRNA-Arg-TCT-4-1* now provides a mechanism for its increased expression in the CNS through enhancer activation-mediated potentiation of Pol III transcription. Our data suggests that this gene represents a rare case of strong tissue selectivity, but enhancer-based regulation may occur for other tRNA genes in cell contexts beyond the ones we profiled here (see also response to Reviewer #3 below).

2. Fig. 1: constant tRNA anticodon pool: numerous literature showed tRNA reprogramming for differentiation versus proliferation (e.g. Pilpel lab). These results should be compared with the findings of this work to reveal differences and commonalities.

Thank you for this suggestion; we have now compared the changes in tRNA anticodon levels we observe to the proliferation-linked and differentiation-linked human tRNA anticodon families identified by the Pilpel lab in Gingold et al., 2014 PMID: 25215487 (new labels in **Fig. 1h** and text on page 8):

“We did not observe a clear separation of anticodon pools between proliferating (hiPSC and NPC) and non-dividing cells (neurons and CM) or a substantial overlap between the tRNA anticodon families that changed significantly in abundance in NPC, neuron, and CM cultures with differentiation-linked (“D”) or proliferation-linked (“P”) tRNAs identified in previous work³⁰ (Fig. 1h, Supplementary Table 1).”

We note that the tRNA anticodon classification in Gingold et al. was not based on their expression pattern in different cell types, but on tRNA anticodons corresponding to codons over-represented in cell cycle-related genes (termed “proliferation-linked tRNAs”) or codons overrepresented in cell differentiation-related genes (termed “differentiation-linked tRNAs”). The tRNA quantification was also performed with tRNA microarrays containing probes for only 294 out of the 619 predicted human tRNA genes. We unfortunately could not perform a direct quantitative comparison with our data, since there are no files associated with the Gingold et al. publication that show the fold-change of tRNA anticodons across cell types. To our knowledge, other studies that examine tRNA pools in proliferating vs differentiated cells were performed in mouse, which also precludes a direct comparison with our data due to the limited sequence conservation between mouse and human tRNA genes (see response to point 4).

3. Fig. 2, codon usage analysis: in many cases, most translation activity in a cell can be for house-keeping proteins, e.g. actin, ribosomal proteins etc. tRNA anticodon expression needs to balance the high translation level for these same protein genes in all cells versus cell type or tissue specific genes. These different groups of genes can also have different codon usages. The ribo-seq data should be analyzed separately for these groups. The low difference shown here may simply be dominated by the house keeping protein genes.

To address this comment, we have now performed new analysis that further stratifies highly expressed mRNAs (top 5% based on TPM values in RNA-Seq) into “shared” and “cell type-specific” sets, and compared the weighted codon usage of each set to tRNA anticodon abundance in the corresponding cell type (page 8 of the revised manuscript) and A-site codon dwell times calculated separately for “shared” and “hiPSC-specific” or “NPC-specific” mRNAs (page 10 of the revised manuscript):

“Interestingly, the codon usage of mRNAs that are highly abundant in all four cell types correlated significantly more strongly to tRNA anticodon levels than the codon usage in highly expressed cell type-specific mRNAs (Extended Data Fig. 2, b-c).”

“We next calculated A-site codon dwell times from mRNAs that are highly expressed in both hiPSC and NPC (shared, n=393) or are cell type-specific (n=114 in hiPSC and n=80 in NPC, Extended Data Fig. 2b). While ~20% of ribosome footprints originated from shared mRNAs, less than 3% mapped to the cell type-specific transcripts, resulting in codon dwell time estimates that were much less concordant between biological replicates (Pearson’s $r \sim 0.95$ for all and shared

mRNAs vs ~0.8 for cell type-specific mRNAs, Extended Data Fig. 2j). Accordingly, A-site codon dwell times for shared mRNAs were more highly correlated between NPC and hiPSC than dwell times for cell-type specific mRNAs (Pearson's $r=0.87$ vs 0.54), but the higher variance of the latter was not accounted for by differences in cognate tRNA levels (Extended Data Fig. 2, k-m). Taken together, these data indicate that the divergence of tRNA anticodon pools between NPC and hiPSC is not sufficient to substantially alter decoding speed."

We have been cautious when interpreting these data since the Odom and Kutter labs have shown that while variations in codon usage do exist among distinct sets of mammalian genes, they are largely driven by underlying GC content and mutational bias, and therefore likely result from evolutionary forces unrelated to codon usage (Rudolph, Schmidt et al., PMID: 27166679). We also note that the ~10-fold lower number of ribosome-protected mRNA footprints that originate from cell type-specific highly expressed transcripts as compared to shared highly expressed mRNAs decrease the correlation of A-site codon dwell time estimates in biological replicates of the same cell type (Extended Data Fig. 2j). This is in accordance with the dependence of A-site prediction accuracy on high sequence coverage documented by the developers of Scikit-ribo (Fig. 6B in Fang et al., 2018 PMID: 29361467). Thus, in our view, comparing A-site codon dwell times calculated from all ribosome-protected mRNA footprints affords the highest accuracy while also being appropriate for gauging differences that may be driven by altered tRNA anticodon availability, which we wanted to examine here. This is because translating ribosomes sample the cellular pool of EF1-GTP-aa-tRNA ternary complexes in an mRNA-independent fashion during decoding (reviewed in Rodnina et al., 2017 PMID: 28138068). The probability that they encounter a tRNA with a matching anticodon is thus a function of the overall "demand" for this particular tRNA by all other ribosomes in the cell, the majority of which are likely translating abundant housekeeping mRNAs.

4. Fig. 4, "house-keeping" tRNA genes: previous work by other groups (e.g. Odom, Collier labs) also showed a common group of tRNA genes expressed among mammalian tissues. How does the result of this work compare to those in the literature? How common or different and possible reasons?

We appreciate the opportunity to clarify how our work compares to studies by the Collier and Odom labs, which we cited in our original manuscript as prior work consistent with our findings of stable anticodon pools across cell types. By examining Pol III binding to tRNA genes in liver and brain tissues at eight mouse developmental stages, the Kutter and Odom labs identified a set of tRNA genes whose expression changed during development and a set that didn't, but found no sequence features or regulatory mechanisms to account for this distinction (Schmidt et al., 2014 PMID: 25122613). This study suggested that tRNA anticodon pools may be stable across tissues, but solely based on Pol III occupancy, as mature tRNA abundance was not measured due to the lack of reliable methods at the time. In Pinkard et al., 2020 (PMID: 32796835), the Collier lab profiled mature tRNAs across seven mouse tissues and found that tRNA anticodon levels are

similar despite variations in isodecoder abundance, but Pol III ChIP-Seq was not performed to identify the underlying tRNA gene regulatory mechanisms. Unfortunately, it is not feasible to directly compare our human data to these prior studies done in mouse because of the extremely limited conservation of mammalian tRNA genes. Using Pol III ChIP-Seq, Kutter et al. 2011 (PMID: 21873999) found that only 79 tRNA genes were commonly expressed in liver samples from six mammalian species (mouse, rat, human, macaque, dog and opossum); only 24 genes bound by Pol III were in locations of conserved synteny and therefore likely present in the early mammalian ancestor (one of these is the CNS-specific *tRNA-Arg-TCT-4-1*). This study also found no human homologs for a third of mouse tRNA genes, and even in closely related species such as mouse and rat, more than half of Pol III-occupied tRNA genes were unique to each species. Accordingly, the Genomic tRNA Database (GtRNAdb) lists only 87 human tRNA transcripts with mouse counterparts. In our view, this substantial tRNA gene divergence among mammals underscores the critical importance of studying tRNA regulation directly in human cells.

Reviewer #2:

Referee report to Manuscript “Selective gene expression maintains human RNA anticodon pools during differentiation.” by Gao, Behrens et al., submitted to Nature Cell Biology.

A quantitative, mechanistic and functional understanding of how tRNA profiles are shaped in mammalian cells, across cell types and during differentiation, continues to be a crucial and challenging frontier in contemporary research in gene expression regulation. A major hindrance to progress has been the inaccessibility of the highly modified, structured and sequence-wise frequently near-identical tRNA species to the chemistry of conventional RNA-seq methodology. The authors’ lab presented an important technical solution to these challenges a few years ago when reporting on the mim-tRNAseq technology and its associated data analysis pipeline (Molecular Cell 2021) – a major advance towards making tRNA profiling more accessible. The current manuscript builds on the mim-tRNAseq method to assess how tRNA profiles evolve and are maintained across cellular differentiation. It further analyses the downstream effects of changed tRNA expression on translation, as well as upstream regulatory mechanisms responsible for the reshaping of the tRNA landscape. The questions that are addressed are thus all very fundamental in nature. A particular strength of the study is the setup and use of a clean, well-controlled cellular differentiation system, relying on hiPSCs that can be differentiated to isogenic NPCs and further to neurons, or to cardiac cells. The integration of the additional datasets (ChIP-Seq, ATAC-Seq, ribo-seq) is exemplary, too.

Altogether, the study presents a coherent model for how tRNA genes are broadly expressed in undifferentiated cells, with a considerable reduction in the expressed tRNA repertoires upon differentiation due to reduced mTORC1 activity, which in turn engenders dephosphorylation of Maf1, which will act to repress PolIII. Of note, the reduction in the tRNA repertoire is happening

principally on the tRNA gene level, without significantly manifesting on the level of anticodon pools that remain relatively stable due to the redundancy in tRNA genes.

This study is rigorously executed, the manuscript is well-written, and it reports on important new insights. I am therefore enthusiastic about this piece and have only few additional, minor comments and suggestions.

We thank the reviewer for their enthusiasm and appreciation of our work.

(1) Figure 2: The strong correlation between A-site abundance for CHX/TIG-treated samples and tRNA anticodon abundance is impressive. In CHX-treated extracts, this correlation is not observable anymore. Most labs using ribo-seq, and most published datasets, are CHX-treated rather than CHX/TIG-treated. Therefore, I think it could be very interesting and helpful for researchers in the field to understand better the general relationship between CHX vs. CHX/TIG data. With such knowledge, it could be possible to “transform” available CHX data in a way that would potentially allow predicting the abundance of cognate tRNA anticodon families in the biological samples the footprints came from. Based on the molecular mechanisms that CHX and TIG act on, one could naively think, for example, that the A-site correlation from CHX/TIG data will convert to a P-site correlation in CHX data. Is this the case in the authors’ datasets? Is there a cross-correlation between CHX and CHX/TIG monosome peaks that would allow formulating rules for the relationship between the datasets?

This would be more of a methodological addition to the manuscript and I certainly do not expect a full-fledged analysis as it would distract from the main thrust of the piece. However, it would be nice to explore and slightly expand on such questions, as they would be of great interest for the field.

We fully agree that it would be beneficial to understand the impact of different translation inhibitors on the measurements of ribosome dwell times at distinct codons in mammalian cells by ribosome profiling. We have now performed additional analyses of P-site codon dwell times in short and long footprints from extracts supplemented with CHX only, or with CHX and TIG (page 10 of the revised manuscript):

“A-site (but not P-site) codon dwell times were strongly and significantly anti-correlated with the abundance of cognate tRNA anticodon families in both hiPSC and NPC (Fig. 2, b-c; Extended Data Fig. 2e), demonstrating the key role of tRNA anticodon availability for decoding rates in human cells.”

“The use of TIG is critical for capturing footprints from decoding ribosomes, since the correlation between A-site codon dwell times and tRNA availability was much more modest for short (Pearson’s $r=0.32$) and not significant for long footprints, or for P-sites of footprints from hiPSC

extracts supplemented only with CHX (Extended Data Fig. 2, f-h)."

While we unfortunately cannot expand on this topic further in the manuscript due to space limitations and narrative coherence, we take the opportunity to do so here and hope the reviewer will find it useful. It is generally accepted in the field that distinct translation inhibitors trap elongating ribosomes in discrete states, which rarely impacts gene-level measurements but can yield footprints of different length and profoundly alter codon-level measurements of translation efficiency (Lareau et al., 2014 PMID: 24842990). Most prior work examining the underlying mechanisms has focused on yeast - not only because of the greater ease in obtaining enough material for library construction, but due to the good correspondence between the tRNA adaptation index (tAI) calculated from tRNA gene copy numbers (dos Reis et al., 2004 PMID: 15448185) to mature tRNA abundance in this organism (Behrens et al., 2021 PMID: 33581077). The Green lab recently demonstrated that the use of a CHX/TIG cocktail in cell lysates substantially increases the correlation between codon dwell times and tAI in yeast (Wu et al., 2019 PMID: 30686592). They were unable to directly test the effects of CHX and TIG on this correlation in human cells due to the poor correspondence between tRNA abundance and gene copy number in human (Behrens et al., 2021 PMID: 33581077), and the lack of high-resolution methods for accurately measuring tRNA levels at the time. The reasons why the use of CHX on its own is less effective in capturing decoding ribosomes (with an empty A site) are probably complex and sample-dependent. The drug binds the E site of eukaryotic ribosomes (Schneider-Poetsch et al., 2010 PMID: 20118940) with a relatively high dissociation constant (1- 4 μ M, Pellegrino et al., 2019 PMID: 30759226). It is therefore conceivable that in the absence of drugs like TIG that stably bind to and occlude the ribosomal A site (Jenner et al., 2013 PMID: 23431179), EF1-GTP-aa-tRNA ternary complexes could diffuse into this site during nuclease digestion, particularly in lysates that are highly concentrated, leading to continued elongation but with kinetics distinct from those in living cells. While it is often presumed that elongation does not occur in lysates, careful work from by Green and Buskirk has demonstrated that "ribosome profiling lysates (from bacterial cells) synthesize proteins robustly in the absence of any added translational inhibitors" (Mohammad et al., 2019 PMID: 30724162). CHX binding to ribosomes is also not irreversible, and ongoing elongation in yeast cells treated with CHX is well-documented (Hussmann et al., 2015 PMID: 26656907). We therefore believe that the answer to the question of whether it is possible to "transform" available CHX data in a way that would allow predicting correlations with tRNA abundance is likely to be complex and highly dependent on the biological samples from which the data originate. We note that for our datasets from hiPSC, the correlation between A-site codon dwell times and tRNA availability was more modest but still significant for short footprints (Pearson's $r=0.32$) and was weakly positive for P-site codon dwell times from long footprints (Pearson's $r=0.24$, $p=0.068$) in libraries prepared from lysates supplemented only with CHX.

(2) Figure 1 - minor comment to panel b: it could be nice to include negative controls in these images to demonstrate specificity of the signal (i.e., not only show presence of signal for the markers in the respective cell types where we expect the marker to be expressed, but also show

panels with cells where the markers are not expressed).

We thank the reviewer for the suggestion. We have now performed immunostainings in cells where the expression of marker genes is not detectable or substantially lower (MAP2 in hiPSC) based on RNA-Seq. These **new data** are included in **Extended Fig. 1b**. We note that the background signal in cells where the marker genes are weakly or not expressed is not uniform and is distinct from the specific subcellular localization of the marker proteins in cell types where they are highly expressed (**Fig. 1b**).

(3) Page 5, lines 18-19: “Using this workflow, we obtained single-transcript resolution data for 373 of the 413 predicted nuclear-encoded tRNA transcripts...”: how does this number, 413, relate to the other number “432 unique tRNA transcripts” found on page 2, line 18? Should this be the same?

We realize that the curated tRNA reference we used in the mim-tRNAseq pipeline was not described in the original manuscript. We now provide **additional details** in the *Methods* section on page 50 of the revised manuscript:

“Briefly, the full set of 619 predicted tRNA genes for the hg38 human genome assembly were downloaded from GtRNAb¹¹⁹ and the 22 mitochondrially encoded human tRNA genes were fetched from mitotRNAb¹²⁰. After intron removal and addition of 5'-G (for tRNA-His) and 3' CCA (for nuclear-encoded transcripts), a curated set of 599 nuclear-encoded tRNA sequences (excluding tRNAs with non-canonical secondary structure alignments or undetermined anticodons) and 22 mitochondrially encoded tRNA sequences was compiled as an alignment reference (--species Hsap). Reads were aligned to this reference...”

The 20 predicted tRNA genes we excluded from the curated reference encode for 19 transcripts that would deviate substantially from the canonical secondary structure of mature tRNAs, or have “undetermined” (NNN) anticodon sequences. This precludes their analysis with the mim-tRNAseq computational pipeline (and probably all others), and they are most likely pseudogenes that have retained some tRNA-like features that enable their identification by tRNAScan-SE. Our curated reference therefore contains 413 of the 432 predicted human tRNA transcripts.

(4) Figure 3a and page 10 line 9: The figure panel shows 26 categories, but the results section mentions “between 1 and 25”. Please clarify.

Thank you for pointing this out. The number of tRNA isodecoders does indeed extend to 26 for one anticodon family as displayed in the figure. The text has been corrected.

(5) Page 12, line 11: For the sentence “Differences in Pol III occupancy thus explain nearly all of the variation in mature tRNA abundance in human cells, suggesting that post-transcriptional

mechanisms contribute little to controlling the levels of individual tRNAs in physiological settings.” I would recommend removing the last three words “in physiological settings” because it invokes a general principle that is maybe too overstated given the singular differentiation system that is used in this study.

We appreciate the opportunity to clarify our phrasing; we meant “physiological” as “non-pathological” conditions (e.g. disease-linked mutations in tRNA-processing or tRNA-modifying enzymes). We have now modified the text to “*post-transcriptional mechanisms contribute little to controlling the levels of individual tRNAs in hiPSC and during their differentiation*”.

(6) Figure 5e, h: The system presented to evaluate the elements involved in tRNA expression is a great setup. However, the two effects – basal expression difference and inducibility – are difficult to discern the way the data is shown. It would (if we take the right panels in Figure 5e as an example) be good to show the quantified fold-change between NPC-wt and hiPSC-wt for Pro-TGG-2-1 and compare it to the FC for the edited version (I can see the latter is lower in basal state, but the precise FC numbers are difficult to estimate from the graph). Moreover, I am wondering if it would be nice to have another panel next to the existing ones that shows the quantification of endogenous Pro-TGG-1-1. This would give the possibility to compare the response for the endogenous/real gene vs. the edited copy. In the end, this could help to give the reader a better idea of what is actually most strongly affected by the edits: basal expression or inducibility? Can the ensemble of intragenic and upstream sequence features explain the full effect?

Thank you for the suggestion; we have now modified Fig. 5 to include this information. We have added panels next to the existing ones that show tRNA-mapped RPC1 ChIP-Seq read fractions at the unedited *tRNA-Pro-TGG-1-1* on chromosome 14. These panels show comparable RPC1 occupancy between wild-type and edited cells, and efficient silencing of this gene in the edited NPC lines. In response to a request from Reviewer #3, the revised manuscript also includes BRF1 occupancy data for the edited cell lines, which mirrors the changes in RPC1 occupancy we observed at *tRNA-Pro-TGG-2-1* (**Extended Data Fig. 5h**). We have also plotted fold changes in the fraction of tRNA-mapped RPC1 ChIP-Seq reads at the edited *tRNA-Pro-TGG-2-1* locus (on chromosome 11) between:

- i) wild-type NPC vs wild-type hiPSC (to measure inducibility)
- ii) edited hiPSC vs wild-type hiPSC (to measure editing effects on basal expression), and
- iii) edited NPC vs edited hiPSC (to measure editing effects on inducibility).

These data show that both the gene body edit and the upstream sequence edit reduced *tRNA-Pro-TGG-2-1* inducibility in NPC to a similar extent; however, basal expression of *tRNA-Pro-TGG-2-1* in hiPSC was more strongly reduced by the gene body edit (to 0.53-fold of wild-type) than by

the upstream sequence edit (0.8-fold of wild-type). While it is tempting to speculate that gene body sequences may thus be more important than upstream sequences for basal tRNA expression, we have refrained from generalizing based on data from editing a single tRNA locus. We note that constructing biallelically edited hiPSC lines and deriving NPC from these lines is an extremely time-consuming process, and the approach we used of editing endogenous loci runs the risk of perturbing cellular tRNA pools to an extent that becomes incompatible with cell viability. The question of whether the ensemble of intragenic and upstream sequence features can explain the full effect is probably also best answered in future work where such edits are combined in non-endogenous tRNA genes (see also response to Reviewer #3).

(7) Page 19, line 5 (from bottom): “AIM2 [...] and CADM3 [...], both of which are conserved in vertebrates.” – it should be stated more clearly if this means that the position/constellation is conserved or if the genes themselves are conserved genes.

We have now clarified in the main text that “*the genomic colocalization of CADM3 and tRNA-Arg-TCT-4-1 is conserved across vertebrates*”.

(8) Page 20/21, in the section on RPC7a loss, I was wondering if it would be better for the flow of the presentation to move up the last part (Extended Data Fig. 6, f-g) and present it first, before the other data in this section?

We thank the reviewer for the suggestion; we chose to first describe the tRNA gene repertoire bound by RPC7a-containing Pol III complexes in wild-type hiPSC before defining the consequences of RPC7a depletion by CRISPRi for Pol III occupancy at tRNA genes.

(9) Discussion, Page 25: I am not quite convinced by the “energy expenditure” argument. Why would it be much more costly to have more diverse repertoires of tRNAs as compared to a more restricted housekeeping set (which, in compensation, would also need to be expressed at higher levels).

We appreciate the opportunity to clarify our reasoning. Our spike-in normalized ChIP-Seq analysis shows that Pol III occupancy decreases at nearly all tRNA genes upon differentiation, albeit to a smaller extent at the highly expressed housekeeping tRNA genes (see S-shaped curve in **Fig. 4f**). This global reduction in Pol III activity (mediated by MAF1) is consistent with a decrease in energy expenditure, a model originally put forward by the Hernandez and Willis labs based on careful analysis of metabolic inefficiencies and the lean phenotype of *Maf1*^{-/-} mice. Globally reduced Pol III activity would also lead to less diverse tRNA repertoires, as the tRNA genes selectively repressed upon MAF1 activation in differentiated cells have weaker promoters than housekeeping tRNA genes. As a result, the proportion of mature tRNAs transcribed from housekeeping genes increases in differentiated cells. We note that our measurements of mature tRNA abundance are relative, as they represent each tRNA transcript as a proportion of all tRNA-mapped reads (in **Fig. 2, b-c** and **Fig. 3c**) or a fold-change relative to hiPSC from DESeq2 (in **Fig.**

1, f and h). We chose not to use spike-ins in an attempt to obtain absolute tRNA abundance measurements because unlike DNA content, which is identical among diploid cells, RNA content varies substantially depending on cell type and developmental stage. Interestingly, Pol III also synthesizes the 5S ribosomal rRNA, and ribosome levels also decrease during neurogenesis through a combination of decreased Pol I rRNA transcription and ribosomal protein synthesis (Chau et al., 2018 PMID: 29745900; Harnett et al., 2022 PMID: 36482253). It is conceivable that this coupling via Pol III may help adjust the overall abundance of ribosomes and tRNAs in response to growth cues while maintaining their stoichiometry.

Reviewer #3:

Gao et al. submitted a manuscript describing RNA polymerase (Pol) III-driven expression patterns of human tRNAs during cell differentiation. The authors used human induced pluripotent stem cells for differentiation into isogenic cardiomyocytes and neuronal cells. Using their recently published mim-tRNAseq technology, they determined genome-wide tRNA expression before, during, and after differentiation. They found that a subset of tRNAs is expressed in all cell types, subsequently termed "housekeeping", a subset of genes is "inactive", and another is "repressed", representing tRNA genes expressed in iPSCs but downregulated in at least one of the differentiated cell types. ChIP-seq with anti-RPC1 antibodies showed that tRNA transcription is well reflected by the presence of Pol III. Consequently, only housekeeping genes of Pol III were recognized after differentiation. In neurons, a particular case of tRNA expression was reported, namely the tRNA-Arg-TCT-4-1 gene located near the neuron-specific *CADM3* gene. The expression of this tRNA gene was dramatically upregulated in neurons. Although strong changes in tRNA expression were observed during differentiation, the overall abundance of anticodon families remained generally stable. Nevertheless, individual cell types could be determined based on tRNA expression patterns. Codon usage and translation speed did not change with altered tRNA expression. Actively transcribed tRNA genes were nucleosome-free and surrounded by trimethylated H3K4-containing nucleosomes (NFRs) in all cell types before and after differentiation. Repressed genes lost H3K4me3 and NFRs after differentiation but gained H3K27me3 instead, the latter being absent at housekeeping tRNA genes. Inactive tRNA genes were not associated with either H3K4me3 or H3K27me3. Moreover, they were embedded in

nucleosome-containing chromatin, as expected. Expression of tRNA genes correlated with A- and B-box consensus sequences. In addition, sequences upstream of the transcription start site influenced differentiation-dependent expression patterns. Differentiation-dependent downregulation of embryonic stem cell-specific POLR3G subunit expression of Pol III did not explain the changes in tRNA transcription. Finally, the authors showed that TORC1-regulated MAF1 activity correlated well with the expression of the repressed tRNAs.

The data presented in this manuscript are important for understanding a possible influence of tRNA transcription on cell differentiation. The cellular system is well chosen and the data are

convincing. The authors did not find any obvious influences of changes in tRNA transcription on differentiation, as anticodon family expression remains stable. However, the notion that different cell types can be identified by their respective tRNA pools suggests that additional mechanisms, such as tRNA modification, may be required to generate tRNA pools adapted to cell type-specific translation requirements. The manuscript presented here contains a wealth of data that will be of great importance for future studies of this type. Therefore, I recommend publishing these data after considering some minor changes.

We highly appreciate the reviewer's positive feedback on the importance of our data.

Page 14, line 7: The authors state that RPC1 is not bound to one third of the predicted tRNAs in immortalized cell lines. This should be toned down to HEK cells, which is the only immortalized cell line analyzed here.

We apologize for the general language. We have now revised the text to "*in two independent hiPSC lines or the immortalized HEK293T*".

Page 16, line 2: Although most tRNA genes are either intragenic or near coding genes I wonder if the assumption that most tRNA genes are intragenic is correct? The authors are certainly right about the phrase "or near coding genes," especially since "near" was not defined, but the wording of the sentence is misleading.

Thank you for the opportunity to clarify; we have now revised the text to include tRNA gene numbers and to be more precise of what distance we consider "near" coding genes (on page 17 of the revised manuscript):

"Although half of the tRNAs with gene-resolved RPC1 ChIP data are either intragenic (234/558, 42%) or near coding genes (≤ 500 bp; 44/558, 8%), their linear distance from active or inactive Pol II genes was not related to RPC1 occupancy (Extended Data Fig. 4d)."

We note that "intragenic" here refers to any sequence within an annotated coding gene (introns, exons, UTRs).

The tRNA and 5'-flanking sequence swapping experiments would be strengthened by BRF1 ChIP results. Moreover, the combined swapping of the corresponding tRNA genes with their 5'-flanking sequences could reveal whether they act independently or synergistically.

To strengthen our conclusions, we have now generated **new data** by performing BRF1 ChIP-Seq in the edited hiPSC and NPC lines. These data confirm that the specific reduction in RPC1 occupancy at the *tRNA-Pro-TGG-2-1* locus when its upstream region is replaced with that of

tRNA-Pro-TGG-1-1 is mirrored by a reduction in BRF1 occupancy (**Extended Data Fig. 5h**). In response to a request from Reviewer #2, we have also included RPC1 and BRF1 ChIP occupancy at the *tRNA-Pro-TGG-1-1* locus to demonstrate that it is unchanged by the edits in *tRNA-Pro-TGG-2-1*.

We agree that it will be interesting to determine whether tRNA gene body and upstream regions act independently or synergistically; however, we note that the tRNA gene body edit already reduced *tRNA-Pro-TGG-2-1* occupancy in hiPSC by 2-fold. As this tRNA encodes a major isodecoder (**Fig. 3d**), it is likely that the further reduction in its expression that could result from also editing its upstream region would be detrimental to cell viability in hiPSC and/or NPC by depleting mature tRNA-Pro-UGG. We elected not to perform the combined swapping experiment suggested both for this reason and because generating this line, deriving NPCs, and performing the molecular analyses would add more than 6 months to our revision timeline. Future work to define the relationship between gene body and upstream regions in governing tRNA gene activity may benefit from an approach that relies on “designer tRNA genes” (with distinct combinations of gene body and upstream regions) inserted in the vicinity of active tRNA genes, rather than on editing endogenous ones. Such an approach would enable the systematic dissection of the regulatory interplay between gene body and upstream sequences while avoiding cellular tRNA pool perturbations.

On page 20, Gao et al. discuss the possibility that tRNA expression is regulated simply by being in proximity to enhancers. For the tRNA Arg-4-1 gene, they show data that support this theory. The authors should also mention whether the expression of the 37 housekeeping and 14 repressed tRNA genes also depends on enhancer activities. In particular, does regulation of the repressed tRNA genes coincide with altered enhancer activity?

In response to this suggestion, we have now expanded our analysis of enhancer-tRNA overlap and its potential regulatory roles. While we believe that enhancer-based regulation of tRNA abundance is very rare, we have now included **new data** in **Supplementary Table S3** to better assess how often it may occur. These data include annotation of *in vivo* enhancer activity based on FANTOM5 CAGE detection of capped enhancer RNAs transcribed by Pol II, which has been performed across “432 primary cell, 135 tissue and 241 cell line samples” and thus covers a wide range of human samples (Andersson et al., 2014 PMID: 24670763). We then focused on the 27 tRNA genes (out of 55 in total) that overlap with transcribed (and therefore active) enhancers. For those, we analyzed whether changes in RPC1 occupancy at the tRNA gene(s) overlapping an active enhancer correlates with changes in mRNA abundance of the coding genes predicted to be its targets. We found such a correlation for *tRNA-Lys-TTT-3-1* and *tRNA-Lys-TTT-3-2*, which overlap an enhancer of *PIK3C2B* and *MDM4* (**Extended data Fig. 5l**). RPC1 occupancy at these two tRNA genes increased in NPC and neurons, along with an increase in *PIK3C2B* and *MDM4* mRNA levels in the same two cell types (page 21-22 of the revised manuscript):

“Among the remaining tRNA genes that overlap transcribed enhancers, tRNA-Lys-TTT-3-1 and tRNA-Lys-TTT-3-2 may also be co-regulated with enhancer targets in NPC and neurons (Extended Data Fig. 5I), but this appears to be a rare regulatory mechanism based on our dataset.”

We note that the effect size of these increases was much smaller than what we observed for tRNA-Arg-TCT-4-1 and CADM3, and we found no other tRNA genes co-regulated with the targets of their overlapping enhancers in the cell contexts we tested here. We have added the following text to the discussion:

“Despite the strong correlation between H3K4me3 and RPC1 ChIP signal at tRNA genes in our datasets and in prior studies⁶³, we found no clear association of tRNA gene activity with Pol II transcription of nearby coding genes. In very rare cases, however – such as we propose for tRNA-Arg-TCT-4-1 – an overlap with an enhancer element may boost the expression of an individual tRNA gene in specific cell contexts. Long-range regulatory DNA interactions, rather than the linear distance to Pol II genes, could thus lead to altered expression of specific tRNA genes in defined cell types. We found some evidence for a potential similar mode of regulation only for tRNA-Lys-TTT-3-1 and tRNA-Lys-TTT-3-2 in NPC and neurons, and it remains possible that other tRNA loci we found to overlap with predicted enhancers may be differentially expressed in cellular contexts where these enhancers are active.”

The paragraph on "global restructuring of tRNA genes upon differentiation" would benefit from the inclusion of H3K9me3 data and/or DNA methylation data. Without such data, it remains unexplained why "inactive tRNA genes" are not transcribed.

We agree with the reviewer that it would be beneficial to determine whether, in addition to their degenerate A- and B-boxes and polyT stretches in upstream regions (**Fig. 5d** and **Extended Data Fig. 5g**), inactive tRNA genes have distinct chromatin marks or a high degree of DNA methylation. To address this question, we have now generated **new data** for H3K9me3 in hiPSC, NPC, and neurons. We validated the quality of the H3K9me3 ChIP data by confirming signal enrichment at centromeric regions, where H3K9me3 marks satellite repeats (**Reviewer Fig. 1**). By contrast, we observed only a weak enrichment of this histone mark at inactive tRNA genes when compared to housekeeping or repressed tRNA loci (**Fig. 5a, Extended Data Fig. 4a**).

Reviewer Figure 1. H3K9me3 ChIP-Seq signal on human chromosome 7 from one biological replicate of hiPSC, NPC, and neurons scaled to reads-per-million (rpm). Centromere region shown by ideogram constriction at q11.1.

With regard to DNA methylation, less than 15% of predicted tRNA genes are covered by commercial DNA methylation arrays (Acton et al., 2021; PMID: 33976121) and whole-genome bisulfite sequencing (WGBS) is prohibitively expensive as it requires a 30x genome coverage with reads of >100 bp (<https://www.encodeproject.org/data-standards/wgbs/>). We reasoned that because of the near-complete overlap of both inactive and Pol III-occupied tRNA genes we found between two independent hiPSC lines (**Extended Data Fig. 3f**), we could examine publicly available WGBS data from human embryonic stem cells (ESC) to investigate whether DNA methylation status correlates with tRNA gene activity. We therefore performed **new analysis** of high-coverage WGBS datasets from two biological replicates of the H1 human ESC line available from ENCODE. We found a strong and significant enrichment for

methylated CpG dinucleotides in inactive tRNA genes and their upstream regions (-125 bp relative to gene start) when compared to housekeeping and repressed tRNAs, which were largely devoid of CpG methylation.

We describe these new results on page 16 of the revised manuscript:

“By contrast, constitutively inactive tRNA genes were found in closed chromatin not marked by either H3K4me3 or H3K27me3, and had only weak H3K9me3 enrichment (Fig. 5a; Extended Data Fig. 4a). Analysis of high-coverage whole-genome bisulfite sequencing datasets from the H1 human embryonic stem cell (hESC) line in ENCODE⁷⁰ revealed significantly higher ($p < 2.22e-16$) and near-complete CpG methylation at inactive tRNA loci when compared to tRNA genes occupied by Pol III in hiPSC (Extended Data Fig. 4c). DNA methylation may thus contribute to silencing inactive tRNA genes.”

Decision Letter, first revision:

29th September 2023

Dear Dr. Nedialkova,

Thank you for submitting your revised manuscript "Selective gene expression maintains human tRNA anticodon pools during differentiation" (NCB-A51199A). It has now been seen by the original referees and their comments are below. The reviewers find that the paper has improved in revision, and therefore we'll be happy in principle to publish it in Nature Cell Biology, pending minor revisions to satisfy the referees' final requests and to comply with our editorial and formatting guidelines.

Please ensure that all figures fit into a single standard page and adhere to a maximum page size of roughly 180mm wide x 200mm high, but also please use the full page space to fill the figure. At present several figures are too tiny to be legible once re-sized during the production process. To ensure legibility once figures are re-sized, please use a font size of no smaller than 6pt Arial or Helvetica throughout the figures.

Thank you again for your interest in Nature Cell Biology Please do not hesitate to contact me if you have any questions.

Sincerely,

Sabrya Carim, PhD
(she/her/hers)
Associate Editor, Nature Cell Biology
Nature Portfolio

Springer Nature
The Campus, 4 Crinan Street, London N1 9XW, UK
sabrya.carim@springernature.com
<https://orcid.org/0000-0001-9485-1938>

Reviewer #1 (Remarks to the Author):

The authors adequately addressed my comments.

Reviewer #2 (Remarks to the Author):

The authors have answered all my comments in a satisfactory fashion. I have no additional points to raise.

Reviewer #3 (Remarks to the Author):

The authors of the revised manuscript have responded to my full satisfaction to the points I raised in my initial review. The new results and discussion of the points mentioned in the reviews have further improved the manuscript. It now represents a very important work in the field of tRNA expression during cellular differentiation. I support publication of the manuscript.

Final Decision Letter:

Dear Dr Nedialkova,

I am pleased to inform you that your manuscript, "Selective gene expression maintains human tRNA anticodon pools during differentiation", has now been accepted for publication in Nature Cell Biology. Congratulations!

Once your paper has been scheduled for online publication, the Nature press office will be in touch to confirm the details. An online order form for reprints of your paper is available at <https://www.nature.com/reprints/author-reprints.html>. All co-authors, authors' institutions and

authors' funding agencies can order reprints using the form appropriate to their geographical region.

Please note that *Nature Cell Biology* is a Transformative Journal (TJ). Authors may publish their research with us through the traditional subscription access route or make their paper immediately open access through payment of an article-processing charge (APC). Authors will not be required to make a final decision about access to their article until it has been accepted. Find out more about Transformative Journals

If you have not already done so, we strongly recommend that you upload the step-by-step protocols used in this manuscript to the Protocol Exchange (www.nature.com/protocolexchange), an open online resource established by Nature Protocols that allows researchers to share their detailed experimental know-how. All uploaded protocols are made freely available, assigned DOIs for ease of citation and are fully searchable through nature.com. Protocols and Nature Portfolio journal papers in which they are used can be linked to one another, and this link is clearly and prominently visible in the online versions of both papers. Authors who performed the specific experiments can act as primary authors for the Protocol as they will be best placed to share the methodology details, but the Corresponding Author of the present research paper should be included as one of the authors. By uploading your Protocols to Protocol Exchange, you are enabling researchers to more readily reproduce or adapt the methodology you use, as well as increasing the visibility of your protocols and papers. You can also establish a dedicated page to collect your lab Protocols. Further information can be found at www.nature.com/protocolexchange/about

With kind regards,

Sabrya Carim, PhD
(she/her/hers)
Associate Editor, Nature Cell Biology
Nature Portfolio

Springer Nature
The Campus, 4 Crinan Street, London N1 9XW, UK
sabrya.carim@springernature.com
<https://orcid.org/0000-0001-9485-1938>

** Visit the Springer Nature Editorial and Publishing website at www.springernature.com/editorial-and-publishing-jobs for more information about our career opportunities. If you have any questions please click here.**